# 2Direction: **Theoretically Faster Distributed Training with Bidirectional Communication Compression**

**Alexander Tyurin**
KAUST
Saudi Arabia
alexandertiurin@gmail.com

**Peter Richtárik**
KAUST
Saudi Arabia
richtarik@gmail.com

## Abstract

We consider distributed convex optimization problems in the regime when the communication between the server and the workers is expensive in both uplink and downlink directions. We develop a new and provably accelerated method, which we call 2Direction, based on fast bidirectional compressed communication and a new bespoke error-feedback mechanism which may be of independent interest. Indeed, we find that the EF and EF21-P mechanisms (Seide et al., 2014; Gruntkowska et al., 2023) that have considerable success in the design of efficient non-accelerated methods are not appropriate for accelerated methods. In particular, we prove that 2Direction improves the previous state-of-the-art communication complexity $\widetilde{\Theta}\left(K \times (L/_{\alpha\mu} + L_{\max}\omega/_{n\mu} + \omega)\right)$ (Gruntkowska et al., 2023) to $\widetilde{\Theta}(K \times (\sqrt{L(\omega+1)/_{\alpha\mu}} + \sqrt{L_{\max}\omega^2/_{n\mu}} + 1/_\alpha + \omega))$ in the $\mu$–strongly-convex setting, where $L$ and $L_{\max}$ are smoothness constants, $n$ is # of workers, $\omega$ and $\alpha$ are compression errors of the $\mathrm{Rand}K$ and $\mathrm{Top}K$ sparsifiers (as examples), $K$ is # of coordinates/bits that the server and workers send to each other. Moreover, our method is the first that improves upon the communication complexity of the vanilla accelerated gradient descent (AGD) method (Nesterov, 2018). We obtain similar improvements in the general convex regime as well. Finally, our theoretical findings are corroborated by experimental evidence.

## 1 Introduction

We consider convex optimization problems in the centralized distributed setting. These types of problems appear in federated learning (Konečný et al., 2016; McMahan et al., 2017) and distributed optimization (Ramesh et al., 2021). In this setting, one of the main problems is the communication bottleneck: the connection link between the server and the workers can be very slow. We focus our attention on methods that aim to address this issue by applying *lossy compression* to the communicated messages (Alistarh et al., 2017; Mishchenko et al., 2019; Gruntkowska et al., 2023).

### 1.1 The problem

Formally, we consider the optimization problem

$$\min_{x \in \mathbb{R}^d} \left\{ f(x) := \tfrac{1}{n} \sum_{i=1}^{n} f_i(x) \right\}, \tag{1}$$

where $n$ is the number of workers and $f_i : \mathbb{R}^d \to \mathbb{R}$ are smooth convex functions for all $i \in [n] := \{1, \ldots, n\}$. We consider the *centralized distributed optimization* setting in which each $i^{\text{th}}$ worker contains the function $f_i$, and all workers are directly connected to a server (Kairouz et al., 2021). In general, we want to find a (possibly random) point $\widehat{x}$ such that $\mathbb{E}\left[f(\widehat{x})\right] - f(x^*) \leq \varepsilon$, where $x^*$ is

an optimal point. In the strongly convex setup, we also want to guarantee that $\mathbb{E}[\|\widetilde{x} - x^*\|^2] \leq \varepsilon$ for some point $\widetilde{x}$.

Virtually all other theoretical works in this genre assume that, compared to the worker-to-server (w2s) communication cost, the server-to-workers (s2w) broadcast is so fast that it can be ignored. We lift this limitation and instead associate a relative cost $r \in [0, 1]$ with the two directions of communication. If $r = 0$, then s2w communication is free, if $r = 1$, then w2s communication is free, and if $r = 1/2$, then the s2w and w2s costs are equal. All our theoretical results hold for any $r \in [0, 1]$. We formalize and elaborate upon this setup in Section 2.

### 1.2 Assumptions

Throughout the paper we rely on several standard assumptions on the functions $f_i$ and $f$.

**Assumption 1.1.** Functions $f_i$ are $L_i$–smooth, i.e., $\|\nabla f_i(x) - \nabla f_i(y)\| \leq L_i \|x - y\|$ for all $x, y \in \mathbb{R}^d$, for all $i \in [n]$. We let $L_{\max} := \max_{i \in [n]} L_i$. Further, let $\widehat{L} > 0$ be a constant such that $\frac{1}{n} \sum_{i=1}^n \|\nabla f_i(x) - \nabla f_i(y)\|^2 \leq \widehat{L}^2 \|x - y\|^2$ for all $x, y \in \mathbb{R}^d$.
Note that if the functions $f_i$ are $L_i$–smooth for all $i \in [n]$, then $\widehat{L} \leq L_{\max}$.

**Assumption 1.2.** Function $f$ is $L$–smooth, i.e., $\|\nabla f(x) - \nabla f(y)\| \leq L \|x - y\|$ for all $x, y \in \mathbb{R}^d$.

**Assumption 1.3.** Functions $f_i$ are convex for all $i \in [n]$, and $f$ is $\mu$-strongly convex with $\mu \geq 0$, attaining a minimum at some point $x^* \in \mathbb{R}^d$.

It is known that the above smoothness constants are related in the following way.

**Lemma 1.4** (Gruntkowska et al. (2023)). *If Assumptions 1.2, 1.1 and 1.3 hold, then $\widehat{L} \leq L_{\max} \leq nL$ and $L \leq \widehat{L} \leq \sqrt{L_{\max}L}$.*

## 2  Motivation: From Unidirectional to Bidirectional Compression

In this work, we distinguish between worker-to-server (w2s=uplink) and server-to-worker (s2w=downlink) communication cost, and define w2s and s2w communication complexities of methods in the following natural way.

**Definition 2.1.** For a centralized distributed method $\mathcal{M}$ aiming to solve problem (1), the communication complexity $\mathfrak{m}_{\mathcal{M}}^{\text{w2s}}$ is the expected number of coordinates/floats[1] that each worker sends to the server to solve problem (1). The quantity $\mathfrak{m}_{\mathcal{M}}^{\text{s2w}}$ is the expected number of floats/coordinates the server broadcasts to the workers to solve problem (1). If $\mathfrak{m}_{\mathcal{M}}^{\text{w2s}} = \mathfrak{m}_{\mathcal{M}}^{\text{s2w}}$, then we use the simplified notation $\mathfrak{m}_{\mathcal{M}} := \mathfrak{m}_{\mathcal{M}}^{\text{s2w}} = \mathfrak{m}_{\mathcal{M}}^{\text{w2s}}$.

Let us illustrate the above concepts on the simplest baseline: vanilla gradient descent (GD). It is well known (Nesterov, 2018) that for $L$–smooth, $\mu$–strongly convex problems, GD returns an $\varepsilon$-solution after $\mathcal{O}\left(L/\mu \log 1/\varepsilon\right)$ iterations. In each iteration, the workers and the server communicate all $\Theta(d)$ coordinates to each other (since no compression is applied). Therefore, the communication complexity of GD is $\mathfrak{m}_{\text{GD}} = \Theta\left(dL/\mu \log 1/\varepsilon\right)$. The same reasoning applies to the accelerated gradient method (AGD) (Nesterov, 2018), whose communication complexity is $\mathfrak{m}_{\text{AGD}} = \Theta(d\sqrt{L/\mu} \log 1/\varepsilon)$.

### 2.1 Compression mappings

In the literature, researchers often use the following two families of compressors:

**Definition 2.2.** A (possibly) stochastic mapping $\mathcal{C} : \mathbb{R}^d \to \mathbb{R}^d$ is a *biased compressor* if there exists $\alpha \in (0, 1]$ such that

$$\mathbb{E}\left[\|\mathcal{C}(x) - x\|^2\right] \leq (1 - \alpha) \|x\|^2, \qquad \forall x \in \mathbb{R}^d. \tag{2}$$

**Definition 2.3.** A stochastic mapping $\mathcal{C} : \mathbb{R}^d \to \mathbb{R}^d$ is an *unbiased compressor* if there exists $\omega \geq 0$ such that

$$\mathbb{E}[\mathcal{C}(x)] = x, \qquad \mathbb{E}\left[\|\mathcal{C}(x) - x\|^2\right] \leq \omega \|x\|^2, \qquad \forall x \in \mathbb{R}^d. \tag{3}$$

---

[1]Some works measure bits instead of coordinates. For computer systems, where coordinates are represented by 32 or 64 bits, these measures are equivalent up to the constant factors 32 or 64.

Table 1: **Communication Rounds in the Strongly Convex Case.** The number of communication rounds and rounds costs to get an $\varepsilon$-solution ($\mathbb{E}\left[\|\widehat{x} - x^*\|^2\right] \leq \varepsilon$) up to logarithmic factors. The table shows the most relevant bidirectional compressed methods that are ordered by the total communication complexity # **Communication Rounds** × **Round Cost** (see (4) for details).

 i. The parameter $r$ weights the importance/speed of uplink and downlink connections. When $r = 1/2$, it means that the uplink and downlink speeds are equal.

 ii. The parameters $K_\omega$ and $K_\alpha$ are the expected densities Definition 2.5 of compressors $\mathcal{C}^D \in \mathbb{U}(\omega)$ and $\mathcal{C}^P \in \mathbb{B}(\alpha)$[a], that operate in the workers and the server accordingly. Less formally, $K_\omega$ and $K_\alpha$ are the number of coordinates/bits that the workers and the server send to each other in each communication round.

| Method | # Communication Rounds | Round Cost[c] |
|---|---|---|
| Dore, Artemis, MURANA[a] (Liu et al., 2020) (Philippenko and Dieuleveut, 2020) (Condat and Richtárik, 2022) | $\widetilde{\Omega}\left(\frac{\omega}{\alpha n}\frac{L_{\max}}{\mu}\right)$[f] | $(1-r)K_\omega + rK_\alpha$ |
| MCM[a] (Philippenko and Dieuleveut, 2021) | $\widetilde{\Omega}\left(\left(\frac{1}{\alpha^{3/2}} + \frac{\omega^{1/2}}{\alpha\sqrt{n}} + \frac{\omega}{n}\right)\frac{L_{\max}}{\mu}\right)$[f] | $(1-r)K_\omega + rK_\alpha$ |
| GD (Nesterov, 2018) | $\frac{L}{\mu}$ | $d$ |
| EF21-P + DIANA (Gruntkowska et al., 2023) | $\frac{L}{\alpha\mu} + \frac{L_{\max}\omega}{n\mu} + \omega$ | $(1-r)K_\omega + rK_\alpha$ |
| AGD (Nesterov, 2018) | $\sqrt{\frac{L}{\mu}}$ | $d$ |
| 2Direction (Remark 5.3)[b], (Theorem 5.2) | $\sqrt{\frac{L(\omega+1)}{\alpha\mu}} + \sqrt{\frac{L_{\max}\omega^2}{n\mu}} + \frac{1}{\alpha} + \omega$ | $(1-r)K_\omega + rK_\alpha$ |
| 2Direction (Remark 5.5)[b], (Theorem 5.4) (requires $L_{\max}/L$)[d] | $\sqrt{\frac{L\max\{1, r(\omega+1)\}}{\alpha\mu}} + \sqrt{\frac{L^{2/3}L_{\max}^{1/3}(\omega+1)}{\alpha n^{1/3}\mu}} + \sqrt{\frac{L^{1/2}L_{\max}^{1/2}(\omega+1)^{3/2}}{\sqrt{\alpha n}\mu}} + \sqrt{\frac{L_{\max}\omega^2}{n\mu}} + \frac{1}{\alpha} + \omega$ | $(1-r)K_\omega + rK_\alpha$ |

[a] The Dore, Artemis, MURANA, and MCM methods do not support *biased* compressors for server-to-worker compression. In these methods, the error $\alpha$ equals $1/(\omega_s+1)$, where the error $\omega_s$ is a parameter of an unbiased compressor that is used in server-to-worker compression. For these methods, we define $1/(\omega_s+1)$ as $\alpha$ to make comparison easy with EF21-P + DIANA and 2Direction.

[b] In this table, we present the simplified iteration complexity of 2Direction assuming that $r \leq 1/2$ and $\omega+1 = \Theta\left(d/K_\omega\right)$. The full complexities are in (13) and (14). In Section 6, we show that 2Direction has no worse total communication complexity than EF21-P + DIANA for all $r \in [0,1]$ and for any choice of compressors.

[c] We define **Round Cost** of a method $\mathcal{M}$ as a constant such that $\mathfrak{m}_{\mathcal{M}}^r = $ # **Communication Rounds** × **Round Cost**, where $\mathfrak{m}_{\mathcal{M}}^r$ is the total communication complexity (4).

[d] 2Direction can have even better total communication complexity if the algorithm can use the ratio $L_{\max}/L$ when selecting the parameters $\tau$ and $p$ in Algorithm 1. For instance, this is the case if we assume that $L_{\max} = L$, which was done by Li et al. (2020); Li and Richtárik (2021), for example.

[f] The notation $\widetilde{\Omega}\left(\cdot\right)$ means "at least up to logarithmic factors."

We will make use of the following assumption.

**Assumption 2.4.** The randomness in all compressors used in our method is drawn independently.

Let us denote the set of mappings satisfying Definition 2.2 and 2.3 by $\mathbb{B}(\alpha)$ and $\mathbb{U}(\omega)$, respectively. The family of biased compressors $\mathbb{B}$ is wider. Indeed, it is well known if $\mathcal{C} \in \mathbb{U}(\omega)$, then $1/(\omega+1) \cdot \mathcal{C} \in \mathbb{B}\left(1/(\omega+1)\right)$. The canonical sparsification operators belonging to these classes are Top$K \in \mathbb{B}(K/d)$ and Rand$K \in \mathbb{U}(d/K - 1)$. The former outputs the $K$ largest values (in magnitude) of the input vector, while the latter outputs $K$ random values of the input vector, scaled by $d/K$ (Beznosikov et al., 2020). Following (Gorbunov et al., 2021; Tyurin and Richtárik, 2023), we now define the *expected density* of a sparsifier as a way to formalize its *compression* performance.

**Definition 2.5.** The expected density of a sparsifier $\mathcal{C} : \mathbb{R}^d \to \mathbb{R}^d$ is the quantity $K_\mathcal{C} := \sup_{x \in \mathbb{R}^d} \mathbb{E}\left[\|\mathcal{C}(x)\|_0\right]$, where $\|y\|_0$ is the number of of non-zero components of $y \in \mathbb{R}^d$.

Trivially, for the Rand$K$ and Top$K$ sparsifiers we have $K_\mathcal{C} = K$.

## 2.2 Unidirectional (i.e., w2s) compression

As mentioned in the introduction, virtually all theoretical works in the area of compressed communication ignore s2w communication cost and instead aim to minimize $\mathfrak{m}_{\mathcal{M}}^{w2s}$. Algorithmic work related

to methods that only perform w2s compression has a long history, and this area is relatively well understood (Alistarh et al., 2017; Mishchenko et al., 2019; Richtárik et al., 2021).

We refer to the work of Gruntkowska et al. (2023) for a detailed discussion of the communication complexities of *non-accelerated* methods in the convex and non-convex settings. For instance, using $\text{Rand}K$, the DIANA method of Mishchenko et al. (2019) provably improves[2] the communication complexity of GD to $\mathfrak{m}_{\text{DIANA}}^{\text{w2s}} = \widetilde{\Theta}\left(d + {}^{KL}/_\mu + {}^{dL_{\max}}/_{n\mu}\right)$. *Accelerated* methods focusing on w2s compression are also well investigated. For example, Li et al. (2020) and Li and Richtárik (2021) developed accelerated methods, which are based on (Mishchenko et al., 2019; Kovalev et al., 2020), and provably improve the w2s complexity of DIANA. Moreover, using $\text{Rand}K$ with $K \leq {}^d/n$, ADIANA improves the communication complexity of AGD to $\mathfrak{m}_{\text{ADIANA}}^{\text{w2s}} = \widetilde{\Theta}(d + d\sqrt{{}^{L_{\max}}/_{n\mu}})$.

### 2.3   Bidirectional (i.e., w2s and s2w) compression

The methods mentioned in Section 2.2 do *not* perform server-to-workers (s2w) compression, and one can show that the server-to-workers (s2w) communication complexities of these methods are worse than $\mathfrak{m}_{\text{AGD}} = \widetilde{\Theta}(d\sqrt{{}^{L}/_\mu})$. For example, using the $\text{Rand}K$, the s2w communication complexity of ADIANA is at least $\mathfrak{m}_{\text{ADIANA}}^{\text{s2w}} = \widetilde{\Omega}(d \times \omega) = \widetilde{\Omega}({}^{d^2}/K)$, which can be ${}^d/K$ times larger than in GD or AGD. Instead of $\mathfrak{m}_{\mathcal{M}}^{\text{w2s}}$, methods performing bidirectional compression attempt to minimize *the total communication complexity*, which we define as a convex combination of the w2s and s2w communication complexities:

$$\mathfrak{m}_{\mathcal{M}}^r := (1 - r)\mathfrak{m}_{\mathcal{M}}^{\text{w2s}} + r\mathfrak{m}_{\mathcal{M}}^{\text{s2w}}. \tag{4}$$

The parameter $r \in [0, 1]$ weights the importance of uplink (w2s) and downlink (s2w) connections[3]. Methods from Section 2.2 assume that $r = 0$, thus ignoring the s2w communication cost. On the other hand, when $r = {}^1/2$, the uplink and downlink communication speeds are equal. By considering any $r \in [0, 1]$, our methods and findings are applicable to more situations arising in practice. Obviously, $\mathfrak{m}_{\text{GD}}^r = \mathfrak{m}_{\text{GD}}$ and $\mathfrak{m}_{\text{AGD}}^r = \mathfrak{m}_{\text{AGD}}$ for all $r \in [0, 1]$. Recently, Gruntkowska et al. (2023) proposed the EF21-P + DIANA method. This is the first method supporting bidirectional compression that provably improves both the w2s and s2w complexities of GD: $\mathfrak{m}_{\text{EF21-P + DIANA}}^r \leq \mathfrak{m}_{\text{GD}}$ for all $r \in [0, 1]$. Bidirectional methods designed before EF21-P + DIANA, including (Tang et al., 2020; Liu et al., 2020; Philippenko and Dieuleveut, 2021), do not guarantee the total communication complexities better than that of GD. The EF21-P + DIANA method is *not* an accelerated method and, in the worst case, can have communication complexities worse than AGD when the condition number ${}^L/_\mu$ is large.

## 3   Contributions

Motivated by the above discussion, in this work we aim to address the following

**Main Problem**:

> **Is it possible to develop a method supporting bidirectional communication compression that improves the current best theoretical total communication complexity of EF21-P + DIANA, and guarantees the total communication complexity to be no worse than the communication complexity $\mathfrak{m}_{\text{AGD}} = \widetilde{\Theta}(d\sqrt{{}^{L}/_\mu})$ of AGD, while improving on AGD in at least some regimes?**

**A)** We develop a new fast method (2Direction; see Algorithm 1) supporting bidirectional communication compression. Our analysis leads to new state-of-the-art complexity rates in the centralized distributed setting (see Table 1), and as a byproduct, we answer Main Problem in the affirmative.

**B)** Gruntkowska et al. (2023) proposed to use the EF21-P error-feedback mechanism (8) to improve the convergence rates of *non-accelerated* methods supporting bidirectional communication compression. EF21-P is a reparameterization of the celebrated EF mechanism (Seide et al., 2014). We tried to use EF21-P in our method as well, but failed. Our failures indicated that a fundamentally

---

[2]Indeed, using Lemma 1.4, $K \leq d$, and $L \geq \mu$, one can easily show that $d + {}^{KL}/_\mu + {}^{dL_{\max}}/_{n\mu} = \mathcal{O}({}^{dL}/_\mu)$.

[3]$\mathfrak{m}_{\mathcal{M}}^r \propto s^{\text{w2s}}\mathfrak{m}_{\mathcal{M}}^{\text{w2s}} + s^{\text{s2w}}\mathfrak{m}_{\mathcal{M}}^{\text{s2w}}$, where $s^{\text{w2s}}$ and $s^{\text{s2w}}$ are connection speeds, and $r = {}^{s^{\text{s2w}}}/_{s^{\text{w2s}}+s^{\text{s2w}}}$.

new approach is needed, and this eventually led us to design a new error-feedback mechanism (9) that is more appropriate for *accelerated* methods. We believe that this is a contribution of independent interest that might motivate future growth in the area.

**C)** Unlike previous theoretical works (Li et al., 2020; Li and Richtárik, 2021) on accelerated methods, we present a unified analysis in both the $\mu$–strongly-convex and general convex cases. Moreover, in the general convex setting and low accuracy regimes, our analysis improves the rate $\mathcal{O}\left(1/\varepsilon^{1/3}\right)$ of Li and Richtárik (2021) to $\mathcal{O}\left(\log 1/\varepsilon\right)$ (see details in Section R).

**D)** Even though our central goal was to obtain new SOTA *theoretical* communication complexities for centralized distributed optimization, we show that the newly developed algorithm enjoys faster communication complexities in practice as well (see details in Section Q).

---

**Algorithm 1** 2Direction: A Fast Gradient Method Supporting Bidirectional Compression

1: **Parameters:** Lipschitz-like parameter $\bar{L} > 0$, strong-convexity parameter $\mu \geq 0$, probability $p \in (0, 1]$, parameter $\Gamma_0 \geq 1$, momentum $\tau \in (0, 1]$, contraction parameter $\alpha \in (0, 1]$ from (2), initial point $x^0 \in \mathbb{R}^d$, initial gradient shifts $h_1^0, \ldots, h_n^0 \in \mathbb{R}^d$, gradient shifts $k^0 \in \mathbb{R}^d$ and $v^0 \in \mathbb{R}^d$

2: Initialize $\beta = 1/(\omega+1)$, $w^0 = z^0 = u^0 = x^0$, and $h^0 = \frac{1}{n}\sum_{i=1}^n h_i^0$

3: **for** $t = 0, 1, \ldots, T-1$ **do**

4:     $\Gamma_{t+1}, \gamma_{t+1}, \theta_{t+1} = \text{CalculateLearningRates}(\Gamma_t, \bar{L}, \mu, p, \alpha, \tau, \beta)$     Get learning rates using Algorithm 2

5:     **for** $i = 1, \ldots, n$ in parallel **do**

6:         $y^{t+1} = \theta_{t+1}w^t + (1 - \theta_{t+1})z^t$

7:         $m_i^{t,y} = \mathcal{C}_i^{D,y}(\nabla f_i(y^{t+1}) - h_i^t)$     Worker $i$ compresses the shifted gradient via the compressor $\mathcal{C}_i^{D,y} \in \mathbb{U}(\omega)$

8:         Send compressed message $m_i^{t,y}$ to the server

9:     **end for**

10:     $g^{t+1} = h^t + \frac{1}{n}\sum_{i=1}^n m_i^{t,y}$

11:     $u^{t+1} = \arg\min_{x\in\mathbb{R}^d} \left\langle g^{t+1}, x \right\rangle + \frac{\bar{L}+\Gamma_t\mu}{2\gamma_{t+1}} \left\| x - u^t \right\|^2 + \frac{\mu}{2} \left\| x - y^{t+1} \right\|^2$     A gradient-like descent step

12:     $q^{t+1} = \arg\min_{x\in\mathbb{R}^d} \left\langle k^t, x \right\rangle + \frac{\bar{L}+\Gamma_t\mu}{2\gamma_{t+1}} \left\| x - w^t \right\|^2 + \frac{\mu}{2} \left\| x - y^{t+1} \right\|^2$

13:     $p^{t+1} = \mathcal{C}^P\left(u^{t+1} - q^{t+1}\right)$     Server compresses the shifted model via the compressor $\mathcal{C}^P \in \mathbb{B}(\alpha)$

14:     $w^{t+1} = q^{t+1} + p^{t+1}$

15:     $x^{t+1} = \theta_{t+1}u^{t+1} + (1 - \theta_{t+1})z^t$

16:     Send compressed message $p^{t+1}$ to all $n$ workers

17:     Flip a coin $c^t \sim \text{Bernoulli}(p)$

18:     $k^{t+1} = \begin{cases} v^t, & c^t = 1 \\ k^t, & c^t = 0 \end{cases}$  and  $z^{t+1} = \begin{cases} x^{t+1}, & c^t = 1 \\ z^t, & c^t = 0 \end{cases}$

19:     **if** $c^t = 1$ **then**

20:         Broadcast non-compressed messages $x^{t+1}$ and $k^{t+1}$ to all $n$ workers     With small probability $p$!

21:     **end if**

22:     **for** $i = 1, \ldots, n$ in parallel **do**

23:         $q^{t+1} = \arg\min_{x\in\mathbb{R}^d} \left\langle k^t, x \right\rangle + \frac{\bar{L}+\Gamma_t\mu}{2\gamma_{t+1}} \left\| x - w^t \right\|^2 + \frac{\mu}{2} \left\| x - y^{t+1} \right\|^2$

24:         $w^{t+1} = q^{t+1} + p^{t+1}$

25:         $z^{t+1} = \begin{cases} x^{t+1}, & c^t = 1 \\ z^t, & c^t = 0 \end{cases}$

26:         $m_i^{t,z} = \mathcal{C}_i^{D,z}(\nabla f_i(z^{t+1}) - h_i^t)$     Worker $i$ compresses the shifted gradient via the compressor $\mathcal{C}_i^{D,z} \in \mathbb{U}(\omega)$

27:         $h_i^{t+1} = h_i^t + \beta m_i^{t,z}$

28:         Send compressed message $m_i^{t,z}$ to the server

29:     **end for**

30:     $v^{t+1} = (1 - \tau)v^t + \tau\left(h^t + \frac{1}{n}\sum_{i=1}^n m_i^{t,z}\right)$

31:     $h^{t+1} = h^t + \beta\frac{1}{n}\sum_{i=1}^n m_i^{t,z}$

32: **end for**

---

## 4  New Method: 2Direction

In order to provide an answer to Main Problem, at the beginning of our research journey we hoped that a rather straightforward approach might bear fruit. In particular, we considered the current state-the-art methods ADIANA (Algorithm 3) (Li et al., 2020), CANITA (Li and Richtárik, 2021) and EF21-P + DIANA (Algorithm 4) (Gruntkowska et al., 2023), and tried to combine the EF21-P compression

---

**Algorithm 2** CalculateLearningRates

---

1: **Parameters:** element $\Gamma_t$; parameter $\bar{L} > 0$; strong-convexity parameter $\mu \geq 0$, probability $p$, contraction parameter $\alpha$, momentum $\tau$, parameter $\beta$
2: Find the largest root $\bar{\theta}_{t+1}$ of the quadratic equation

$$p\bar{L}\Gamma_t\bar{\theta}_{t+1}^2 + p(\bar{L} + \Gamma_t\mu)\bar{\theta}_{t+1} - (\bar{L} + \Gamma_t\mu) = 0$$

3: $\theta_{\min} = \frac{1}{4}\min\left\{1, \frac{\alpha}{p}, \frac{\tau}{p}, \frac{\beta}{p}\right\}$; $\quad \theta_{t+1} = \min\{\bar{\theta}_{t+1}, \theta_{\min}\}$; $\quad \gamma_{t+1} = \frac{p\theta_{t+1}\Gamma_t}{1-p\theta_{t+1}}$; $\quad \Gamma_{t+1} = \Gamma_t + \gamma_{t+1}$

---

mechanism on the server side with the ADIANA (accelerated DIANA) compression mechanism on the workers' side. In short, we were aiming to develop a "EF21-P + ADIANA" method. Note that while EF21-P + DIANA provides the current SOTA communication complexity among all methods supporting bidirectional compression, the method is not "accelerated". On the other hand, while ADIANA (in the strongly convex regime) and CANITA (in the convex regime) are "accelerated", they support unidirectional (uplink) compression only.

In Sections B and C we list the ADIANA and EF21-P + DIANA methods, respectively. One can see that in order to calculate $x^{t+1}$, $y^{t+1}$, and $z^{t+1}$ in ADIANA (Algorithm 3), it is sufficient for the server to broadcast the point $u^{t+1}$. At first sight, it seems that we might be able to develop a "EF21-P + ADIANA" method by replacing Line 15 in Algorithm 3 with Lines 12, 13, 14, and 16 from Algorithm 4. With these changes, we can try to calculate $x^{t+1}$ and $y^{t+1}$ using the formulas

$$y^{t+1} = \theta_{t+1}w^t + (1 - \theta_{t+1})z^t, \tag{5}$$

$$x^{t+1} = \theta_{t+1}w^{t+1} + (1 - \theta_{t+1})z^t \tag{6}$$

$$w^{t+1} = w^t + \mathcal{C}^P\left(u^{t+1} - w^t\right), \tag{7}$$

instead of Lines 7 and 17 in Algorithm 3. Unfortunately, all our attempts of making this work failed, and we now believe that this "naive" approach will not lead to a resolution of Main Problem. Let us briefly explain why we think so, and how we ultimately managed to resolve Main Problem.

♦ The first issue arises from the fact that the point $x^{t+1}$, and, consequently, the point $z^{t+1}$, depend on $w^{t+1}$ instead of $u^{t+1}$, and thus the error from the *primal* (i.e., server) compressor $\mathcal{C}^P$ affects them. In our proofs, we do not know how to prove a good convergence rate with (6). Therefore, we decided to use the original update (Line 17 from Algorithm 3) instead. We can do this almost for free because in Algorithm 3 the point $x^{t+1}$ is only used in Line 18 (Algorithm 3) with small probability $p$. In the final version of our algorithm 2Direction (see Algorithm 1), we broadcast a non-compressed messages $x^{t+1}$ with probability $p$. In Section 6, we show that $p$ is so small that these non-compressed rare messages do not affect the total communication complexity of Algorithm 3.

♦ The second issue comes from the observation that we can not perform the same trick for point $y^{t+1}$ since it is required in each iteration of Algorithm 3. We tried to use (5) and (7), but this still does not work. Deeper understanding of this can only be gained by a detailed examination our proof and the proofs of (Li et al., 2020; Gruntkowska et al., 2023). One way to explain the difficulty is to observe that in non-accelerated methods (Gorbunov et al., 2020; Gruntkowska et al., 2023), the variance-reducing shifts $h^t$ converge to *the fixed vector* $\nabla f(x^*)$, while in the accelerated methods (Li et al., 2020; Li and Richtárik, 2021), these shifts $h^t$ converge to *the non-fixed vectors* $\nabla f(z^t)$ in the corresponding Lyapunov functions. Assume that $\mu = 0$. Then, instead of the EF21-P mechanism

$$w^{t+1} = w^t + \mathcal{C}^P\left(u^{t+1} - w^t\right) \overset{\text{Line 13 in Alg. 3}}{=} w^t + \mathcal{C}^P\left(u^t - \tfrac{\gamma_{t+1}}{\bar{L}}g^{t+1} - w^t\right)$$
$$\overset{\nabla f(x^*)=0}{=} w^t - \tfrac{\gamma_{t+1}}{\bar{L}}\nabla f(x^*) + \mathcal{C}^P\left(u^t - \tfrac{\gamma_{t+1}}{\bar{L}}(g^{t+1} - \nabla f(x^*)) - w^t\right), \tag{8}$$

we propose to perform the step

$$w^{t+1} = w^t - \tfrac{\gamma_{t+1}}{\bar{L}}\nabla f(z^t) + \mathcal{C}^P\left(u^t - \tfrac{\gamma_{t+1}}{\bar{L}}(g^{t+1} - \nabla f(z^t)) - w^t\right) \tag{9}$$
$$= q^{t+1} + \mathcal{C}^P\left(u^{t+1} - q^{t+1}\right), \text{ where}$$

$$u^{t+1} = \underset{x\in\mathbb{R}^d}{\arg\min}\left\langle g^{t+1}, x\right\rangle + \tfrac{\bar{L}}{2\gamma_{t+1}}\|x - u^t\|^2, \ q^{t+1} = \underset{x\in\mathbb{R}^d}{\arg\min}\left\langle \nabla f(z^t), x\right\rangle + \tfrac{\bar{L}}{2\gamma_{t+1}}\|x - w^t\|^2.$$

Unlike (8), step (9) resolves all our previous problems, and we were able to obtain new SOTA rates.

♦ However, step (9) is not implementable since the server and the nodes need to know the vector $\nabla f(z^t)$. The last crucial observation is the same as with the points $x^t$ and $z^t$: the vector $\nabla f(z^t)$ changes with probability $p$ since the point $z^t$ changes with probability $p$. Intuitively, this means that it easier to communicate $\nabla f(z^t)$ between the server and the workers. We do this using two auxiliary control vectors, $v^t$ and $k^t$. The former "learns" the value of $\nabla f(z^t)$ in Line 30 (Algorithm 1), and the latter is used in Line 14 (Algorithm 1) instead of $\nabla f(z^t)$. Then, when the algorithm updates $z^{t+1}$, it also updates $k^t$ in Line 18 (Algorithm 1) and the updated non-compressed vector $k^t$ is broadcast to the workers.

The described changes are highlighted in Lines 6, 12, 13, 14, 18, 20, 23, 24 and 30 of our new Algorithm 1 in green color. Other steps of the algorithm correspond to the original ADIANA method (Algorithm 3). Remarkably, all these new steps are only required to substitute a single Line 15 of Algorithm 3!

## 5 Theoretical Communication Complexity of 2Direction

Having outlined our thought process when developing 2Direction (Algorithm 1), we are now ready to present our theoretical iteration and communication complexity results. Note that 2Direction depends on two hyper-parameters, probability $p$ (used in Lines 4 and 17) and momentum $\tau$ (used in Lines 4 and 30). Further, while Li et al. (2020); Li and Richtárik (2021) assume a strong relationship between $L$ and $L_{\max}$ ($L = L_{\max}$), Gruntkowska et al. (2023) differentiate between $L$ and $L_{\max}$, and thus perform a more general and analysis of their method. In order to perform a fair comparison to the above results, we have decided to minimize the *total communication complexity* $\mathfrak{m}^r_{\mathcal{M}}$ as a function of the hyper-parameters $p$ and $\tau$, depending on whether the ratio $L_{\max}/L$ is known or not.

Defining $R^2 := \left\| x^0 - x^* \right\|^2$, Theorems E.12 and E.14 state that 2Direction converges after

$$T := \widetilde{\Theta}\left( \max\left\{ \sqrt{\frac{L}{\alpha p \mu}}, \sqrt{\frac{L}{\alpha \tau \mu}}, \sqrt{\frac{\sqrt{LL_{\max}}(\omega+1)\sqrt{\omega\tau}}{\alpha\sqrt{n}\mu}}, \sqrt{\frac{\sqrt{LL_{\max}}\sqrt{\omega+1}\sqrt{\omega\tau}}{\alpha\sqrt{p}\sqrt{n}\mu}}, \right.\right.$$
$$\left.\left. \sqrt{\frac{L_{\max}\omega(\omega+1)^2 p}{n\mu}}, \sqrt{\frac{L_{\max}\omega}{np\mu}}, \frac{1}{\alpha}, \frac{1}{\tau}, (\omega+1), \frac{1}{p} \right\}\right), \text{ or} \tag{10}$$

$$T_{\mathrm{gc}} := \Theta\left( \max\left\{ \sqrt{\frac{LR^2}{\alpha p \varepsilon}}, \sqrt{\frac{LR^2}{\alpha \tau \varepsilon}}, \sqrt{\frac{\sqrt{LL_{\max}}(\omega+1)\sqrt{\omega\tau}R^2}{\alpha\sqrt{n}\varepsilon}}, \sqrt{\frac{\sqrt{LL_{\max}}\sqrt{\omega+1}\sqrt{\omega\tau}R^2}{\alpha\sqrt{p}\sqrt{n}\varepsilon}}, \right.\right.$$
$$\left.\left. \sqrt{\frac{L_{\max}\omega(\omega+1)^2 pR^2}{n\varepsilon}}, \sqrt{\frac{L_{\max}\omega R^2}{np\varepsilon}} \right\}\right) + \widetilde{\Theta}\left( \max\left\{ \frac{1}{\alpha}, \frac{1}{\tau}, (\omega+1), \frac{1}{p} \right\}\right) \tag{11}$$

iterations, in the $\mu$–strongly convex and general convex regimes, respectively. These complexities depend on two hyper-parameters: $p \in (0, 1]$ and $\tau \in (0, 1]$. For simplicity, in what follows we consider the $\mu$–strongly convex case only[4]. While the iteration complexities (10) and (11) are clearly important, in the context of our paper, optimizing communication complexity (4) is more important. In the following simple theorem, we give expressions for the communication complexities of 2Direction, taking into account both workers-to-server (w2s) and server-to-workers (s2w) communication.

**Theorem 5.1.** *Assume that $\mathcal{C}^{D,\cdot}_i$ and $\mathcal{C}^P$ have expected densities equal to $K_\omega$ and $K_\alpha$, respectively (see Definition 2.5). In view of Theorem E.12, in expectation, the w2s and s2w communication complexities are equal to*

$$\mathfrak{m}^{\text{w2s}}_{\text{new}} = \widetilde{\Theta}\left( K_\omega \times T + d \right) \quad \text{and} \quad \mathfrak{m}^{\text{s2w}}_{\text{new}} = \widetilde{\Theta}\left( (K_\alpha + pd) \times T + d \right). \tag{12}$$

*Proof.* The first complexity in (12) follows because w2s communication involves $\mathcal{C}^{D,y}_i(\cdot)$ and $\mathcal{C}^{D,z}_i(\cdot)$ only. The second complexity in (12) follows because s2w communication involves $\mathcal{C}^P_i(\cdot)$, plus two non-compressed vectors $x^{t+1}$ and $k^{t+1}$ with the probability $p$. The term $d$ comes from the fact that non-compressed vectors are communicated in the initialization phase. $\square$

---

[4] One can always take $\mu = \varepsilon/R^2$ to understand the dependencies in the general convex case.

## 5.1 The ratio $L_{\max}/L$ is not known

In the following theorem, we consider the regime when the exact value of $L_{\max}/L$ is not known. Hence, we seek to find $p$ and $\tau$ that minimize the worst case $\mathfrak{m}_{\text{new}}^r$ (see (4)) w.r.t. $L_{\max} \in [L, nL]$.

**Theorem 5.2.** *Choose* $r \in [0,1]$ *and let* $\mu_{\omega,\alpha}^r := \frac{rd}{(1-r)K_\omega + rK_\alpha}$. *In view of Theorem 5.1, the values* $p = \min\left\{\frac{1}{\omega+1}, \frac{1}{\mu_{\omega,\alpha}^r}\right\}$ *and* $\tau = \frac{p^{1/3}}{(\omega+1)^{2/3}}$ *minimize* $\max\limits_{L_{\max} \in [L,nL]} \mathfrak{m}_{\text{new}}^r$. *This choice leads to the following number of communication rounds:*

$$T^{\text{realistic}} := \widetilde{\Theta}\left(\max\left\{\sqrt{\frac{L\max\{\omega+1, \mu_{\omega,\alpha}^r\}}{\alpha\mu}}, \sqrt{\frac{L_{\max}\omega\max\{\omega+1,\mu_{\omega,\alpha}^r\}}{n\mu}}, \frac{1}{\alpha}, (\omega+1), \mu_{\omega,\alpha}^r\right\}\right). \quad (13)$$

*The total communication complexity thus equals* $\mathfrak{m}_{\text{realistic}}^r = \widetilde{\Theta}\left(((1-r)K_\omega + rK_\alpha)\,T_{\text{realistic}} + d\right)$.

*Remark* 5.3. To simplify the rate (13) and understand the quantity $\mu_{\omega,\alpha}^r$, let $\mathcal{C}_i^{D,\cdot}$ be the Rand$K$ sparsifier[5] and consider the case when the s2w communication is not slower than the w2s communication, i.e., $r \leq 1/2$. Then $T^{\text{realistic}} = \widetilde{\Theta}\left(\max\left\{\sqrt{\frac{L(\omega+1)}{\alpha\mu}}, \sqrt{\frac{L_{\max}\omega(\omega+1)}{np\mu}}, \frac{1}{\alpha}, (\omega+1)\right\}\right)$ and $\mu_{\omega,\alpha}^r \leq \omega+1$. Indeed, this follows from $r \leq 1/2$ and the fact that $\omega+1 = d/K_\omega$ for the Rand$K$ compressor: $\mu_{\omega,\alpha}^r := \frac{rd}{(1-r)K_\omega + rK_\alpha} \leq \frac{r}{1-r} \times \frac{d}{K_\omega} \leq \omega+1$.

## 5.2 The ratio $L_{\max}/L$ is known

We now consider the case when we have information about the ratio $L_{\max}/L$.

**Theorem 5.4.** *Choose* $r \in [0,1]$, *and let* $\mu_{\omega,\alpha}^r := \frac{rd}{(1-r)K_\omega + rK_\alpha}$. *In view of Theorem 5.1, the values* $p$ *and* $\tau$ *given by* (63) *and* (58), *respectively, minimize* $\mathfrak{m}_{\text{new}}^r$ *from* (10). *This choice leads to the following number of communication rounds:*

$$T^{\text{optimistic}} = \widetilde{\Theta}\left(\max\left\{\sqrt{\frac{L\max\{1, \mu_{\omega,\alpha}^r\}}{\alpha\mu}}, \sqrt{\frac{L^{2/3}L_{\max}^{1/3}(\omega+1)}{\alpha n^{1/3}\mu}}, \sqrt{\frac{L^{1/2}L_{\max}^{1/2}(\omega+1)^{3/2}}{\sqrt{\alpha n}\mu}},\right.\right.$$
$$\left.\left.\sqrt{\frac{L_{\max}\omega\max\{\omega+1, \mu_{\omega,\alpha}^r\}}{n\mu}}, \frac{1}{\alpha}, (\omega+1), \mu_{\omega,\alpha}^r\right\}\right). \quad (14)$$

*The total communication complexity thus equals* $\mathfrak{m}_{\text{optimistic}}^r = \widetilde{\Theta}\left(((1-r)K_\omega + rK_\alpha)\,T_{\text{optimistic}} + d\right)$.

Note that information about $L_{\max}/L$ leads to a better rate that in Theorem 5.2.

*Remark* 5.5. To simplify the rate (14), let $\mathcal{C}_i^{D,\cdot}$ be the Rand$K$ sparsifier[6] and consider the case when the s2w communication is not slower than the w2s communication, i.e., $r \leq 1/2$. Then $T^{\text{optimistic}} = \widetilde{\Theta}\left(\max\left\{\sqrt{\frac{L\max\{1, r(\omega+1)\}}{\alpha\mu}}, \sqrt{\frac{L^{2/3}L_{\max}^{1/3}(\omega+1)}{\alpha n^{1/3}\mu}}, \sqrt{\frac{L^{1/2}L_{\max}^{1/2}(\omega+1)^{3/2}}{\sqrt{\alpha n}\mu}}, \sqrt{\frac{L_{\max}\omega(\omega+1)}{n\mu}}, \frac{1}{\alpha}, (\omega+1)\right\}\right)$. Indeed, this follows from $r \leq 1/2$ and the fact that $\omega+1 = d/K_\omega$ for the Rand$K$ compressor: $\mu_{\omega,\alpha}^r := \frac{rd}{(1-r)K_\omega + rK_\alpha} \leq \frac{r}{1-r} \times \frac{d}{K_\omega} \leq 2r(\omega+1)$.

# 6 Theoretical Comparison with Previous State of the Art

We now show that the communication complexity of 2Direction is always *no worse* than that of EF21 + DIANA and AGD. Crucially, in some regimes, it can be substantially better. Furthermore, we show that if the s2w communication cost is zero (i.e., if $r = 0$), the 2Direction obtains the same communication complexity as ADIANA (Li et al., 2020) (see Section S).

**Comparison with EF21 + DIANA.** The EF21-P + DIANA method has the communication complexities that equal

$$\mathfrak{m}_{\text{EF21-P + DIANA}}^{\text{w2s}} = \widetilde{\Theta}\left(K_\omega \times T^{\text{EF21-P + DIANA}} + d\right) \quad \text{and} \quad \mathfrak{m}_{\text{EF21-P + DIANA}}^{\text{s2w}} = \widetilde{\Theta}\left(K_\alpha \times T^{\text{EF21-P + DIANA}} + d\right),$$

---

[5]It is sufficient to assume that $\omega+1 = \Theta(d/K_\omega)$.

[6]It is sufficient to assume that $\omega+1 = \Theta(d/K_\omega)$.

where $K_\omega$ and $K_\alpha$ are the expected densities of $\mathcal{C}_i^D$ and $\mathcal{C}^P$ in Algorithm 4. The last term $d$ comes from the fact that EF21-P + DIANA sends non-compressed vectors in the initialization phase. Let us define $K_{\omega,\alpha}^r := (1-r)K_\omega + rK_\alpha$. Therefore, the total communication complexity equals

$$\mathfrak{m}_{\text{EF21-P + DIANA}}^r = \widetilde{\Theta}\left( K_{\omega,\alpha}^r \left( \frac{L}{\alpha\mu} + \frac{\omega L_{\max}}{n\mu} + \omega \right) + d \right). \tag{15}$$

Theorem 5.2 ensures that the total communication complexity of 2Direction is

$$\mathfrak{m}_{\text{realistic}}^r = \widetilde{\Theta}\left( K_{\omega,\alpha}^r \left( \sqrt{\frac{L\max\{\omega+1, \mu_{\omega,\alpha}^r\}}{\alpha\mu}} + \sqrt{\frac{L_{\max}\omega\max\{\omega+1, \mu_{\omega,\alpha}^r\}}{n\mu}} + \frac{1}{\alpha} + \omega + \mu_{\omega,\alpha}^r \right) + d \right). \tag{16}$$

One can see that (16) is an accelerated rate; it has much better dependence on the condition numbers $L/\mu$ and $L_{\max}/\mu$. In Section E.8, we prove the following simple theorem, which means that 2Direction is not worse than EF21-P + DIANA.

**Theorem 6.1.** *For all $r \in [0,1]$, $\mathfrak{m}_{\text{realistic}}^r = \widetilde{\mathcal{O}}\left(\mathfrak{m}_{\text{EF21-P + DIANA}}^r\right)$.*

**Comparison with AGD.** To compare the abstract complexity (16) with the non-abstract complexity $\widetilde{\Theta}(d\sqrt{L/\mu})$, we take the RandK and TopK compressors in Algorithm 1.

**Theorem 6.2.** *For all $r \in [0,1]$ and for all $K \in [d]$, let us take the RandK and TopK compressors with the parameters (expected densities) i) $K_\omega = K$ and $K_\alpha = \min\{\lceil 1-r/rK \rceil, d\}$ for $r \in [0, 1/2]$, ii) $K_\omega = \min\{\lceil r/1-rK \rceil, d\}$ and $K_\alpha = K$ for $r \in (1/2, 1]$. Then we have $\mathfrak{m}_{\text{realistic}}^r = \widetilde{\mathcal{O}}(\mathfrak{m}_{\text{AGD}})$.*

Theorem 6.2 states that the total communication complexity of 2Direction is not worse than that of AGD. It can be strictly better in the regimes when $\alpha > K/d$ (for TopK), $L_{\max} < nL$, and $n > 1$.

## 7 Proof Sketch

After we settle on the final version of Algorithm 1, the proof technique is pretty standard at the beginning (Li et al., 2020; Tyurin and Richtárik, 2023; Gruntkowska et al., 2023). We proved a descent lemma (Lemma E.4) and lemmas that control the convergences of auxiliary sequences (Lemmas E.5, E.6, E.7, E.8). Using these lemmas, we construct the Lyapunov function (45) with the coefficients $\kappa, \rho, \lambda$ and $\nu_t \geq 0$ for all $t \geq 0$.

One of the main problems was to find appropriate $\bar{L}, \kappa, \rho, \lambda$ and $\nu_t \geq 0$ such that we get a convergence. In more details, to get a converge it is sufficient to find $\bar{L}, \kappa, \rho, \lambda$ and $\nu_t \geq 0$ such that (46), (47) and (48) hold. Using the symbolic computation SymPy library (Meurer et al., 2017), we found appropriate $\kappa, \rho, \lambda$ and $\nu_t \geq 0$ ((90), (88), (82), (83)) such that the inequalities hold. But that is not all. To get a convergence, we also found the bounds on the parameter $\bar{L}$, which essentially describe the speed of the convergence. In raw form, using symbolic computations, we obtained a huge number of bounds on $\bar{L}$ (see Sections I, J, L, N). It was clear that most of them are redundant and it was required to find the essential ones. After a close look at the bounds on $\bar{L}$, we found out that it is sufficient to require (44) to ensure that all inequalities from Sections I, J, L and N hold (see Section O for details).

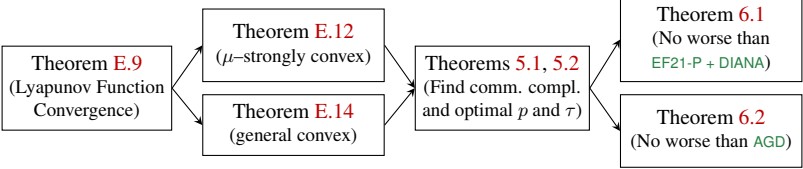

Figure 1: Roadmap to our resolution of Main Problem.

## 8 Limitations and Future Work

In contrast to Algorithm 4 (EF21+DIANA), in which the server always broadcasts compressed vectors, in our Algorithm 1 (2Direction) the server needs to broadcast non-compressed vectors with

probability $p$ (see Line 20). While in Section 6 we explain that this does not have an adverse effect on the theoretical communication complexity since $p$ is small, one may wonder whether it might be possible to achieve the same (or better) bounds as ours without having to resort to intermittent non-compressed broadcasts. This remains an open problem; possibly a challenging one. Another limitation comes from the fact that 2Direction requires more iterations than AGD in general (this is the case of all methods that reduce communication complexity). While, indeed, (10) can be higher than $\widetilde{\Theta}(\sqrt{L/\mu})$, the total communication complexity of 2Direction is not worse than that of AGD.

### Acknowledgements

This work of P. Richtárik and A. Tyurin was supported by the KAUST Baseline Research Scheme (KAUST BRF) and the KAUST Extreme Computing Research Center (KAUST ECRC), and the work of P. Richtárik was supported by the SDAIA-KAUST Center of Excellence in Data Science and Artificial Intelligence (SDAIA-KAUST AI).

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

# Appendix

## Contents

# A    Table of Notations

| Notation | Meaning |
|---|---|
| $g = \mathcal{O}(f)$ | Exist $C > 0$ such that $g(z) \leq C \times f(z)$ for all $z \in \mathcal{Z}$ |
| $g = \Omega(f)$ | Exist $C > 0$ such that $g(z) \geq C \times f(z)$ for all $z \in \mathcal{Z}$ |
| $g = \Theta(f)$ | $g = \mathcal{O}(f)$ and $g = \Omega(f)$ |
| $g = \widetilde{\mathcal{O}}(f)$ | Exist $C > 0$ such that $g(z) \leq C \times f(z) \times \log(\mathrm{poly}(z))$ for all $z \in \mathcal{Z}$ |
| $g = \widetilde{\Omega}(f)$ | Exist $C > 0$ such that $g(z) \geq C \times f(z) \times \log(\mathrm{poly}(z))$ for all $z \in \mathcal{Z}$ |
| $g = \widetilde{\Theta}(f)$ | $g = \widetilde{\mathcal{O}}(f)$ and $g = \widetilde{\Omega}(f)$ |
| $\{a, \ldots, b\}$ | Set $\{i \in \mathbb{Z} \mid a \leq i \leq b\}$ |
| $[n]$ | $\{1, \ldots, n\}$ |

# B    The Original ADIANA Algorithm

In this section, we present the ADIANA algorithm from (Li et al., 2020). In the following method, the notations, parameterization, and order of steps can be slightly different, but the general idea is the same.

---

**Algorithm 3** Accelerated DIANA (ADIANA) by Li et al. (2020)

---

1: **Parameters:** Lipschitz-like parameter $\bar{L} > 0$, strong-convexity parameter $\mu \geq 0$, probability $p$, parameter $\Gamma_0$, initial point $x^0 \in \mathbb{R}^d$, initial gradient shifts $h_1^0, \ldots, h_n^0 \in \mathbb{R}^d$
2: Initialize $\beta = 1/\omega+1$ and $z^0 = u^0 = x^0$
3: Initialize $h^0 = \frac{1}{n} \sum_{i=1}^n h_i^0$
4: **for** $t = 0, 1, \ldots, T-1$ **do**
5:     $\Gamma_{t+1}, \gamma_{t+1}, \theta_{t+1} = \text{CalculateOriginalLearningRates}(\ldots)$
6:     **for** $i = 1, \ldots, n$ in parallel **do**
7:         $y^{t+1} = \theta_{t+1} u^t + (1 - \theta_{t+1}) z^t$
8:         $m_i^{t,y} = \mathcal{C}_i^{D,y}(\nabla f_i(y^{t+1}) - h_i^t)$     Worker $i$ compresses the shifted gradient via the compressor $\mathcal{C}_i^{D,y} \in \mathbb{U}(\omega)$
9:         Send compressed message $m_i^{t,y}$ to the server
10:    **end for**
11:    $m^{t,y} = \frac{1}{n} \sum_{i=1}^n m_i^{t,y}$     Server averages the messages
12:    $g^{t+1} = h^t + m^{t,y} \equiv \frac{1}{n} \sum_{i=1}^n \left( h_i^t + m_i^{t,y} \right)$
13:    $u^{t+1} = \arg\min_{x \in \mathbb{R}^d} \left\langle g^{t+1}, x \right\rangle + \frac{\bar{L} + \Gamma_t \mu}{2\gamma_{t+1}} \left\| x - u^t \right\|^2 + \frac{\mu}{2} \left\| x - y^{t+1} \right\|^2$     Server does a gradient-like descent
     step
14:    Flip a coin $c^t \sim \text{Bernoulli}(p)$
15:    Broadcast non-compressed messages $u^{t+1}$ to all $n$ workers
16:    **for** $i = 1, \ldots, n$ in parallel **do**
17:        $x^{t+1} = \theta_{t+1} u^{t+1} + (1 - \theta_{t+1}) z^t$
18:        $z^{t+1} = \begin{cases} x^{t+1}, & c^t = 1 \\ z^t, & c^t = 0 \end{cases}$
19:        $m_i^{t,z} = \mathcal{C}_i^{D,z}(\nabla f_i(z^{t+1}) - h_i^t)$     Worker $i$ compresses the shifted gradient via the compressor $\mathcal{C}_i^{D,z} \in \mathbb{U}(\omega)$
20:        $h_i^{t+1} = h_i^t + \beta m_i^{t,z}$     Update the control variables
21:        Send compressed message $m_i^{t,z}$ to the server
22:    **end for**
23:    $h^{t+1} = h^t + \beta \frac{1}{n} \sum_{i=1}^n m_i^{t,z}$     Server averages the messages
24: **end for**

---

## C  The Original EF21-P + DIANA Algorithm

In this section, we present the EF21-P + DIANA algorithm from (Gruntkowska et al., 2023). In the following method, the notations, parameterization, and order of steps can be slightly different, but the general idea is the same.

---

**Algorithm 4** EF21-P + DIANA by Gruntkowska et al. (2023)

1: **Parameters:** learning rates $\gamma > 0$ and $\beta > 0$, initial model $u^0 \in \mathbb{R}^d$, initial gradient shifts $h_1^0, \dots, h_n^0 \in \mathbb{R}^d$, average of the initial gradient shifts $h^0 = \frac{1}{n} \sum_{i=1}^n h_i^0$, initial model shift $w^0 = u^0 \in \mathbb{R}^d$
2: **for** $t = 0, 1, \dots, T - 1$ **do**
3:     **for** $i = 1, \dots, n$ in parallel **do**
4:         $m_i^t = \mathcal{C}_i^D(\nabla f_i(w^t) - h_i^t)$          Worker $i$ compresses the shifted gradient via the dual compressor $\mathcal{C}_i^D \in \mathbb{U}(\omega)$
5:         Send compressed message $m_i^t$ to the server
6:         $h_i^{t+1} = h_i^t + \beta m_i^t$          Worker $i$ updates its local gradient shift with stepsize $\beta$
7:     **end for**
8:     $m^t = \frac{1}{n} \sum_{i=1}^n m_i^t$          Server averages the $n$ messages received from the workers
9:     $h^{t+1} = h^t + \beta m^t$          Server updates the average gradient shift so that $h^t = \frac{1}{n} \sum_{i=1}^n h_i^t$
10:    $g^t = h^t + m^t$          Server computes the gradient estimator
11:    $u^{t+1} = u^t - \gamma g^t$          Server takes a gradient-type step with stepsize $\gamma$
12:    $p^{t+1} = \mathcal{C}^P(u^{t+1} - w^t)$          Server compresses the shifted model via the primal compressor $\mathcal{C}^P \in \mathbb{B}(\alpha)$
13:    $w^{t+1} = w^t + p^{t+1}$          Server updates the model shift
14:    Broadcast compressed message $p^{t+1}$ to all $n$ workers
15:    **for** $i = 1, \dots, n$ in parallel **do**
16:        $w^{t+1} = w^t + p^{t+1}$          Worker $i$ updates its local copy of the model shift
17:    **end for**
18: **end for**

---

## D  Useful Identities and Inequalities

For all $x, y, x_1, \dots, x_n \in \mathbb{R}^d$, $s > 0$ and $\alpha \in (0, 1]$, we have:

$$\|x + y\|^2 \leq (1 + s) \|x\|^2 + (1 + s^{-1}) \|y\|^2, \tag{17}$$

$$\|x + y\|^2 \leq 2 \|x\|^2 + 2 \|y\|^2, \tag{18}$$

$$\langle x, y \rangle \leq \frac{\|x\|^2}{2s} + \frac{s \|y\|^2}{2}, \tag{19}$$

$$(1 - \alpha) \left(1 + \frac{\alpha}{2}\right) \leq 1 - \frac{\alpha}{2}, \tag{20}$$

$$(1 - \alpha) \left(1 + \frac{2}{\alpha}\right) \leq \frac{2}{\alpha}, \tag{21}$$

$$\langle a, b \rangle = \frac{1}{2} \left( \|a\|^2 + \|b\|^2 - \|a - b\|^2 \right). \tag{22}$$

**Variance decomposition:** For any random vector $X \in \mathbb{R}^d$ and any non-random $c \in \mathbb{R}^d$, we have

$$\mathbb{E}\left[ \|X - c\|^2 \right] = \mathbb{E}\left[ \|X - \mathbb{E}[X]\|^2 \right] + \|\mathbb{E}[X] - c\|^2. \tag{23}$$

**Lemma D.1** (Nesterov (2018)). *Let $f : \mathbb{R}^d \to \mathbb{R}$ be a function for which Assumptions 1.2 and 1.3 are satisfied. Then for all $x, y \in \mathbb{R}^d$ we have:*

$$\|\nabla f(x) - \nabla f(y)\|^2 \leq 2L(f(x) - f(y) - \langle \nabla f(y), x - y \rangle). \tag{24}$$

## E  Proofs of Theorems

### E.1  Analysis of learning rates

In this section, we establish inequalities for the sequences from Algorithm 2.

**Lemma E.1.** *Suppose that parameter $\bar{L} > 0$, strong-convexity parameter $\mu \geq 0$, probability $p \in (0, 1]$, $\bar{L} \geq \mu$, and $\Gamma_0 \geq 1$. Then the sequences generated by Algorithm 2 have the following properties:*

1. *The quantities $\theta_{t+1}, \gamma_{t+1}$, and $\Gamma_{t+1}$ are well-defined and $\theta_{t+1}, \gamma_{t+1} \geq 0$ for all $t \geq 0$.*

2. *$\gamma_{t+1} = p\theta_{t+1}\Gamma_{t+1}$ for all $t \geq 0$.*

3. *$\bar{L}\theta_{t+1}\gamma_{t+1} \leq (\bar{L} + \Gamma_t\mu)$ for all $t \geq 0$.*

4.
$$\Gamma_t \geq \frac{\Gamma_0}{2} \exp\left(t \min\left\{\sqrt{\frac{p\mu}{4\bar{L}}}, p\theta_{\min}\right\}\right)$$

   *for all $t \geq 0$.*

5.
$$\Gamma_t \geq \begin{cases} \frac{\Gamma_0}{2} \exp\left(tp\theta_{\min}\right), & t < \bar{t} \\ \frac{1}{4p\theta_{\min}^2} + \frac{p(t-\bar{t})^2}{16}, & t \geq \bar{t}. \end{cases}$$

   *where $\bar{t} := \max\left\{\left\lceil \frac{1}{p\theta_{\min}} \log \frac{1}{2\Gamma_0 p\theta_{\min}^2} \right\rceil, 0\right\}$.*

6. *$\{\theta_{t+1}\}_{t=0}^{\infty}$ is a non-increasing sequence.*

*Proof.*

1. Note that $\bar{\theta}_{t+1}$ is the largest root of
$$p\bar{L}\Gamma_t\bar{\theta}_{t+1}^2 + p(\bar{L} + \Gamma_t\mu)\bar{\theta}_{t+1} - (\bar{L} + \Gamma_t\mu) = 0. \tag{25}$$

   We fix $t \geq 0$. Assume that $\Gamma_t > 0$.

   Then
$$p\bar{L}\Gamma_t \times 0 + p(\bar{L} + \Gamma_t\mu) \times 0 - (\bar{L} + \Gamma_t\mu) < 0.$$

   Therefore, the largest root $\bar{\theta}_{t+1}$ is well-defined and $\bar{\theta}_{t+1} \geq 0$, and $\theta_{t+1} = \min\{\bar{\theta}_{t+1}, \theta_{\min}\} \geq 0$. Next, $\gamma_{t+1}$ is well-defined and
$$\gamma_{t+1} = p\theta_{t+1}\Gamma_t/(1 - p\theta_{t+1}) \geq 0$$

   since $p\theta_{t+1} \in [0, 1/4]$. Finally, $\Gamma_{t+1} = \Gamma_t + \gamma_{t+1} > 0$. We showed that, for all $t \geq 0$, if $\Gamma_t > 0$, then $\theta_{t+1}, \gamma_{t+1} \geq 0$ and $\Gamma_{t+1} > 0$. Note that $\Gamma_0 > 0$, thus $\theta_{t+1}, \gamma_{t+1} \geq 0$ and $\Gamma_{t+1} > 0$ for all $t \geq 0$.

2. From the definition of $\gamma_{t+1}$ and $\Gamma_{t+1}$, we have
$$(1 - p\theta_{t+1})\gamma_{t+1} = p\theta_{t+1}\Gamma_t,$$

   which is equivalent to
$$\gamma_{t+1} = p\theta_{t+1}\left(\Gamma_t + \gamma_{t+1}\right) = p\theta_{t+1}\Gamma_{t+1}.$$

3. Recall again that $\bar{\theta}_{t+1} \geq 0$ is the largest root of
$$p\bar{L}\Gamma_t\bar{\theta}_{t+1}^2 + p(\bar{L} + \Gamma_t\mu)\bar{\theta}_{t+1} - (\bar{L} + \Gamma_t\mu) = 0.$$

   If $\bar{\theta}_{t+1} \leq \theta_{\min}$, then
$$p\bar{L}\Gamma_t\theta_{t+1}^2 + p(\bar{L} + \Gamma_t\mu)\theta_{t+1} - (\bar{L} + \Gamma_t\mu) = 0.$$

   Otherwise, if $\bar{\theta}_{t+1} > \theta_{\min}$, since
$$p\bar{L}\Gamma_t \times 0 + p(\bar{L} + \Gamma_t\mu) \times 0 - (\bar{L} + \Gamma_t\mu) < 0,$$

and $\theta_{t+1} = \theta_{\min} < \bar{\theta}_{t+1}$, then

$$p\bar{L}\Gamma_t\theta_{t+1}^2 + p(\bar{L} + \Gamma_t\mu)\theta_{t+1} - (\bar{L} + \Gamma_t\mu) \leq 0. \tag{26}$$

In all cases, the inequality (26) holds. From this inequality, we can get

$$p\bar{L}\Gamma_t\theta_{t+1}^2 \leq (\bar{L} + \Gamma_t\mu)(1 - p\theta_{t+1})$$

and

$$\bar{L}\theta_{t+1}\frac{p\theta_{t+1}\Gamma_t}{(1 - p\theta_{t+1})} \leq (\bar{L} + \Gamma_t\mu).$$

Using the definition of $\gamma_{t+1}$, we obtain

$$\bar{L}\theta_{t+1}\gamma_{t+1} \leq (\bar{L} + \Gamma_t\mu)$$

for all $t \geq 0$.

4. Let us find the largest root of the quadratic equation (25):

$$\bar{\theta}_{t+1} := \frac{-p(\bar{L} + \Gamma_t\mu) + \sqrt{p^2(\bar{L} + \Gamma_t\mu)^2 + 4p\bar{L}\Gamma_t(\bar{L} + \Gamma_t\mu)}}{2p\bar{L}\Gamma_t}. \tag{27}$$

Let us define $a := p^2\left(\bar{L} + \Gamma_t\mu\right)^2$ and $b := 4\bar{L}p\left(\bar{L} + \Gamma_t\mu\right)\Gamma_t$, then

$$\bar{\theta}_{t+1} = \frac{-\sqrt{a} + \sqrt{a + b}}{2\bar{L}p\Gamma_t}$$

Since $\Gamma_t \geq 1$ for all $t \geq 0$, and $\bar{L} \geq \mu$, we have

$$a = p^2\left(\bar{L} + \Gamma_t\mu\right)^2 \leq p\left(\bar{L} + \Gamma_t\mu\right)^2 \leq p\left(\bar{L} + \Gamma_t\mu\right)\left(\Gamma_t\bar{L} + \Gamma_t\mu\right) \leq 2\bar{L}p\left(\bar{L} + \Gamma_t\mu\right)\Gamma_t = \frac{b}{2}.$$

Using $\sqrt{x + y} \geq \left(\sqrt{x} + \sqrt{y}\right)/\sqrt{2}$ for all $x, y \geq 0$, and $\sqrt{b} \geq \sqrt{2}\sqrt{a}$ we have

$$\begin{aligned}
\bar{\theta}_{t+1} &= \frac{-\sqrt{a} + \sqrt{a + b}}{2\bar{L}p\Gamma_t} \geq \frac{-\sqrt{a} + \frac{1}{\sqrt{2}}\sqrt{a} + \frac{1}{\sqrt{2}}\sqrt{b}}{2\bar{L}p\Gamma_t} \\
&= \frac{\left(\frac{1}{\sqrt{2}} - 1\right)\sqrt{a} + \frac{1}{\sqrt{2}}\left(\frac{1}{\sqrt{2}} + 1 - \frac{1}{\sqrt{2}}\right)\sqrt{b}}{2\bar{L}p\Gamma_t} \geq \frac{\frac{1}{\sqrt{2}}\left(\frac{1}{\sqrt{2}}\right)\sqrt{b}}{2\bar{L}p\Gamma_t} = \frac{\sqrt{b}}{4\bar{L}p\Gamma_t}.
\end{aligned}$$

Therefore

$$\bar{\theta}_{t+1} \geq \frac{\sqrt{4\bar{L}p\left(\bar{L} + \Gamma_t\mu\right)\Gamma_t}}{4\bar{L}p\Gamma_t} = \sqrt{\frac{\left(\bar{L} + \Gamma_t\mu\right)}{4\bar{L}p\Gamma_t}} \geq \max\left\{\sqrt{\frac{1}{4p\Gamma_t}}, \sqrt{\frac{\mu}{4p\bar{L}}}\right\}$$

and

$$\theta_{t+1} \geq \min\left\{\max\left\{\sqrt{\frac{1}{4p\Gamma_t}}, \sqrt{\frac{\mu}{4p\bar{L}}}\right\}, \theta_{\min}\right\}. \tag{28}$$

Next, since $p\theta_{t+1} \in [0, 1/4]$, we have

$$\gamma_{t+1} := p\theta_{t+1}\Gamma_t/(1 - p\theta_{t+1}) \geq p\theta_{t+1}\Gamma_t(1 + p\theta_{t+1}) \tag{29}$$

and

$$\Gamma_{t+1} := \Gamma_t + \gamma_{t+1} \geq \left(1 + p\theta_{t+1} + p^2\theta_{t+1}^2\right)\Gamma_t.$$

Using (28) and (29), we obtain

$$\begin{aligned}
\Gamma_{t+1} &\geq \left(1 + p\min\left\{\sqrt{\frac{\mu}{4p\bar{L}}}, \theta_{\min}\right\}\right)\Gamma_t = \left(1 + \min\left\{\sqrt{\frac{p\mu}{4\bar{L}}}, p\theta_{\min}\right\}\right)\Gamma_t \\
&\geq \Gamma_0\left(1 + \min\left\{\sqrt{\frac{p\mu}{4\bar{L}}}, p\theta_{\min}\right\}\right)^{t+1} \geq \frac{\Gamma_0}{2}\exp\left((t + 1)\min\left\{\sqrt{\frac{p\mu}{4\bar{L}}}, p\theta_{\min}\right\}\right),
\end{aligned} \tag{30}$$

where we use that $1 + x \geq e^x/2$ for all $x \in [0, 1]$.

5. Using (28) and (29), we have

$$\Gamma_{t+1} \geq \left(1 + p\theta_{t+1} + p^2\theta_{t+1}^2\right)\Gamma_t$$

$$\geq \Gamma_t + p\min\left\{\sqrt{\frac{1}{4p\Gamma_t}}, \theta_{\min}\right\}\Gamma_t + p^2\min\left\{\sqrt{\frac{1}{4p\Gamma_t}}, \theta_{\min}\right\}^2\Gamma_t.$$

The sequence $\Gamma_{t+1}$ is strongly increasing. Thus, exists the minimal $\widehat{t} \geq 0$ such that $\Gamma_{\widehat{t}} \geq (4p\theta_{\min}^2)^{-1}$. For all $0 \leq t < \widehat{t}$, it holds that $\Gamma_t < (4p\theta_{\min}^2)^{-1}$, $\sqrt{\frac{1}{4p\Gamma_t}} > \theta_{\min}$, and

$$\Gamma_{t+1} \geq \Gamma_t + p\theta_{\min}\Gamma_t + p^2\theta_{\min}^2\Gamma_t \geq \Gamma_t + p\theta_{\min}\Gamma_t \geq \Gamma_0 \left(1 + p\theta_{\min}\right)^{t+1} \geq \frac{\Gamma_0}{2}\exp\left((t+1)p\theta_{\min}\right). \tag{31}$$

Therefore, if $\widehat{t} > 0$, then

$$\frac{1}{4p\theta_{\min}^2} > \Gamma_{\widehat{t}-1} \geq \frac{\Gamma_0}{2}\exp\left((\widehat{t}-1)p\theta_{\min}\right).$$

Thus, we have the following bound for $\widehat{t}$:

$$\widehat{t} \leq \bar{t} := \max\left\{\left\lceil\frac{1}{p\theta_{\min}}\log\frac{1}{2\Gamma_0 p\theta_{\min}^2}\right\rceil, 0\right\}.$$

For all $t \geq \widehat{t}$, we have $\sqrt{\frac{1}{4p\Gamma_t}} \leq \theta_{\min}$ and

$$\Gamma_{t+1} \geq \Gamma_t + p\sqrt{\frac{1}{4p\Gamma_t}}\Gamma_t + p^2\frac{1}{4p\Gamma_t}\Gamma_t = \Gamma_t + \sqrt{\frac{p}{4}}\sqrt{\Gamma_t} + \frac{p}{4}.$$

Using mathematical induction, let us show that

$$\Gamma_t \geq \frac{1}{4p\theta_{\min}^2} + \frac{p(t-\widehat{t})^2}{16} \tag{32}$$

for all $t \geq \widehat{t}$. For $t = \widehat{t}$ it is true since $\Gamma_{\widehat{t}} \geq \left(4p\theta_{\min}^2\right)^{-1}$ by the definition of $\widehat{t}$. Next, for some $t \geq \widehat{t}$, assume that (32) holds, then

$$\Gamma_{t+1} \geq \Gamma_t + \frac{\sqrt{p}}{2}\sqrt{\Gamma_t} + \frac{p}{4}$$

$$\geq \frac{1}{4p\theta_{\min}^2} + \frac{p(t-\widehat{t})^2}{16} + \frac{\sqrt{p}}{2}\sqrt{\frac{1}{4p\theta_{\min}^2} + \frac{p(t-\widehat{t})^2}{16}} + \frac{p}{4}$$

$$\geq \frac{1}{4p\theta_{\min}^2} + \frac{p(t-\widehat{t})^2}{16} + \frac{p(t-\widehat{t})}{8} + \frac{p}{4}$$

$$\geq \frac{1}{4p\theta_{\min}^2} + \frac{p(t-\widehat{t}+1)^2}{16}.$$

We proved the inequality using mathematical induction. Combining (31) and (32), we obtain the unified inequality for $\Gamma_t$:

$$\Gamma_t \geq \min\left\{\frac{\Gamma_0}{2}\exp\left(tp\theta_{\min}\right), \frac{1}{4p\theta_{\min}^2} + \frac{p(t-\widehat{t})^2}{16}\right\}$$

for all $t \geq 0$. Also, if $t < \bar{t}$, then the first term in the minimum is less or equal than the second one. Therefore,

$$\Gamma_t \geq \begin{cases} \frac{\Gamma_0}{2}\exp\left(tp\theta_{\min}\right), & t < \bar{t} \\ \min\left\{\frac{\Gamma_0}{2}\exp\left(tp\theta_{\min}\right), \frac{1}{4p\theta_{\min}^2} + \frac{p(t-\widehat{t})^2}{16}\right\}, & t \geq \bar{t}. \end{cases} \tag{33}$$

Let us bound the second term. Recall that $\bar{t} \geq \hat{t}$, thus, if $t \geq \bar{t}$, then $(t - \hat{t})^2 \geq (t - \bar{t})^2$. In (33), we can change $\hat{t}$ to $\bar{t}$ and get

$$
\Gamma_t \geq \begin{cases} \frac{\Gamma_0}{2} \exp\left(tp\theta_{\min}\right), & t < \bar{t} \\ \min\left\{ \frac{\Gamma_0}{2} \exp\left(tp\theta_{\min}\right), \frac{1}{4p\theta_{\min}^2} + \frac{p(t-\bar{t})^2}{16} \right\}, & t \geq \bar{t}. \end{cases}
$$

Using the Taylor expansion of the exponent at the point $\bar{t}$, we get

$$
\frac{\Gamma_0}{2} \exp\left(tp\theta_{\min}\right) \geq \frac{\Gamma_0}{2} \exp\left(\bar{t}p\theta_{\min}\right) + \frac{\Gamma_0}{2} p^2 \theta_{\min}^2 \exp\left(\bar{t}p\theta_{\min}\right) \frac{(t-\bar{t})^2}{2}
$$

$$
\geq \frac{1}{4p\theta_{\min}^2} + \frac{p}{8}(t-\bar{t})^2 \geq \frac{1}{4p\theta_{\min}^2} + \frac{p}{16}(t-\bar{t})^2.
$$

for all $t \geq \bar{t}$. Finally, we can conclude that

$$
\Gamma_t \geq \begin{cases} \frac{\Gamma_0}{2} \exp\left(tp\theta_{\min}\right), & t < \bar{t} \\ \frac{1}{4p\theta_{\min}^2} + \frac{p(t-\bar{t})^2}{16}, & t \geq \bar{t}. \end{cases}
$$

6. Let us rewrite (27):

$$
\bar{\theta}_{t+1} = \frac{-p(\bar{L} + \Gamma_t \mu) + \sqrt{p^2(\bar{L} + \Gamma_t \mu)^2 + 4p\bar{L}\Gamma_t(\bar{L} + \Gamma_t \mu)}}{2p\bar{L}\Gamma_t}
$$

$$
= -\frac{1}{2}\left(\frac{1}{\Gamma_t} + \frac{\mu}{\bar{L}}\right) + \sqrt{\frac{1}{4}\left(\frac{1}{\Gamma_t} + \frac{\mu}{\bar{L}}\right)^2 + \frac{1}{p}\left(\frac{1}{\Gamma_t} + \frac{\mu}{\bar{L}}\right)}.
$$

Let us temporarily denote $a_t := \left(\frac{1}{\Gamma_t} + \frac{\mu}{\bar{L}}\right)$ for all $t \geq 0$. Note that $a_t$ is a non-increasing sequence, since $\Gamma_{t+1} \geq \Gamma_t$ for all $t \geq 0$. Therefore,

$$
\bar{\theta}_{t+1} = -\frac{1}{2}a_t + \sqrt{\frac{1}{4}a_t^2 + \frac{1}{p}a_t}.
$$

Let us take the derivative of the last term w.r.t. $a_t$ and compare it to zero:

$$
-\frac{1}{2} + \frac{\frac{1}{2}a_t + \frac{1}{p}}{2\sqrt{\frac{1}{4}a_t^2 + \frac{1}{p}a_t}} \geq 0 \Leftrightarrow \frac{1}{2}a_t + \frac{1}{p} \geq \sqrt{\frac{1}{4}a_t^2 + \frac{1}{p}a_t}
$$

$$
\Leftrightarrow \frac{1}{4}a_t^2 + \frac{1}{p}a_t + \frac{1}{p^2} \geq \frac{1}{4}a_t^2 + \frac{1}{p}a_t \Leftrightarrow \frac{1}{p^2} \geq 0.
$$

Thus, $\bar{\theta}_{t+1}$ is a non-decreasing sequence w.r.t. $a_t$. But the sequence $a_t$ is non-increasing, therefore $\bar{\theta}_{t+1}$ is a non-increasing sequence w.r.t. $t$. It is left to use that $\theta_{t+1}$ is the minimum of $\bar{\theta}_{t+1}$ and the constant quantity.

$\square$

## E.2 Generic lemmas

First, we prove a well-known lemma from the theory of accelerated methods (Lan, 2020; Stonyakin et al., 2021).

**Lemma E.2.** *Let us take vectors $a, b, g \in \mathbb{R}^d$, numerical quantities $\alpha, \beta \geq 0$, and*

$$
u = \underset{x \in \mathbb{R}^d}{\arg\min} \langle g, x \rangle + \frac{\alpha}{2}\|x - a\|^2 + \frac{\beta}{2}\|x - b\|^2.
$$

*Then*

$$
\langle g, x \rangle + \frac{\alpha}{2}\|x - a\|^2 + \frac{\beta}{2}\|x - b\|^2 \geq \langle g, u \rangle + \frac{\alpha}{2}\|u - a\|^2 + \frac{\beta}{2}\|u - b\|^2 + \frac{\alpha + \beta}{2}\|x - u\|^2
$$

(34)

*for all $x \in \mathbb{R}^d$.*

*Proof.* A function

$$\widehat{f}(x) := \langle g, x \rangle + \frac{\alpha}{2} \|x - a\|^2 + \frac{\beta}{2} \|x - b\|^2$$

is strongly-convex with the parameter $\alpha + \beta$. From the strong-convexity and the optimality of $u$, we obtain

$$\widehat{f}(x) \geq \widehat{f}(u) + \left\langle \widehat{f}(u), x - u \right\rangle + \frac{\alpha + \beta}{2} \|x - u\|^2 = \widehat{f}(u) + \frac{\alpha + \beta}{2} \|x - u\|^2$$

for all $x \in \mathbb{R}^d$. This inequality is equivalent to (34). $\square$

We assume that the conditional expectation $\mathbb{E}_t[\cdot]$ is conditioned on the randomness from the first $t$ iterations. Also, let us define $D_f(x, y) := f(x) - f(y) - \langle \nabla f(y), x - y \rangle$.

**Lemma E.3.** *Suppose that Assumptions 1.2, 1.1, 1.3 and 2.4 hold. For Algorithm 1, the following inequality holds:*

$$\mathbb{E}_t \left[ \left\| g^{t+1} - \nabla f(y^{t+1}) \right\|^2 \right]$$
$$\leq \frac{2\omega}{n^2} \sum_{i=1}^{n} \left\| \nabla f_i(z^t) - h_i^t \right\|^2 + \frac{4\omega L_{\max}}{n} (f(z^t) - f(y^{t+1}) - \langle \nabla f(y^{t+1}), z^t - y^{t+1} \rangle). \tag{35}$$

*Proof.* Using the definition of $g^{t+1}$, we have

$$\mathbb{E}_t \left[ \left\| g^{t+1} - \nabla f(y^{t+1}) \right\|^2 \right]$$
$$= \mathbb{E}_t \left[ \left\| h^t + \frac{1}{n} \sum_{i=1}^{n} \mathcal{C}_i^{D,y}(\nabla f_i(y^{t+1}) - h_i^t) - \nabla f(y^{t+1}) \right\|^2 \right]$$
$$= \frac{1}{n^2} \sum_{i=1}^{n} \mathbb{E}_t \left[ \left\| \mathcal{C}_i^{D,y}(\nabla f_i(y^{t+1}) - h_i^t) - \left( \nabla f(y^{t+1}) - h_i^t \right) \right\|^2 \right],$$

where we use the independence and the unbiasedness of the compressors (see Assumption 2.4). Using Definition 2.3 and Assumption 1.1, we have

$$\mathbb{E}_t \left[ \left\| g^{t+1} - \nabla f(y^{t+1}) \right\|^2 \right]$$
$$\leq \frac{\omega}{n^2} \sum_{i=1}^{n} \left\| \nabla f_i(y^{t+1}) - h_i^t \right\|^2$$
$$\overset{(18)}{\leq} \frac{2\omega}{n^2} \sum_{i=1}^{n} \left\| \nabla f_i(z^t) - h_i^t \right\|^2 + \frac{2\omega}{n^2} \sum_{i=1}^{n} \left\| \nabla f_i(y^{t+1}) - \nabla f_i(z^t) \right\|^2$$
$$\overset{L.D.1}{\leq} \frac{2\omega}{n^2} \sum_{i=1}^{n} \left\| \nabla f_i(z^t) - h_i^t \right\|^2 + \frac{2\omega}{n^2} \sum_{i=1}^{n} 2L_i(f_i(z^t) - f_i(y^{t+1}) - \langle \nabla f_i(y^{t+1}), z^t - y^{t+1} \rangle).$$

Using that $L_{\max} = \max_{i \in [n]} L_i$, we obtain (35). $\square$

**Lemma E.4.** *Suppose that Assumptions 1.2, 1.1, 1.3 and 2.4 hold. For Algorithm 1, the following inequality holds:*

$$\mathbb{E}_t \left[ f(z^{t+1}) - f(x^*) \right]$$
$$\leq (1 - p\theta_{t+1}) \left( f(z^t) - f(x^*) \right)$$
$$+ \frac{2p\omega}{n\bar{L}} \left( \frac{1}{n} \sum_{i=1}^{n} \left\| h_i^t - \nabla f_i(z^t) \right\|^2 \right)$$
$$+ p\theta_{t+1} \left( \frac{\bar{L} + \Gamma_t \mu}{2\gamma_{t+1}} \|u^t - x^*\|^2 - \frac{\bar{L} + \Gamma_{t+1}\mu}{2\gamma_{t+1}} \mathbb{E}_t \left[ \|u^{t+1} - x^*\|^2 \right] \right)$$

$$+ p \left( \frac{4\omega L_{\max}}{n\bar{L}} + \theta_{t+1} - 1 \right) D_f(z^t, y^{t+1})$$

$$+ \frac{pL}{2} \mathbb{E}_t \left[ \left\| x^{t+1} - y^{t+1} \right\|^2 \right] - \frac{p\theta_{t+1}^2 \bar{L}}{2} \mathbb{E}_t \left[ \left\| u^{t+1} - u^t \right\|^2 \right].$$

*Proof.* Using Assumption 1.2, we have

$$f(x^{t+1}) - f(x^*) \le f(y^{t+1}) - f(x^*) + \left\langle \nabla f(y^{t+1}), x^{t+1} - y^{t+1} \right\rangle + \frac{L}{2} \left\| x^{t+1} - y^{t+1} \right\|^2.$$

Using the definition of $x^{t+1}$, we obtain

$$
\begin{aligned}
f(x^{t+1}) - f(x^*) &\le (1 - \theta_{t+1}) \left( f(y^{t+1}) - f(x^*) + \left\langle \nabla f(y^{t+1}), z^t - y^{t+1} \right\rangle \right) \\
&\quad + \theta_{t+1} \left( f(y^{t+1}) - f(x^*) + \left\langle \nabla f(y^{t+1}), u^{t+1} - y^{t+1} \right\rangle \right) \\
&\quad + \frac{L}{2} \left\| x^{t+1} - y^{t+1} \right\|^2 \\
&= (1 - \theta_{t+1}) \left( f(y^{t+1}) - f(x^*) + \left\langle \nabla f(y^{t+1}), z^t - y^{t+1} \right\rangle \right) \\
&\quad + \theta_{t+1} \left( f(y^{t+1}) - f(x^*) + \left\langle g^{t+1}, u^{t+1} - y^{t+1} \right\rangle \right) \\
&\quad + \theta_{t+1} \left( \left\langle \nabla f(y^{t+1}) - g^{t+1}, u^{t+1} - y^{t+1} \right\rangle \right) \\
&\quad + \frac{L}{2} \left\| x^{t+1} - y^{t+1} \right\|^2,
\end{aligned}
\tag{36}
$$

in the last inequality we add and subtract $g^{t+1}$. Using the definition of $u^{t+1}$ and Lemma E.2 with $x = x^*$, we have

$$
\begin{aligned}
\left\langle g^{t+1}, u^{t+1} - y^{t+1} \right\rangle &\le \left\langle g^{t+1}, x^* - y^{t+1} \right\rangle + \frac{\bar{L} + \Gamma_t \mu}{2\gamma_{t+1}} \left\| x^* - u^t \right\|^2 + \frac{\mu}{2} \left\| x^* - y^{t+1} \right\|^2 \\
&\quad - \frac{\bar{L} + \Gamma_t \mu}{2\gamma_{t+1}} \left\| u^{t+1} - u^t \right\|^2 - \frac{\mu}{2} \left\| u^t - y^{t+1} \right\|^2 - \frac{\bar{L} + \Gamma_{t+1} \mu}{2\gamma_{t+1}} \left\| x^* - u^{t+1} \right\|^2.
\end{aligned}
$$

We use the fact that $\Gamma_{t+1} = \Gamma_t + \gamma_{t+1}$. Since $\left\| u^t - y^{t+1} \right\|^2 \ge 0$, we have

$$
\begin{aligned}
\left\langle g^{t+1}, u^{t+1} - y^{t+1} \right\rangle &\le \left\langle g^{t+1}, x^* - y^{t+1} \right\rangle + \frac{\bar{L} + \Gamma_t \mu}{2\gamma_{t+1}} \left\| x^* - u^t \right\|^2 + \frac{\mu}{2} \left\| x^* - y^{t+1} \right\|^2 \\
&\quad - \frac{\bar{L} + \Gamma_t \mu}{2\gamma_{t+1}} \left\| u^{t+1} - u^t \right\|^2 - \frac{\bar{L} + \Gamma_{t+1} \mu}{2\gamma_{t+1}} \left\| x^* - u^{t+1} \right\|^2.
\end{aligned}
$$

By substituting this inequality to (36), we get

$$
\begin{aligned}
f(x^{t+1}) &- f(x^*) \\
&\le (1 - \theta_{t+1}) \left( f(y^{t+1}) - f(x^*) + \left\langle \nabla f(y^{t+1}), z^t - y^{t+1} \right\rangle \right) \\
&\quad + \theta_{t+1} \left( f(y^{t+1}) - f(x^*) + \left\langle g^{t+1}, x^* - y^{t+1} \right\rangle \right) \\
&\quad + \theta_{t+1} \left( \frac{\bar{L} + \Gamma_t \mu}{2\gamma_{t+1}} \left\| x^* - u^t \right\|^2 + \frac{\mu}{2} \left\| x^* - y^{t+1} \right\|^2 - \frac{\bar{L} + \Gamma_t \mu}{2\gamma_{t+1}} \left\| u^{t+1} - u^t \right\|^2 - \frac{\bar{L} + \Gamma_{t+1} \mu}{2\gamma_{t+1}} \left\| x^* - u^{t+1} \right\|^2 \right) \\
&\quad + \theta_{t+1} \left( \left\langle \nabla f(y^{t+1}) - g^{t+1}, u^{t+1} - y^{t+1} \right\rangle \right) \\
&\quad + \frac{L}{2} \left\| x^{t+1} - y^{t+1} \right\|^2.
\end{aligned}
$$

Using $\mu$–strong convexity, we have

$$f(x^*) \ge f(y^{t+1}) + \left\langle \nabla f(y^{t+1}), x^* - y^{t+1} \right\rangle + \frac{\mu}{2} \left\| x^* - y^{t+1} \right\|^2$$

and

$$
\begin{aligned}
f(x^{t+1}) &- f(x^*) \\
&\le (1 - \theta_{t+1}) \left( f(y^{t+1}) - f(x^*) + \left\langle \nabla f(y^{t+1}), z^t - y^{t+1} \right\rangle \right)
\end{aligned}
$$

$$
\begin{aligned}
&+ \theta_{t+1} \left( \left\langle g^{t+1} - \nabla f(y^{t+1}), x^* - y^{t+1} \right\rangle \right) \\
&+ \theta_{t+1} \left( \frac{\bar{L} + \Gamma_t \mu}{2\gamma_{t+1}} \left\| x^* - u^t \right\|^2 - \frac{\bar{L} + \Gamma_t \mu}{2\gamma_{t+1}} \left\| u^{t+1} - u^t \right\|^2 - \frac{\bar{L} + \Gamma_{t+1} \mu}{2\gamma_{t+1}} \left\| x^* - u^{t+1} \right\|^2 \right) \\
&+ \theta_{t+1} \left( \left\langle \nabla f(y^{t+1}) - g^{t+1}, u^{t+1} - y^{t+1} \right\rangle \right) \\
&+ \frac{L}{2} \left\| x^{t+1} - y^{t+1} \right\|^2 .
\end{aligned}
$$

Let us take the conditional expectation $\mathbb{E}_t \left[ \cdot \right]$ conditioned on the randomness from the first $t$ iterations:

$$
\begin{aligned}
&\mathbb{E}_t \left[ f(x^{t+1}) - f(x^*) \right] \\
&\leq (1 - \theta_{t+1}) \left( f(y^{t+1}) - f(x^*) + \left\langle \nabla f(y^{t+1}), z^t - y^{t+1} \right\rangle \right) \\
&\quad + \theta_{t+1} \left( \left\langle \mathbb{E}_t \left[ g^{t+1} - \nabla f(y^{t+1}) \right], x^* - y^{t+1} \right\rangle \right) \\
&\quad + \theta_{t+1} \left( \frac{\bar{L} + \Gamma_t \mu}{2\gamma_{t+1}} \left\| x^* - u^t \right\|^2 - \frac{\bar{L} + \Gamma_t \mu}{2\gamma_{t+1}} \mathbb{E}_t \left[ \left\| u^{t+1} - u^t \right\|^2 \right] - \frac{\bar{L} + \Gamma_{t+1} \mu}{2\gamma_{t+1}} \mathbb{E}_t \left[ \left\| x^* - u^{t+1} \right\|^2 \right] \right) \\
&\quad + \theta_{t+1} \mathbb{E}_t \left[ \left\langle \nabla f(y^{t+1}) - g^{t+1}, u^{t+1} - y^{t+1} \right\rangle \right] \\
&\quad + \frac{L}{2} \mathbb{E}_t \left[ \left\| x^{t+1} - y^{t+1} \right\|^2 \right] \\
&= (1 - \theta_{t+1}) \left( f(y^{t+1}) - f(x^*) + \left\langle \nabla f(y^{t+1}), z^t - y^{t+1} \right\rangle \right) \\
&\quad + \theta_{t+1} \left( \frac{\bar{L} + \Gamma_t \mu}{2\gamma_{t+1}} \left\| x^* - u^t \right\|^2 - \frac{\bar{L} + \Gamma_t \mu}{2\gamma_{t+1}} \mathbb{E}_t \left[ \left\| u^{t+1} - u^t \right\|^2 \right] - \frac{\bar{L} + \Gamma_{t+1} \mu}{2\gamma_{t+1}} \mathbb{E}_t \left[ \left\| x^* - u^{t+1} \right\|^2 \right] \right) \\
&\quad + \theta_{t+1} \mathbb{E}_t \left[ \left\langle \nabla f(y^{t+1}) - g^{t+1}, u^{t+1} - y^{t+1} \right\rangle \right] \\
&\quad + \frac{L}{2} \mathbb{E}_t \left[ \left\| x^{t+1} - y^{t+1} \right\|^2 \right] ,
\end{aligned}
\tag{37}
$$

where use that $\mathbb{E}_t \left[ g^{t+1} \right] = \nabla f(y^{t+1})$. We can find $u^{t+1}$ analytically and obtain that

$$
u^{t+1} = \frac{\bar{L} + \Gamma_t \mu}{\bar{L} + \Gamma_{t+1} \mu} u^t + \frac{\mu \gamma_{t+1}}{\bar{L} + \Gamma_{t+1} \mu} y^{t+1} - \frac{\gamma_{t+1}}{\bar{L} + \Gamma_{t+1} \mu} g^{t+1} .
$$

Therefore, using that $\mathbb{E}_t \left[ g^{t+1} \right] = \nabla f(y^{t+1})$ and $u^t$ and $y^{t+1}$ are conditionally nonrandom, we obtain

$$
\mathbb{E}_t \left[ \left\langle \nabla f(y^{t+1}) - g^{t+1}, u^{t+1} - y^{t+1} \right\rangle \right] = \frac{\gamma_{t+1}}{\bar{L} + \Gamma_{t+1} \mu} \mathbb{E}_t \left[ \left\| g^{t+1} - \nabla f(y^{t+1}) \right\|^2 \right] . \tag{38}
$$

Combining (35) from Lemma E.3 with (37) and (38), one can get

$$
\begin{aligned}
&\mathbb{E}_t \left[ f(x^{t+1}) - f(x^*) \right] \\
&\leq (1 - \theta_{t+1}) \left( f(y^{t+1}) - f(x^*) + \left\langle \nabla f(y^{t+1}), z^t - y^{t+1} \right\rangle \right) \\
&\quad + \theta_{t+1} \left( \frac{\bar{L} + \Gamma_t \mu}{2\gamma_{t+1}} \left\| x^* - u^t \right\|^2 - \frac{\bar{L} + \Gamma_t \mu}{2\gamma_{t+1}} \mathbb{E}_t \left[ \left\| u^{t+1} - u^t \right\|^2 \right] - \frac{\bar{L} + \Gamma_{t+1} \mu}{2\gamma_{t+1}} \mathbb{E}_t \left[ \left\| x^* - u^{t+1} \right\|^2 \right] \right) \\
&\quad + \frac{\theta_{t+1} \gamma_{t+1}}{\bar{L} + \Gamma_{t+1} \mu} \left( \frac{2\omega}{n^2} \sum_{i=1}^n \left\| \nabla f_i(z^t) - h_i^t \right\|^2 + \frac{4\omega L_{\max}}{n} \left( f(z^t) - f(y^{t+1}) - \left\langle \nabla f(y^{t+1}), z^t - y^{t+1} \right\rangle \right) \right) \\
&\quad + \frac{L}{2} \mathbb{E}_t \left[ \left\| x^{t+1} - y^{t+1} \right\|^2 \right] .
\end{aligned}
$$

Using the notation $D_f(x, y) := f(x) - f(y) - \langle \nabla f(y), x - y \rangle$, we get

$$\mathbb{E}_t \left[ f(x^{t+1}) - f(x^*) \right]$$

$$\leq (1 - \theta_{t+1}) \left( f(z^t) - f(x^*) - D_f(z^t, y^{t+1}) \right)$$

$$+ \theta_{t+1} \left( \frac{\bar{L} + \Gamma_t \mu}{2\gamma_{t+1}} \left\| x^* - u^t \right\|^2 - \frac{\bar{L} + \Gamma_t \mu}{2\gamma_{t+1}} \mathbb{E}_t \left[ \left\| u^{t+1} - u^t \right\|^2 \right] - \frac{\bar{L} + \Gamma_{t+1} \mu}{2\gamma_{t+1}} \mathbb{E}_t \left[ \left\| x^* - u^{t+1} \right\|^2 \right] \right)$$

$$+ \frac{\theta_{t+1} \gamma_{t+1}}{\bar{L} + \Gamma_{t+1} \mu} \left( \frac{2\omega}{n^2} \sum_{i=1}^{n} \left\| \nabla f_i(z^t) - h_i^t \right\|^2 + \frac{4\omega L_{\max}}{n} D_f(z^t, y^{t+1}) \right)$$

$$+ \frac{L}{2} \mathbb{E}_t \left[ \left\| x^{t+1} - y^{t+1} \right\|^2 \right]$$

$$= (1 - \theta_{t+1}) \left( f(z^t) - f(x^*) \right)$$

$$+ \frac{\theta_{t+1} \gamma_{t+1}}{\bar{L} + \Gamma_{t+1} \mu} \frac{2\omega}{n} \left( \frac{1}{n} \sum_{i=1}^{n} \left\| \nabla f_i(z^t) - h_i^t \right\|^2 \right)$$

$$+ \theta_{t+1} \left( \frac{\bar{L} + \Gamma_t \mu}{2\gamma_{t+1}} \left\| x^* - u^t \right\|^2 - \frac{\bar{L} + \Gamma_{t+1} \mu}{2\gamma_{t+1}} \mathbb{E}_t \left[ \left\| x^* - u^{t+1} \right\|^2 \right] \right)$$

$$+ \left( \frac{\theta_{t+1} \gamma_{t+1}}{\bar{L} + \Gamma_{t+1} \mu} \frac{4\omega L_{\max}}{n} + \theta_{t+1} - 1 \right) D_f(z^t, y^{t+1})$$

$$+ \frac{L}{2} \mathbb{E}_t \left[ \left\| x^{t+1} - y^{t+1} \right\|^2 \right] - \frac{\theta_{t+1} (\bar{L} + \Gamma_t \mu)}{2\gamma_{t+1}} \mathbb{E}_t \left[ \left\| u^{t+1} - u^t \right\|^2 \right].$$

In the last equality, we simply regrouped the terms. Using the definition of $z^{t+1}$, we get

$$\mathbb{E}_t \left[ f(z^{t+1}) - f(x^*) \right] = p \mathbb{E}_t \left[ f(x^{t+1}) - f(x^*) \right] + (1 - p) \left( f(z^t) - f(x^*) \right)$$

$$\leq p(1 - \theta_{t+1}) \left( f(z^t) - f(x^*) \right)$$

$$+ p \frac{\theta_{t+1} \gamma_{t+1}}{\bar{L} + \Gamma_{t+1} \mu} \frac{2\omega}{n} \left( \frac{1}{n} \sum_{i=1}^{n} \left\| \nabla f_i(z^t) - h_i^t \right\|^2 \right)$$

$$+ p \theta_{t+1} \left( \frac{\bar{L} + \Gamma_t \mu}{2\gamma_{t+1}} \left\| x^* - u^t \right\|^2 - \frac{\bar{L} + \Gamma_{t+1} \mu}{2\gamma_{t+1}} \mathbb{E}_t \left[ \left\| x^* - u^{t+1} \right\|^2 \right] \right)$$

$$+ p \left( \frac{\theta_{t+1} \gamma_{t+1}}{\bar{L} + \Gamma_{t+1} \mu} \frac{4\omega L_{\max}}{n} + \theta_{t+1} - 1 \right) D_f(z^t, y^{t+1})$$

$$+ \frac{pL}{2} \mathbb{E}_t \left[ \left\| x^{t+1} - y^{t+1} \right\|^2 \right] - \frac{p\theta_{t+1} (\bar{L} + \Gamma_t \mu)}{2\gamma_{t+1}} \mathbb{E}_t \left[ \left\| u^{t+1} - u^t \right\|^2 \right] + (1 - p) \left( f(z^t) - f(x^*) \right)$$

$$= (1 - p\theta_{t+1}) \left( f(z^t) - f(x^*) \right)$$

$$+ p \frac{\theta_{t+1} \gamma_{t+1}}{\bar{L} + \Gamma_{t+1} \mu} \frac{2\omega}{n} \left( \frac{1}{n} \sum_{i=1}^{n} \left\| \nabla f_i(z^t) - h_i^t \right\|^2 \right)$$

$$+ p \theta_{t+1} \left( \frac{\bar{L} + \Gamma_t \mu}{2\gamma_{t+1}} \left\| x^* - u^t \right\|^2 - \frac{\bar{L} + \Gamma_{t+1} \mu}{2\gamma_{t+1}} \mathbb{E}_t \left[ \left\| x^* - u^{t+1} \right\|^2 \right] \right)$$

$$+ p \left( \frac{\theta_{t+1} \gamma_{t+1}}{\bar{L} + \Gamma_{t+1} \mu} \frac{4\omega L_{\max}}{n} + \theta_{t+1} - 1 \right) D_f(z^t, y^{t+1})$$

$$+ \frac{pL}{2} \mathbb{E}_t \left[ \left\| x^{t+1} - y^{t+1} \right\|^2 \right] - \frac{p\theta_{t+1} (\bar{L} + \Gamma_t \mu)}{2\gamma_{t+1}} \mathbb{E}_t \left[ \left\| u^{t+1} - u^t \right\|^2 \right].$$

In the last equality, we grouped the terms with $f(z^t) - f(x^*)$. In Algorithm 2, we choose the learning rates so that (see Lemma E.1)

$$\bar{L} \theta_{t+1} \gamma_{t+1} \leq \bar{L} + \Gamma_t \mu.$$

Since $\Gamma_{t+1} \geq \Gamma_t$ for all $t \in \mathbb{N}_0$, thus

$$\frac{\theta_{t+1} \gamma_{t+1}}{\bar{L} + \Gamma_{t+1} \mu} \leq \frac{\theta_{t+1} \gamma_{t+1}}{\bar{L} + \Gamma_t \mu} \leq \frac{1}{\bar{L}}$$

and

$$\mathbb{E}_t \left[ f(z^{t+1}) - f(x^*) \right]$$
$$\leq (1 - p\theta_{t+1}) \left( f(z^t) - f(x^*) \right)$$
$$+ p\frac{2\omega}{n\bar{L}} \left( \frac{1}{n} \sum_{i=1}^{n} \left\| \nabla f_i(z^t) - h_i^t \right\|^2 \right)$$
$$+ p\theta_{t+1} \left( \frac{\bar{L} + \Gamma_t \mu}{2\gamma_{t+1}} \left\| x^* - u^t \right\|^2 - \frac{\bar{L} + \Gamma_{t+1} \mu}{2\gamma_{t+1}} \mathbb{E}_t \left[ \left\| x^* - u^{t+1} \right\|^2 \right] \right)$$
$$+ p \left( \frac{4\omega L_{\max}}{n\bar{L}} + \theta_{t+1} - 1 \right) D_f(z^t, y^{t+1})$$
$$+ \frac{pL}{2} \mathbb{E}_t \left[ \left\| x^{t+1} - y^{t+1} \right\|^2 \right] - \frac{p\theta_{t+1}^2 \bar{L}}{2} \mathbb{E}_t \left[ \left\| u^{t+1} - u^t \right\|^2 \right].$$

$\square$

### E.3 Construction of the Lyapunov function

In this section, we provide lemmas that will help us to construct a Lyapunov function.

**Lemma E.5.** *Suppose that Assumptions 1.2, 1.1, 1.3 and 2.4 hold. The parameter $\beta \leq 1/\omega+1$. Then, for Algorithm 1, the following inequality holds:*

$$\mathbb{E}_t \left[ \frac{1}{n} \sum_{i=1}^{n} \left\| h_i^{t+1} - \nabla f_i(z^{t+1}) \right\|^2 \right] \tag{39}$$

$$\leq 8p \left( 1 + \frac{p}{\beta} \right) L_{\max} D_f(z^t, y^{t+1}) + 4p \left( 1 + \frac{p}{\beta} \right) \widehat{L}^2 \mathbb{E}_t \left[ \left\| x^{t+1} - y^{t+1} \right\|^2 \right] + \left( 1 - \frac{\beta}{2} \right) \frac{1}{n} \sum_{i=1}^{n} \left\| h_i^t - \nabla f_i(z^t) \right\|^2. \tag{40}$$

*Proof.* Using the definition of $h_i^{t+1}$, we have

$$\mathbb{E}_t \left[ \frac{1}{n} \sum_{i=1}^{n} \left\| h_i^{t+1} - \nabla f_i(z^{t+1}) \right\|^2 \right]$$

$$= \mathbb{E}_t \left[ \frac{1}{n} \sum_{i=1}^{n} \left\| h_i^t + \beta \mathcal{C}_i^{D,z}(\nabla f_i(z^{t+1}) - h_i^t) - \nabla f_i(z^{t+1}) \right\|^2 \right]$$

$$= \mathbb{E}_t \left[ \frac{1}{n} \sum_{i=1}^{n} \left\| h_i^t - \nabla f_i(z^{t+1}) \right\|^2 \right]$$

$$+ \mathbb{E}_t \left[ \frac{2\beta}{n} \sum_{i=1}^{n} \left\langle h_i^t - \nabla f_i(z^{t+1}), \mathcal{C}_i^{D,z}(\nabla f_i(z^{t+1}) - h_i^t) \right\rangle + \frac{\beta^2}{n} \sum_{i=1}^{n} \left\| \mathcal{C}_i^{D,z}(\nabla f_i(z^{t+1}) - h_i^t) \right\|^2 \right].$$

Note that $\mathbb{E}_{\mathcal{C}} \left[ \mathcal{C}_i^{D,z}(\nabla f_i(z^{t+1}) - h_i^t) \right] = \nabla f_i(z^{t+1}) - h_i^t$ and

$$\mathbb{E}_{\mathcal{C}} \left[ \left\| \mathcal{C}_i^{D,z}(\nabla f_i(z^{t+1}) - h_i^t) \right\|^2 \right] \leq (\omega + 1) \left\| \nabla f_i(z^{t+1}) - h_i^t \right\|^2,$$

where $\mathbb{E}_{\mathcal{C}} \left[ \cdot \right]$ is a conditional expectation that is conditioned on $z^{t+1}$ and $h_i^t$. Therefore,

$$\mathbb{E}_t \left[ \frac{1}{n} \sum_{i=1}^{n} \left\| h_i^{t+1} - \nabla f_i(z^{t+1}) \right\|^2 \right]$$

$$\leq \mathbb{E}_t \left[ \frac{1}{n} \sum_{i=1}^{n} \left\| h_i^t - \nabla f_i(z^{t+1}) \right\|^2 \right]$$

$$+ \mathbb{E}_t \left[ \frac{2\beta}{n} \sum_{i=1}^{n} \langle h_i^t - \nabla f_i(z^{t+1}), \nabla f_i(z^{t+1}) - h_i^t \rangle + \frac{\beta^2(\omega+1)}{n} \sum_{i=1}^{n} \left\| \nabla f_i(z^{t+1}) - h_i^t \right\|^2 \right]$$

$$= \left(1 - 2\beta + \beta^2(\omega+1)\right) \mathbb{E}_t \left[ \frac{1}{n} \sum_{i=1}^{n} \left\| h_i^t - \nabla f_i(z^{t+1}) \right\|^2 \right].$$

Since $\beta \leq 1/\omega+1$, we have

$$\mathbb{E}_t \left[ \frac{1}{n} \sum_{i=1}^{n} \left\| h_i^{t+1} - \nabla f_i(z^{t+1}) \right\|^2 \right] \leq (1-\beta) \mathbb{E}_t \left[ \frac{1}{n} \sum_{i=1}^{n} \left\| h_i^t - \nabla f_i(z^{t+1}) \right\|^2 \right].$$

Next, we use the definition of $z^{t+1}$ and obtain

$$\mathbb{E}_t \left[ \frac{1}{n} \sum_{i=1}^{n} \left\| h_i^{t+1} - \nabla f_i(z^{t+1}) \right\|^2 \right]$$

$$\leq (1-\beta) \, p \, \mathbb{E}_t \left[ \frac{1}{n} \sum_{i=1}^{n} \left\| h_i^t - \nabla f_i(x^{t+1}) \right\|^2 \right] + (1-\beta)(1-p) \frac{1}{n} \sum_{i=1}^{n} \left\| h_i^t - \nabla f_i(z^t) \right\|^2$$

$$\overset{(17)}{\leq} \left(1 + \frac{2p}{\beta}\right)(1-\beta)\, p\, \mathbb{E}_t \left[ \frac{1}{n} \sum_{i=1}^{n} \left\| f_i(z^t) - \nabla f_i(x^{t+1}) \right\|^2 \right] + \left(1 + \frac{\beta}{2p}\right)(1-\beta)\, p \frac{1}{n} \sum_{i=1}^{n} \left\| h_i^t - \nabla f_i(z^t) \right\|^2$$

$$+ (1-\beta)(1-p) \mathbb{E}_t \left[ \frac{1}{n} \sum_{i=1}^{n} \left\| h_i^t - \nabla f_i(z^t) \right\|^2 \right]$$

$$= \left(1 + \frac{2p}{\beta}\right)(1-\beta)\, p\, \mathbb{E}_t \left[ \frac{1}{n} \sum_{i=1}^{n} \left\| f_i(z^t) - \nabla f_i(x^{t+1}) \right\|^2 \right] + (1-\beta)\left(1 + \frac{\beta}{2}\right) \frac{1}{n} \sum_{i=1}^{n} \left\| h_i^t - \nabla f_i(z^t) \right\|^2.$$

Using $1 - \beta \leq 1$, (18) and (20), we get

$$\mathbb{E}_t \left[ \frac{1}{n} \sum_{i=1}^{n} \left\| h_i^{t+1} - \nabla f_i(z^{t+1}) \right\|^2 \right]$$

$$\leq 4p \left(1 + \frac{p}{\beta}\right) \frac{1}{n} \sum_{i=1}^{n} \left\| f_i(z^t) - \nabla f_i(y^{t+1}) \right\|^2 + 4p \left(1 + \frac{p}{\beta}\right) \mathbb{E}_t \left[ \frac{1}{n} \sum_{i=1}^{n} \left\| f_i(x^{t+1}) - \nabla f_i(y^{t+1}) \right\|^2 \right]$$

$$+ \left(1 - \frac{\beta}{2}\right) \frac{1}{n} \sum_{i=1}^{n} \left\| h_i^t - \nabla f_i(z^t) \right\|^2.$$

From Assumptions 1.1 and 1.3 and Lemma D.1, we obtain

$$\mathbb{E}_t \left[ \frac{1}{n} \sum_{i=1}^{n} \left\| h_i^{t+1} - \nabla f_i(z^{t+1}) \right\|^2 \right]$$

$$\leq 8p \left(1 + \frac{p}{\beta}\right) L_{\max} \left( f(z^t) - f(y^{t+1}) - \langle \nabla f(y^{t+1}), z^t - y^{t+1} \rangle \right) + 4p \left(1 + \frac{p}{\beta}\right) \widehat{L}^2 \mathbb{E}_t \left[ \left\| x^{t+1} - y^{t+1} \right\|^2 \right]$$

$$+ \left(1 - \frac{\beta}{2}\right) \frac{1}{n} \sum_{i=1}^{n} \left\| h_i^t - \nabla f_i(z^t) \right\|^2$$

$$= 8p \left(1 + \frac{p}{\beta}\right) L_{\max} D_f(z^t, y^{t+1}) + 4p \left(1 + \frac{p}{\beta}\right) \widehat{L}^2 \mathbb{E}_t \left[ \left\| x^{t+1} - y^{t+1} \right\|^2 \right] + \left(1 - \frac{\beta}{2}\right) \frac{1}{n} \sum_{i=1}^{n} \left\| h_i^t - \nabla f_i(z^t) \right\|^2.$$

$\square$

**Lemma E.6.** *Suppose that Assumptions 1.2, 1.1, 1.3 and 2.4 hold. Then, for Algorithm 1, the following inequality holds:*

$$\mathbb{E}_t\left[\left\|w^{t+1}-u^{t+1}\right\|^2\right] \le \left(1-\frac{\alpha}{2}\right)\left\|w^t-u^t\right\|^2 + \frac{4}{\alpha}\left(\frac{\gamma_{t+1}}{\bar{L}+\Gamma_{t+1}\mu}\right)^2\left\|k^t-\nabla f(z^t)\right\|^2$$

$$+ \frac{2\omega}{n}\left(\frac{\gamma_{t+1}}{\bar{L}+\Gamma_{t+1}\mu}\right)^2\frac{1}{n}\sum_{i=1}^n\left\|\nabla f_i(z^t)-h_i^t\right\|^2 + \left(\frac{\gamma_{t+1}}{\bar{L}+\Gamma_{t+1}\mu}\right)^2\left(\frac{4\omega L_{\max}}{n}+\frac{8L}{\alpha}\right)D_f(z^t,y^{t+1}).$$

$$\tag{41}$$

*Proof.* Using the definition of $w^{t+1}$ and Definition 2.2, we get the following inequality

$$\mathbb{E}_t\left[\left\|w^{t+1}-u^{t+1}\right\|^2\right] = \mathbb{E}_t\left[\left\|\mathcal{C}^P\left(u^{t+1}-q^{t+1}\right)-(u^{t+1}-q^{t+1})\right\|^2\right] \le (1-\alpha)\mathbb{E}_t\left[\left\|u^{t+1}-q^{t+1}\right\|^2\right].$$

We can find the analytical formulas for $u^{t+1}$ and $q^{t+1}$, and obtain that

$$u^{t+1} = \frac{\bar{L}+\Gamma_t\mu}{\bar{L}+\Gamma_{t+1}\mu}u^t + \frac{\mu\gamma_{t+1}}{\bar{L}+\Gamma_{t+1}\mu}y^{t+1} - \frac{\gamma_{t+1}}{\bar{L}+\Gamma_{t+1}\mu}g^{t+1}$$

and

$$q^{t+1} = \frac{\bar{L}+\Gamma_t\mu}{\bar{L}+\Gamma_{t+1}\mu}w^t + \frac{\mu\gamma_{t+1}}{\bar{L}+\Gamma_{t+1}\mu}y^{t+1} - \frac{\gamma_{t+1}}{\bar{L}+\Gamma_{t+1}\mu}k^t.$$

Therefore,

$$\mathbb{E}_t\left[\left\|w^{t+1}-u^{t+1}\right\|^2\right]$$

$$\le (1-\alpha)\mathbb{E}_t\left[\left\|\frac{\bar{L}+\Gamma_t\mu}{\bar{L}+\Gamma_{t+1}\mu}(w^t-u^t)-\frac{\gamma_{t+1}}{\bar{L}+\Gamma_{t+1}\mu}(k^t-g^{t+1})\right\|^2\right]$$

$$\overset{(23)}{=} (1-\alpha)\left\|\frac{\bar{L}+\Gamma_t\mu}{\bar{L}+\Gamma_{t+1}\mu}(w^t-u^t)-\frac{\gamma_{t+1}}{\bar{L}+\Gamma_{t+1}\mu}(k^t-\nabla f(y^{t+1}))\right\|^2$$

$$+ (1-\alpha)\left(\frac{\gamma_{t+1}}{\bar{L}+\Gamma_{t+1}\mu}\right)^2\mathbb{E}_t\left[\left\|g^{t+1}-\nabla f(y^{t+1})\right\|^2\right]$$

$$\overset{(17)}{\le} \left(1-\frac{\alpha}{2}\right)\left(\frac{\bar{L}+\Gamma_t\mu}{\bar{L}+\Gamma_{t+1}\mu}\right)^2\left\|w^t-u^t\right\|^2 + \frac{2}{\alpha}\left(\frac{\gamma_{t+1}}{\bar{L}+\Gamma_{t+1}\mu}\right)^2\left\|k^t-\nabla f(y^{t+1})\right\|^2$$

$$+ (1-\alpha)\left(\frac{\gamma_{t+1}}{\bar{L}+\Gamma_{t+1}\mu}\right)^2\mathbb{E}_t\left[\left\|g^{t+1}-\nabla f(y^{t+1})\right\|^2\right]$$

$$\overset{(18)}{\le} \left(1-\frac{\alpha}{2}\right)\left(\frac{\bar{L}+\Gamma_t\mu}{\bar{L}+\Gamma_{t+1}\mu}\right)^2\left\|w^t-u^t\right\|^2$$

$$+ \frac{4}{\alpha}\left(\frac{\gamma_{t+1}}{\bar{L}+\Gamma_{t+1}\mu}\right)^2\left\|k^t-\nabla f(z^t)\right\|^2$$

$$+ \frac{4}{\alpha}\left(\frac{\gamma_{t+1}}{\bar{L}+\Gamma_{t+1}\mu}\right)^2\left\|\nabla f(z^t)-\nabla f(y^{t+1})\right\|^2$$

$$+ (1-\alpha)\left(\frac{\gamma_{t+1}}{\bar{L}+\Gamma_{t+1}\mu}\right)^2\mathbb{E}_t\left[\left\|g^{t+1}-\nabla f(y^{t+1})\right\|^2\right].$$

One can substitute (35) from Lemma E.3 to the last inequality and get

$$\mathbb{E}_t\left[\left\|w^{t+1}-u^{t+1}\right\|^2\right]$$

$$\le \left(1-\frac{\alpha}{2}\right)\left(\frac{\bar{L}+\Gamma_t\mu}{\bar{L}+\Gamma_{t+1}\mu}\right)^2\left\|w^t-u^t\right\|^2$$

$$+ \frac{4}{\alpha}\left(\frac{\gamma_{t+1}}{\bar{L}+\Gamma_{t+1}\mu}\right)^2\left\|k^t-\nabla f(z^t)\right\|^2$$

$$+ \frac{4}{\alpha} \left( \frac{\gamma_{t+1}}{\bar{L} + \Gamma_{t+1}\mu} \right)^2 \left\| \nabla f(z^t) - \nabla f(y^{t+1}) \right\|^2$$

$$+ (1-\alpha) \left( \frac{\gamma_{t+1}}{\bar{L} + \Gamma_{t+1}\mu} \right)^2 \left( \frac{2\omega}{n^2} \sum_{i=1}^{n} \left\| \nabla f_i(z^t) - h_i^t \right\|^2 + \frac{4\omega L_{\max}}{n} D_f(z^t, y^{t+1}) \right).$$

Using Lemma D.1 and $1 - \alpha \le 1$, we obtain

$$\mathbb{E}_t \left[ \left\| w^{t+1} - u^{t+1} \right\|^2 \right]$$

$$\le \left( 1 - \frac{\alpha}{2} \right) \left( \frac{\bar{L} + \Gamma_t \mu}{\bar{L} + \Gamma_{t+1}\mu} \right)^2 \left\| w^t - u^t \right\|^2$$

$$+ \frac{4}{\alpha} \left( \frac{\gamma_{t+1}}{\bar{L} + \Gamma_{t+1}\mu} \right)^2 \left\| k^t - \nabla f(z^t) \right\|^2$$

$$+ \frac{2\omega}{n} \left( \frac{\gamma_{t+1}}{\bar{L} + \Gamma_{t+1}\mu} \right)^2 \frac{1}{n} \sum_{i=1}^{n} \left\| \nabla f_i(z^t) - h_i^t \right\|^2 + \left( \frac{\gamma_{t+1}}{\bar{L} + \Gamma_{t+1}\mu} \right)^2 \left( \frac{4\omega L_{\max}}{n} + \frac{8L}{\alpha} \right) D_f(z^t, y^{t+1})$$

$$\le \left( 1 - \frac{\alpha}{2} \right) \left\| w^t - u^t \right\|^2$$

$$+ \frac{4}{\alpha} \left( \frac{\gamma_{t+1}}{\bar{L} + \Gamma_{t+1}\mu} \right)^2 \left\| k^t - \nabla f(z^t) \right\|^2$$

$$+ \frac{2\omega}{n} \left( \frac{\gamma_{t+1}}{\bar{L} + \Gamma_{t+1}\mu} \right)^2 \frac{1}{n} \sum_{i=1}^{n} \left\| \nabla f_i(z^t) - h_i^t \right\|^2 + \left( \frac{\gamma_{t+1}}{\bar{L} + \Gamma_{t+1}\mu} \right)^2 \left( \frac{4\omega L_{\max}}{n} + \frac{8L}{\alpha} \right) D_f(z^t, y^{t+1}),$$

where we use that $\Gamma_{t+1} \ge \Gamma_t$ for all $t \ge 0$. $\qquad \square$

**Lemma E.7.** *Suppose that Assumptions 1.2 and 1.3 hold. Then, for Algorithm 1, the following inequality holds:*

$$\mathbb{E}_t \left[ \left\| k^{t+1} - \nabla f(z^{t+1}) \right\|^2 \right]$$

$$\le 2p \mathbb{E}_t \left[ \left\| v^t - \nabla f(z^t) \right\|^2 \right] + 8pL D_f(z^t, y^{t+1}) + 4pL^2 \mathbb{E}_t \left[ \left\| x^{t+1} - y^{t+1} \right\|^2 \right] + (1-p) \left\| k^t - \nabla f(z^t) \right\|^2.$$
$$\tag{42}$$

*Proof.* Note that $k^{t+1}$ and $z^{t+1}$ are coupled by the same random variable $c^t$. Therefore,

$$\mathbb{E}_t \left[ \left\| k^{t+1} - \nabla f(z^{t+1}) \right\|^2 \right]$$

$$= p \mathbb{E}_t \left[ \left\| v^t - \nabla f(x^{t+1}) \right\|^2 \right] + (1-p) \left\| k^t - \nabla f(z^t) \right\|^2$$

$$\overset{(18)}{\le} 2p \mathbb{E}_t \left[ \left\| v^t - \nabla f(z^t) \right\|^2 \right] + 4p \left\| \nabla f(z^t) - \nabla f(y^{t+1}) \right\|^2 + 4p \mathbb{E}_t \left[ \left\| \nabla f(y^{t+1}) - \nabla f(x^{t+1}) \right\|^2 \right]$$

$$+ (1-p) \left\| k^t - \nabla f(z^t) \right\|^2$$

$$\le 2p \mathbb{E}_t \left[ \left\| v^t - \nabla f(z^t) \right\|^2 \right] + 8pL D_f(z^t, y^{t+1}) + 4pL^2 \mathbb{E}_t \left[ \left\| x^{t+1} - y^{t+1} \right\|^2 \right] + (1-p) \left\| k^t - \nabla f(z^t) \right\|^2,$$

where we use Assumptions 1.2 and Lemma D.1. $\qquad \square$

**Lemma E.8.** *Suppose that Assumptions 1.2, 1.1 and 1.3 hold. The momentum $\tau \in (0,1]$ and the probability $p \in (0,1]$. Then, for Algorithm 1, the following inequality holds:*

$$\mathbb{E}_t \left[ \left\| v^{t+1} - \nabla f(z^{t+1}) \right\|^2 \right]$$

$$\le \left( 1 - \frac{\tau}{2} \right) \left\| v^t - \nabla f(z^t) \right\|^2 + \frac{2\tau^2 \omega}{n^2} \sum_{i=1}^{n} \left\| h_i^t - \nabla f_i(z^t) \right\|^2$$

$$+ \left( 4p \left( 1 + \frac{2p}{\tau} \right) L + \frac{8p\tau^2 \omega L_{\max}}{n} \right) D_f(z^t, y^{t+1}) + \left( 2p \left( 1 + \frac{2p}{\tau} \right) L^2 + \frac{4p\tau^2 \omega \widehat{L}^2}{n} \right) \mathbb{E}_t \left[ \left\| x^{t+1} - y^{t+1} \right\|^2 \right].$$
$$\tag{43}$$

*Proof.* Using the definition of $v^{t+1}$, we get

$$\mathbb{E}_t\left[\left\|v^{t+1} - \nabla f(z^{t+1})\right\|^2\right] = \mathbb{E}_t\left[\left\|(1-\tau)v^t + \tau\left(h^t + \frac{1}{n}\sum_{i=1}^n \mathcal{C}_i^{D,z}\left(\nabla f_i(z^{t+1}) - h_i^t\right)\right) - \nabla f(z^{t+1})\right\|^2\right].$$

Assumption 2.4, including the independence and the unbiasedness of the compressors, insures that

$$\mathbb{E}_t\left[\left\|v^{t+1} - \nabla f(z^{t+1})\right\|^2\right]$$

$$\stackrel{(23)}{=} (1-\tau)^2\mathbb{E}_t\left[\left\|v^t - \nabla f(z^{t+1})\right\|^2\right] + \tau^2\mathbb{E}_t\left[\left\|\frac{1}{n}\sum_{i=1}^n \mathcal{C}_i^{D,z}\left(\nabla f_i(z^{t+1}) - h_i^t\right) - (\nabla f(z^{t+1}) - h^t)\right\|^2\right]$$

$$= (1-\tau)^2\mathbb{E}_t\left[\left\|v^t - \nabla f(z^{t+1})\right\|^2\right] + \frac{\tau^2}{n^2}\sum_{i=1}^n \mathbb{E}_t\left[\left\|\mathcal{C}_i^{D,z}\left(\nabla f_i(z^{t+1}) - h_i^t\right) - (\nabla f(z^{t+1}) - h^t)\right\|^2\right]$$

$$\leq (1-\tau)^2\mathbb{E}_t\left[\left\|v^t - \nabla f(z^{t+1})\right\|^2\right] + \frac{\tau^2\omega}{n^2}\sum_{i=1}^n \mathbb{E}_t\left[\left\|\nabla f_i(z^{t+1}) - h_i^t\right\|^2\right].$$

Using the definition of $z^{t+1}$, we have

$$\mathbb{E}_t\left[\left\|v^{t+1} - \nabla f(z^{t+1})\right\|^2\right]$$

$$\leq (1-\tau)^2(1-p)\left\|v^t - \nabla f(z^t)\right\|^2 + (1-\tau)^2 p\mathbb{E}_t\left[\left\|v^t - \nabla f(x^{t+1})\right\|^2\right]$$

$$\quad + \frac{(1-p)\tau^2\omega}{n^2}\sum_{i=1}^n \left\|h_i^t - \nabla f_i(z^t)\right\|^2 + \frac{p\tau^2\omega}{n^2}\sum_{i=1}^n \mathbb{E}_t\left[\left\|h_i^t - \nabla f_i(x^{t+1})\right\|^2\right]$$

$$\stackrel{(17),(18)}{\leq} (1-\tau)^2(1-p)\left\|v^t - \nabla f(z^t)\right\|^2$$

$$\quad + (1-\tau)^2\left(1 + \frac{\tau}{2p}\right)p\left\|v^t - \nabla f(z^t)\right\|^2 + (1-\tau)^2\left(1 + \frac{2p}{\tau}\right)p\mathbb{E}_t\left[\left\|\nabla f(z^t) - \nabla f(x^{t+1})\right\|^2\right]$$

$$\quad + \frac{(1-p)\tau^2\omega}{n^2}\sum_{i=1}^n \left\|h_i^t - \nabla f_i(z^t)\right\|^2$$

$$\quad + \frac{2p\tau^2\omega}{n^2}\sum_{i=1}^n \mathbb{E}_t\left[\left\|h_i^t - \nabla f_i(z^t)\right\|^2\right] + \frac{2p\tau^2\omega}{n^2}\sum_{i=1}^n \mathbb{E}_t\left[\left\|\nabla f_i(z^t) - \nabla f_i(x^{t+1})\right\|^2\right]$$

$$= (1-\tau)^2\left(1 + \frac{\tau}{2}\right)\left\|v^t - \nabla f(z^t)\right\|^2 + (1-\tau)^2\left(1 + \frac{2p}{\tau}\right)p\mathbb{E}_t\left[\left\|\nabla f(z^t) - \nabla f(x^{t+1})\right\|^2\right]$$

$$\quad + \frac{(1+p)\tau^2\omega}{n^2}\sum_{i=1}^n \left\|h_i^t - \nabla f_i(z^t)\right\|^2 + \frac{2p\tau^2\omega}{n^2}\sum_{i=1}^n \mathbb{E}_t\left[\left\|\nabla f_i(z^t) - \nabla f_i(x^{t+1})\right\|^2\right].$$

Using $0 \leq 1 - \tau \leq 1$, $p \in (0,1]$ and (20), we get

$$\mathbb{E}_t\left[\left\|v^{t+1} - \nabla f(z^{t+1})\right\|^2\right]$$

$$\leq \left(1 - \frac{\tau}{2}\right)\left\|v^t - \nabla f(z^t)\right\|^2 + \left(1 + \frac{2p}{\tau}\right)p\mathbb{E}_t\left[\left\|\nabla f(z^t) - \nabla f(x^{t+1})\right\|^2\right]$$

$$\quad + \frac{2\tau^2\omega}{n^2}\sum_{i=1}^n \left\|h_i^t - \nabla f_i(z^t)\right\|^2 + \frac{2p\tau^2\omega}{n^2}\sum_{i=1}^n \mathbb{E}_t\left[\left\|\nabla f_i(z^t) - \nabla f_i(x^{t+1})\right\|^2\right]$$

$$\stackrel{(18)}{\leq} \left(1 - \frac{\tau}{2}\right)\left\|v^t - \nabla f(z^t)\right\|^2$$

$$\quad + 2p\left(1 + \frac{2p}{\tau}\right)\left\|\nabla f(z^t) - \nabla f(y^{t+1})\right\|^2 + 2p\left(1 + \frac{2p}{\tau}\right)\mathbb{E}_t\left[\left\|\nabla f(y^{t+1}) - \nabla f(x^{t+1})\right\|^2\right]$$

$$+ \frac{2\tau^2\omega}{n^2} \sum_{i=1}^{n} \left\| h_i^t - \nabla f_i(z^t) \right\|^2$$

$$+ \frac{4p\tau^2\omega}{n^2} \sum_{i=1}^{n} \left\| \nabla f_i(z^t) - \nabla f_i(y^{t+1}) \right\|^2 + \frac{4p\tau^2\omega}{n^2} \sum_{i=1}^{n} \mathbb{E}_t \left[ \left\| \nabla f_i(x^{t+1}) - \nabla f_i(y^{t+1}) \right\|^2 \right].$$

It is left to use Assumptions 1.2 and 1.1 with Lemma D.1 to obtain

$$\mathbb{E}_t \left[ \left\| v^{t+1} - \nabla f(z^{t+1}) \right\|^2 \right]$$

$$\leq \left( 1 - \frac{\tau}{2} \right) \left\| v^t - \nabla f(z^t) \right\|^2$$

$$+ 4p \left( 1 + \frac{2p}{\tau} \right) L D_f(z^t, y^{t+1}) + 2p \left( 1 + \frac{2p}{\tau} \right) L^2 \mathbb{E}_t \left[ \left\| y^{t+1} - x^{t+1} \right\|^2 \right]$$

$$+ \frac{2\tau^2\omega}{n^2} \sum_{i=1}^{n} \left\| h_i^t - \nabla f_i(z^t) \right\|^2$$

$$+ \frac{8p\tau^2\omega L_{\max}}{n} D_f(z^t, y^{t+1}) + \frac{4p\tau^2\omega \widehat{L}^2}{n} \mathbb{E}_t \left[ \left\| x^{t+1} - y^{t+1} \right\|^2 \right]$$

$$= \left( 1 - \frac{\tau}{2} \right) \left\| v^t - \nabla f(z^t) \right\|^2 + \frac{2\tau^2\omega}{n^2} \sum_{i=1}^{n} \left\| h_i^t - \nabla f_i(z^t) \right\|^2$$

$$+ \left( 4p \left( 1 + \frac{2p}{\tau} \right) L + \frac{8p\tau^2\omega L_{\max}}{n} \right) D_f(z^t, y^{t+1}) + \left( 2p \left( 1 + \frac{2p}{\tau} \right) L^2 + \frac{4p\tau^2\omega \widehat{L}^2}{n} \right) \mathbb{E}_t \left[ \left\| x^{t+1} - y^{t+1} \right\|^2 \right].$$

$\square$

### E.4 Main theorem

**Theorem E.9.** *Suppose that Assumptions 1.2, 1.1, 1.3, 2.4 hold. Let*

$$\bar{L} = 660508 \times \max \left\{ \frac{L}{\alpha}, \frac{Lp}{\alpha\tau}, \frac{\sqrt{LL_{\max}} p \sqrt{\omega\tau}}{\alpha\beta\sqrt{n}}, \frac{\sqrt{LL_{\max}} \sqrt{p} \sqrt{\omega\tau}}{\alpha\sqrt{\beta}\sqrt{n}}, \frac{L_{\max}\omega p^2}{\beta^2 n}, \frac{L_{\max}\omega}{n} \right\}, \quad (44)$$

*$\beta \leq \frac{1}{\omega+1}$, and $\theta_{\min} = \frac{1}{4} \min \left\{ 1, \frac{\alpha}{p}, \frac{\tau}{p}, \frac{\beta}{p} \right\}$. For all $t \geq 0$, Algorithm 1 guarantees that*

$$\Gamma_{t+1} \left( \mathbb{E} \left[ f(z^{t+1}) - f(x^*) \right] + \kappa \mathbb{E} \left[ \frac{1}{n} \sum_{i=1}^{n} \left\| h_i^{t+1} - \nabla f_i(z^{t+1}) \right\|^2 \right] + \nu_{t+1} \mathbb{E} \left[ \left\| w^{t+1} - u^{t+1} \right\|^2 \right] \right.$$

$$\left. + \rho \mathbb{E} \left[ \left\| k^{t+1} - \nabla f(z^{t+1}) \right\|^2 \right] + \lambda \mathbb{E} \left[ \left\| v^{t+1} - \nabla f(z^{t+1}) \right\|^2 \right] \right) + \frac{\bar{L} + \Gamma_{t+1}\mu}{2} \mathbb{E} \left[ \left\| u^{t+1} - x^* \right\|^2 \right]$$

$$\leq \Gamma_t \left( \mathbb{E} \left[ f(z^t) - f(x^*) \right] + \kappa \mathbb{E} \left[ \frac{1}{n} \sum_{i=1}^{n} \left\| h_i^t - \nabla f_i(z^t) \right\|^2 \right] + \nu_t \mathbb{E} \left[ \left\| w^t - u^t \right\|^2 \right] \right.$$

$$\left. + \rho \mathbb{E} \left[ \left\| k^t - \nabla f(z^t) \right\|^2 \right] + \lambda \mathbb{E} \left[ \left\| v^t - \nabla f(z^t) \right\|^2 \right] \right) + \frac{\bar{L} + \Gamma_t\mu}{2} \mathbb{E} \left[ \left\| u^t - x^* \right\|^2 \right]$$

(45)

*for some $\kappa, \rho, \lambda, \nu_t \geq 0$.*

*Proof.* We fix some constants $\kappa, \rho, \lambda, \nu_t \geq 0$ for all $t \geq 0$ that we define later. By combining Lemma E.4 with $\kappa \times$ (39) from Lemma E.5, $\nu_t \times$ (41) from Lemma E.6, $\rho \times$ (42) from Lemma E.7, and $\lambda \times$ (43) from Lemma E.8, we get the following inequality:

$$\mathbb{E}_t \left[ f(z^{t+1}) - f(x^*) \right] + \kappa \mathbb{E}_t \left[ \frac{1}{n} \sum_{i=1}^{n} \left\| h_i^{t+1} - \nabla f_i(z^{t+1}) \right\|^2 \right] + \nu_t \mathbb{E}_t \left[ \left\| w^{t+1} - u^{t+1} \right\|^2 \right]$$

$$+ \rho\mathbb{E}_t\left[\left\|k^{t+1} - \nabla f(z^{t+1})\right\|^2\right] + \lambda\mathbb{E}_t\left[\left\|v^{t+1} - \nabla f(z^{t+1})\right\|^2\right]$$

$$\leq (1 - p\theta_{t+1})\left(f(z^t) - f(x^*)\right)$$

$$+ \frac{2p\omega}{n\bar{L}}\left(\frac{1}{n}\sum_{i=1}^n \left\|h_i^t - \nabla f_i(z^t)\right\|^2\right)$$

$$+ p\theta_{t+1}\left(\frac{\bar{L} + \Gamma_t\mu}{2\gamma_{t+1}}\left\|u^t - x^*\right\|^2 - \frac{\bar{L} + \Gamma_{t+1}\mu}{2\gamma_{t+1}}\mathbb{E}_t\left[\left\|u^{t+1} - x^*\right\|^2\right]\right)$$

$$+ p\left(\frac{4\omega L_{\max}}{n\bar{L}} + \theta_{t+1} - 1\right)D_f(z^t, y^{t+1})$$

$$+ \frac{pL}{2}\mathbb{E}_t\left[\left\|x^{t+1} - y^{t+1}\right\|^2\right] - \frac{p\theta_{t+1}^2\bar{L}}{2}\mathbb{E}_t\left[\left\|u^{t+1} - u^t\right\|^2\right]$$

$$+ \kappa\left(8p\left(1 + \frac{p}{\beta}\right)L_{\max}D_f(z^t, y^{t+1}) + 4p\left(1 + \frac{p}{\beta}\right)\widehat{L}^2\mathbb{E}_t\left[\left\|x^{t+1} - y^{t+1}\right\|^2\right] + \left(1 - \frac{\beta}{2}\right)\frac{1}{n}\sum_{i=1}^n \left\|h_i^t - \nabla f_i(z^t)\right\|^2\right)$$

$$+ \nu_t\left(\left(1 - \frac{\alpha}{2}\right)\left\|w^t - u^t\right\|^2 + \frac{4}{\alpha}\left(\frac{\gamma_{t+1}}{\bar{L} + \Gamma_{t+1}\mu}\right)^2\left\|k^t - \nabla f(z^t)\right\|^2\right.$$

$$\left. + \frac{2\omega}{n}\left(\frac{\gamma_{t+1}}{\bar{L} + \Gamma_{t+1}\mu}\right)^2\frac{1}{n}\sum_{i=1}^n \left\|\nabla f_i(z^t) - h_i^t\right\|^2 + \left(\frac{\gamma_{t+1}}{\bar{L} + \Gamma_{t+1}\mu}\right)^2\left(\frac{4\omega L_{\max}}{n} + \frac{8L}{\alpha}\right)D_f(z^t, y^{t+1})\right)$$

$$+ \rho\left(2p\mathbb{E}_t\left[\left\|v^t - \nabla f(z^t)\right\|^2\right] + 8pLD_f(z^t, y^{t+1}) + 4pL^2\mathbb{E}_t\left[\left\|x^{t+1} - y^{t+1}\right\|^2\right] + (1 - p)\left\|k^t - \nabla f(z^t)\right\|^2\right)$$

$$+ \lambda\left(\left(1 - \frac{\tau}{2}\right)\left\|v^t - \nabla f(z^t)\right\|^2 + \frac{2\tau^2\omega}{n^2}\sum_{i=1}^n \left\|h_i^t - \nabla f_i(z^t)\right\|^2\right.$$

$$\left. + \left(4p\left(1 + \frac{2p}{\tau}\right)L + \frac{8p\tau^2\omega L_{\max}}{n}\right)D_f(z^t, y^{t+1}) + \left(2p\left(1 + \frac{2p}{\tau}\right)L^2 + \frac{4p\tau^2\omega\widehat{L}^2}{n}\right)\mathbb{E}_t\left[\left\|x^{t+1} - y^{t+1}\right\|^2\right]\right).$$

We regroup the terms and obtain

$$\mathbb{E}_t\left[f(z^{t+1}) - f(x^*)\right] + \kappa\mathbb{E}_t\left[\frac{1}{n}\sum_{i=1}^n \left\|h_i^{t+1} - \nabla f_i(z^{t+1})\right\|^2\right] + \nu_t\mathbb{E}_t\left[\left\|w^{t+1} - u^{t+1}\right\|^2\right]$$

$$+ \rho\mathbb{E}_t\left[\left\|k^{t+1} - \nabla f(z^{t+1})\right\|^2\right] + \lambda\mathbb{E}_t\left[\left\|v^{t+1} - \nabla f(z^{t+1})\right\|^2\right]$$

$$\leq (1 - p\theta_{t+1})\left(f(z^t) - f(x^*)\right)$$

$$+ p\theta_{t+1}\left(\frac{\bar{L} + \Gamma_t\mu}{2\gamma_{t+1}}\left\|u^t - x^*\right\|^2 - \frac{\bar{L} + \Gamma_{t+1}\mu}{2\gamma_{t+1}}\mathbb{E}_t\left[\left\|u^{t+1} - x^*\right\|^2\right]\right)$$

$$+ p\left(\frac{4\omega L_{\max}}{n\bar{L}} + \kappa 8\left(1 + \frac{p}{\beta}\right)L_{\max} + \nu_t\left(\frac{\gamma_{t+1}}{\bar{L} + \Gamma_{t+1}\mu}\right)^2\left(\frac{4\omega L_{\max}}{pn} + \frac{8L}{p\alpha}\right) + \rho 8L +\right.$$

$$\left. + \lambda\left(4\left(1 + \frac{2p}{\tau}\right)L + \frac{8\tau^2\omega L_{\max}}{n}\right) + \theta_{t+1} - 1\right)D_f(z^t, y^{t+1})$$

$$+ \left(\frac{pL}{2} + \kappa 4p\left(1 + \frac{p}{\beta}\right)\widehat{L}^2 + \rho 4pL^2 + \lambda\left(2p\left(1 + \frac{2p}{\tau}\right)L^2 + \frac{4p\tau^2\omega\widehat{L}^2}{n}\right)\right)\mathbb{E}_t\left[\left\|x^{t+1} - y^{t+1}\right\|^2\right]$$

$$- \frac{p\theta_{t+1}^2\bar{L}}{2}\mathbb{E}_t\left[\left\|u^{t+1} - u^t\right\|^2\right]$$

$$+ \nu_t\left(1 - \frac{\alpha}{2}\right)\left\|w^t - u^t\right\|^2$$

$$+ \left(\nu_t\frac{4}{\alpha}\left(\frac{\gamma_{t+1}}{\bar{L} + \Gamma_{t+1}\mu}\right)^2 + \rho(1 - p)\right)\left\|k^t - \nabla f(z^t)\right\|^2$$

$$+ \left( \rho 2p + \lambda \left( 1 - \frac{\tau}{2} \right) \right) \left\| v^t - \nabla f(z^t) \right\|^2$$
$$+ \left( \frac{2p\omega}{n\bar{L}} + \nu_t \frac{2\omega}{n} \left( \frac{\gamma_{t+1}}{\bar{L} + \Gamma_{t+1}\mu} \right)^2 + \lambda \frac{2\tau^2\omega}{n} + \kappa \left( 1 - \frac{\beta}{2} \right) \right) \left( \frac{1}{n} \sum_{i=1}^{n} \left\| h_i^t - \nabla f_i(z^t) \right\|^2 \right).$$

Using $x^{t+1} - y^{t+1} = \theta_{t+1} \left( u^{t+1} - w^t \right)$, we get (we mark the changes with color)

$$\mathbb{E}_t \left[ f(z^{t+1}) - f(x^*) \right] + \kappa \mathbb{E}_t \left[ \frac{1}{n} \sum_{i=1}^{n} \left\| h_i^{t+1} - \nabla f_i(z^{t+1}) \right\|^2 \right] + \nu_t \mathbb{E}_t \left[ \left\| w^{t+1} - u^{t+1} \right\|^2 \right]$$
$$+ \rho \mathbb{E}_t \left[ \left\| k^{t+1} - \nabla f(z^{t+1}) \right\|^2 \right] + \lambda \mathbb{E}_t \left[ \left\| v^{t+1} - \nabla f(z^{t+1}) \right\|^2 \right]$$
$$\leq (1 - p\theta_{t+1}) \left( f(z^t) - f(x^*) \right)$$
$$+ p\theta_{t+1} \left( \frac{\bar{L} + \Gamma_t \mu}{2\gamma_{t+1}} \left\| u^t - x^* \right\|^2 - \frac{\bar{L} + \Gamma_{t+1}\mu}{2\gamma_{t+1}} \mathbb{E}_t \left[ \left\| u^{t+1} - x^* \right\|^2 \right] \right)$$
$$+ p \left( \frac{4\omega L_{\max}}{n\bar{L}} + \kappa 8 \left( 1 + \frac{p}{\beta} \right) L_{\max} + \nu_t \left( \frac{\gamma_{t+1}}{\bar{L} + \Gamma_{t+1}\mu} \right)^2 \left( \frac{4\omega L_{\max}}{pn} + \frac{8L}{p\alpha} \right) + \rho 8L + \right.$$
$$\left. + \lambda \left( 4 \left( 1 + \frac{2p}{\tau} \right) L + \frac{8\tau^2 \omega L_{\max}}{n} \right) + \theta_{t+1} - 1 \right) D_f(z^t, y^{t+1})$$
$$+ \color{red}{\theta_{t+1}^2} \left( \frac{pL}{2} + \kappa 4p \left( 1 + \frac{p}{\beta} \right) \widehat{L}^2 + \rho 4pL^2 + \lambda \left( 2p \left( 1 + \frac{2p}{\tau} \right) L^2 + \frac{4p\tau^2 \omega \widehat{L}^2}{n} \right) \right) \mathbb{E}_t \left[ \left\| u^{t+1} - w^t \right\|^2 \right]$$
$$- \frac{p\theta_{t+1}^2 \bar{L}}{2} \mathbb{E}_t \left[ \left\| u^{t+1} - u^t \right\|^2 \right]$$
$$+ \nu_t \left( 1 - \frac{\alpha}{2} \right) \left\| w^t - u^t \right\|^2$$
$$+ \left( \nu_t \frac{4}{\alpha} \left( \frac{\gamma_{t+1}}{\bar{L} + \Gamma_{t+1}\mu} \right)^2 + \rho(1 - p) \right) \left\| k^t - \nabla f(z^t) \right\|^2$$
$$+ \left( \rho 2p + \lambda \left( 1 - \frac{\tau}{2} \right) \right) \left\| v^t - \nabla f(z^t) \right\|^2$$
$$+ \left( \frac{2p\omega}{n\bar{L}} + \nu_t \frac{2\omega}{n} \left( \frac{\gamma_{t+1}}{\bar{L} + \Gamma_{t+1}\mu} \right)^2 + \lambda \frac{2\tau^2\omega}{n} + \kappa \left( 1 - \frac{\beta}{2} \right) \right) \left( \frac{1}{n} \sum_{i=1}^{n} \left\| h_i^t - \nabla f_i(z^t) \right\|^2 \right).$$

The inequality (17) implies $\left\| u^{t+1} - w^t \right\|^2 \leq 2 \left\| u^{t+1} - u^t \right\|^2 + 2 \left\| u^t - w^t \right\|^2$ and

$$\mathbb{E}_t \left[ f(z^{t+1}) - f(x^*) \right] + \kappa \mathbb{E}_t \left[ \frac{1}{n} \sum_{i=1}^{n} \left\| h_i^{t+1} - \nabla f_i(z^{t+1}) \right\|^2 \right] + \nu_t \mathbb{E}_t \left[ \left\| w^{t+1} - u^{t+1} \right\|^2 \right]$$
$$+ \rho \mathbb{E}_t \left[ \left\| k^{t+1} - \nabla f(z^{t+1}) \right\|^2 \right] + \lambda \mathbb{E}_t \left[ \left\| v^{t+1} - \nabla f(z^{t+1}) \right\|^2 \right]$$
$$\leq (1 - p\theta_{t+1}) \left( f(z^t) - f(x^*) \right)$$
$$+ p\theta_{t+1} \left( \frac{\bar{L} + \Gamma_t \mu}{2\gamma_{t+1}} \left\| u^t - x^* \right\|^2 - \frac{\bar{L} + \Gamma_{t+1}\mu}{2\gamma_{t+1}} \mathbb{E}_t \left[ \left\| u^{t+1} - x^* \right\|^2 \right] \right)$$
$$+ p \left( \frac{4\omega L_{\max}}{n\bar{L}} + \kappa 8 \left( 1 + \frac{p}{\beta} \right) L_{\max} + \nu_t \left( \frac{\gamma_{t+1}}{\bar{L} + \Gamma_{t+1}\mu} \right)^2 \left( \frac{4\omega L_{\max}}{pn} + \frac{8L}{p\alpha} \right) + \rho 8L + \right.$$
$$\left. + \lambda \left( 4 \left( 1 + \frac{2p}{\tau} \right) L + \frac{8\tau^2 \omega L_{\max}}{n} \right) + \theta_{t+1} - 1 \right) D_f(z^t, y^{t+1})$$
$$+ \color{red}{2\theta_{t+1}^2} \left( \frac{pL}{2} + \kappa 4p \left( 1 + \frac{p}{\beta} \right) \widehat{L}^2 + \rho 4pL^2 + \lambda \left( 2p \left( 1 + \frac{2p}{\tau} \right) L^2 + \frac{4p\tau^2 \omega \widehat{L}^2}{n} \right) \right) \mathbb{E}_t \left[ \left\| u^{t+1} - u^t \right\|^2 \right]$$
$$- \frac{p\theta_{t+1}^2 \bar{L}}{2} \mathbb{E}_t \left[ \left\| u^{t+1} - u^t \right\|^2 \right]$$

$$+ \left( 2\theta_{t+1}^2 \left( \frac{pL}{2} + \kappa 4p \left( 1 + \frac{p}{\beta} \right) \widehat{L}^2 + \rho 4pL^2 + \lambda \left( 2p \left( 1 + \frac{2p}{\tau} \right) L^2 + \frac{4p\tau^2 \omega \widehat{L}^2}{n} \right) \right) + \nu_t \left( 1 - \frac{\alpha}{2} \right) \right) \left\| w^t - u^t \right\|^2$$

$$+ \left( \nu_t \frac{4}{\alpha} \left( \frac{\gamma_{t+1}}{\bar{L} + \Gamma_{t+1}\mu} \right)^2 + \rho(1-p) \right) \left\| k^t - \nabla f(z^t) \right\|^2$$

$$+ \left( \rho 2p + \lambda \left( 1 - \frac{\tau}{2} \right) \right) \left\| v^t - \nabla f(z^t) \right\|^2$$

$$+ \left( \frac{2p\omega}{n\bar{L}} + \nu_t \frac{2\omega}{n} \left( \frac{\gamma_{t+1}}{\bar{L} + \Gamma_{t+1}\mu} \right)^2 + \lambda \frac{2\tau^2 \omega}{n} + \kappa \left( 1 - \frac{\beta}{2} \right) \right) \left( \frac{1}{n} \sum_{i=1}^n \left\| h_i^t - \nabla f_i(z^t) \right\|^2 \right).$$

Now, we want to find appropriate $\kappa$, $\rho$, $\lambda$, and $\nu_t$ such that

$$2\theta_{t+1}^2 \left( \frac{pL}{2} + \kappa 4p \left( 1 + \frac{p}{\beta} \right) \widehat{L}^2 + \rho 4pL^2 + \lambda \left( 2p \left( 1 + \frac{2p}{\tau} \right) L^2 + \frac{4p\tau^2 \omega \widehat{L}^2}{n} \right) \right) + \nu_t \left( 1 - \frac{\alpha}{2} \right) \leq \nu_t \left( 1 - \frac{\alpha}{4} \right),$$

$$\nu_t \frac{4}{\alpha} \left( \frac{\gamma_{t+1}}{\bar{L} + \Gamma_{t+1}\mu} \right)^2 + \rho(1-p) \leq \rho \left( 1 - \frac{p}{2} \right),$$

$$\rho 2p + \lambda \left( 1 - \frac{\tau}{2} \right) \leq \lambda \left( 1 - \frac{\tau}{4} \right),$$

$$\frac{2p\omega}{n\bar{L}} + \nu_t \frac{2\omega}{n} \left( \frac{\gamma_{t+1}}{\bar{L} + \Gamma_{t+1}\mu} \right)^2 + \lambda \frac{2\tau^2 \omega}{n} + \kappa \left( 1 - \frac{\beta}{2} \right) \leq \kappa \left( 1 - \frac{\beta}{4} \right).$$

(46)

We analyze inequalities (46) in the following lemma:

**Lemma E.10** (First Symbolically Computed). *Assume that for the parameter $\bar{L}$, the inequalities from Sections I and J hold. Then, for all $t \geq 0$, exists $\rho$ in (90), $\kappa$ in (88), $\lambda$ in (82), and $\nu_t$ in (83) such that (46) holds.*

We proof lemma separately in Section G. Using the lemma, we have

$$\mathbb{E}_t \left[ f(z^{t+1}) - f(x^*) \right] + \kappa \mathbb{E}_t \left[ \frac{1}{n} \sum_{i=1}^n \left\| h_i^{t+1} - \nabla f_i(z^{t+1}) \right\|^2 \right] + \nu_t \mathbb{E}_t \left[ \left\| w^{t+1} - u^{t+1} \right\|^2 \right]$$

$$+ \rho \mathbb{E}_t \left[ \left\| k^{t+1} - \nabla f(z^{t+1}) \right\|^2 \right] + \lambda \mathbb{E}_t \left[ \left\| v^{t+1} - \nabla f(z^{t+1}) \right\|^2 \right]$$

$$\leq (1 - p\theta_{t+1}) \left( f(z^t) - f(x^*) \right)$$

$$+ p\theta_{t+1} \left( \frac{\bar{L} + \Gamma_t\mu}{2\gamma_{t+1}} \left\| u^t - x^* \right\|^2 - \frac{\bar{L} + \Gamma_{t+1}\mu}{2\gamma_{t+1}} \mathbb{E}_t \left[ \left\| u^{t+1} - x^* \right\|^2 \right] \right)$$

$$+ p \left( \frac{4\omega L_{\max}}{n\bar{L}} + \kappa 8 \left( 1 + \frac{p}{\beta} \right) L_{\max} + \nu_t \left( \frac{\gamma_{t+1}}{\bar{L} + \Gamma_{t+1}\mu} \right)^2 \left( \frac{4\omega L_{\max}}{pn} + \frac{8L}{p\alpha} \right) + \rho 8L + \right.$$

$$\left. + \lambda \left( 4 \left( 1 + \frac{2p}{\tau} \right) L + \frac{8\tau^2 \omega L_{\max}}{n} \right) + \theta_{t+1} - 1 \right) D_f(z^t, y^{t+1})$$

$$+ 2\theta_{t+1}^2 \left( \frac{pL}{2} + \kappa 4p \left( 1 + \frac{p}{\beta} \right) \widehat{L}^2 + \rho 4pL^2 + \lambda \left( 2p \left( 1 + \frac{2p}{\tau} \right) L^2 + \frac{4p\tau^2 \omega \widehat{L}^2}{n} \right) \right) \mathbb{E}_t \left[ \left\| u^{t+1} - u^t \right\|^2 \right]$$

$$- \frac{p\theta_{t+1}^2 \bar{L}}{2} \mathbb{E}_t \left[ \left\| u^{t+1} - u^t \right\|^2 \right]$$

$$+ \nu_t \left( 1 - \frac{\alpha}{4} \right) \left\| w^t - u^t \right\|^2$$

$$+ \rho \left( 1 - \frac{p}{2} \right) \left\| k^t - \nabla f(z^t) \right\|^2$$

$$+ \lambda \left( 1 - \frac{\tau}{4} \right) \left\| v^t - \nabla f(z^t) \right\|^2$$

$$+ \kappa \left( 1 - \frac{\beta}{4} \right) \left( \frac{1}{n} \sum_{i=1}^n \left\| h_i^t - \nabla f_i(z^t) \right\|^2 \right).$$

Let us separately analyze the terms w.r.t. $D_f(z^t, y^{t+1})$ and $\mathbb{E}_t\left[\|u^{t+1} - u^t\|^2\right]$:

**Lemma E.11** (Second Symbolically Computed). *Consider the parameters $\rho$, $\kappa$, $\lambda$, and $\nu_t$ from Lemma E.10. Assume that for the parameter $\bar{L}$, the inequalities from Sections L and N hold, and the step size $\theta_{t+1} \leq 1/4$ for all $t \geq 0$. Then, for all $t \geq 0$, the following inequalities are satisfied:*

$$p\left(\frac{4\omega L_{\max}}{n\bar{L}} + \kappa 8\left(1 + \frac{p}{\beta}\right)L_{\max} + \nu_t\left(\frac{\gamma_{t+1}}{\bar{L} + \Gamma_{t+1}\mu}\right)^2\left(\frac{4\omega L_{\max}}{pn} + \frac{8L}{p\alpha}\right) + \rho 8L + \right.$$
$$\left. + \lambda\left(4\left(1 + \frac{2p}{\tau}\right)L + \frac{8\tau^2\omega L_{\max}}{n}\right) + \theta_{t+1} - 1\right)D_f(z^t, y^{t+1}) \leq 0 \tag{47}$$

*and*

$$2\theta_{t+1}^2\left(\frac{pL}{2} + \kappa 4p\left(1 + \frac{p}{\beta}\right)\widehat{L}^2 + \rho 4pL^2 + \lambda\left(2p\left(1 + \frac{2p}{\tau}\right)L^2 + \frac{4p\tau^2\omega\widehat{L}^2}{n}\right)\right)\mathbb{E}_t\left[\|u^{t+1} - u^t\|^2\right]$$
$$- \frac{p\theta_{t+1}^2\bar{L}}{2}\mathbb{E}_t\left[\|u^{t+1} - u^t\|^2\right] \leq 0. \tag{48}$$

We prove Lemma E.11 in Section H. Using the lemma, we get

$$\mathbb{E}_t\left[f(z^{t+1}) - f(x^*)\right] + \kappa\mathbb{E}_t\left[\frac{1}{n}\sum_{i=1}^n\|h_i^{t+1} - \nabla f_i(z^{t+1})\|^2\right] + \nu_t\mathbb{E}_t\left[\|w^{t+1} - u^{t+1}\|^2\right]$$
$$+ \rho\mathbb{E}_t\left[\|k^{t+1} - \nabla f(z^{t+1})\|^2\right] + \lambda\mathbb{E}_t\left[\|v^{t+1} - \nabla f(z^{t+1})\|^2\right]$$
$$\leq (1 - p\theta_{t+1})\left(f(z^t) - f(x^*)\right)$$
$$+ p\theta_{t+1}\left(\frac{\bar{L} + \Gamma_t\mu}{2\gamma_{t+1}}\|u^t - x^*\|^2 - \frac{\bar{L} + \Gamma_{t+1}\mu}{2\gamma_{t+1}}\mathbb{E}_t\left[\|u^{t+1} - x^*\|^2\right]\right)$$
$$+ \nu_t\left(1 - \frac{\alpha}{4}\right)\|w^t - u^t\|^2$$
$$+ \rho\left(1 - \frac{p}{2}\right)\|k^t - \nabla f(z^t)\|^2$$
$$+ \lambda\left(1 - \frac{\tau}{4}\right)\|v^t - \nabla f(z^t)\|^2$$
$$+ \kappa\left(1 - \frac{\beta}{4}\right)\left(\frac{1}{n}\sum_{i=1}^n\|h_i^t - \nabla f_i(z^t)\|^2\right).$$

Note that $0 \leq \nu_{t+1} \leq \nu_t$ for all $t \geq 0$, since $\theta_{t+1}$ is a non-increasing sequence (see Lemma E.1 and the definition of $\nu_t$ in (83)). Using

$$\theta_{t+1} \leq \frac{1}{4}\min\left\{1, \frac{\alpha}{p}, \frac{\tau}{p}, \frac{\beta}{p}\right\}$$

for all $t \geq 0$, we obtain

$$\mathbb{E}_t\left[f(z^{t+1}) - f(x^*)\right] + \kappa\mathbb{E}_t\left[\frac{1}{n}\sum_{i=1}^n\|h_i^{t+1} - \nabla f_i(z^{t+1})\|^2\right] + \nu_{t+1}\mathbb{E}_t\left[\|w^{t+1} - u^{t+1}\|^2\right]$$
$$+ \rho\mathbb{E}_t\left[\|k^{t+1} - \nabla f(z^{t+1})\|^2\right] + \lambda\mathbb{E}_t\left[\|v^{t+1} - \nabla f(z^{t+1})\|^2\right]$$
$$\leq (1 - p\theta_{t+1})\left(f(z^t) - f(x^*) + \kappa\left(\frac{1}{n}\sum_{i=1}^n\|h_i^t - \nabla f_i(z^t)\|^2\right) + \nu_t\|w^t - u^t\|^2\right.$$
$$\left. + \rho\|k^t - \nabla f(z^t)\|^2 + \lambda\|v^t - \nabla f(z^t)\|^2\right)$$

$$+ p\theta_{t+1}\left(\frac{\bar{L}+\Gamma_t\mu}{2\gamma_{t+1}}\left\|u^t - x^*\right\|^2 - \frac{\bar{L}+\Gamma_{t+1}\mu}{2\gamma_{t+1}}\mathbb{E}_t\left[\left\|u^{t+1}-x^*\right\|^2\right]\right).$$

Let us multiply the inequality by $\frac{\gamma_{t+1}}{p\theta_{t+1}}$ :

$$\frac{\gamma_{t+1}}{p\theta_{t+1}}\left(\mathbb{E}_t\left[f(z^{t+1}) - f(x^*)\right] + \kappa\mathbb{E}_t\left[\frac{1}{n}\sum_{i=1}^n\left\|h_i^{t+1} - \nabla f_i(z^{t+1})\right\|^2\right] + \nu_{t+1}\mathbb{E}_t\left[\left\|w^{t+1} - u^{t+1}\right\|^2\right]\right.$$

$$\left.+\rho\mathbb{E}_t\left[\left\|k^{t+1} - \nabla f(z^{t+1})\right\|^2\right] + \lambda\mathbb{E}_t\left[\left\|v^{t+1} - \nabla f(z^{t+1})\right\|^2\right]\right)$$

$$\leq \left(\frac{\gamma_{t+1}}{p\theta_{t+1}} - \gamma_{t+1}\right)\left(f(z^t) - f(x^*) + \kappa\left(\frac{1}{n}\sum_{i=1}^n\left\|h_i^t - \nabla f_i(z^t)\right\|^2\right) + \nu_t\left\|w^t - u^t\right\|^2\right.$$

$$\left.+ \rho\left\|k^t - \nabla f(z^t)\right\|^2 + \lambda\left\|v^t - \nabla f(z^t)\right\|^2\right)$$

$$+ \left(\frac{\bar{L}+\Gamma_t\mu}{2}\left\|u^t - x^*\right\|^2 - \frac{\bar{L}+\Gamma_{t+1}\mu}{2}\mathbb{E}_t\left[\left\|u^{t+1}-x^*\right\|^2\right]\right).$$

It is left to use that $\Gamma_{t+1} := \Gamma_t + \gamma_{t+1}$ and $\gamma_{t+1} = p\theta_{t+1}\Gamma_{t+1}$ (see Lemma E.1) and take the full expectation to obtain (45).

In the proof we require that, for the parameter $\bar{L}$, the inequalities from Sections I, J, L and N hold. In Section O, we show that these inequalities follow from (44). □

### E.5 Strongly-convex case

**Theorem E.12.** *Suppose that Assumptions 1.2, 1.1, 1.3, 2.4 hold. Let*

$$\bar{L} = 660508 \times \max\left\{\frac{L}{\alpha}, \frac{Lp}{\alpha\tau}, \frac{\sqrt{LL_{\max}}p\sqrt{\omega\tau}}{\alpha\beta\sqrt{n}}, \frac{\sqrt{LL_{\max}}\sqrt{p}\sqrt{\omega\tau}}{\alpha\sqrt{\beta}\sqrt{n}}, \frac{L_{\max}\omega p^2}{\beta^2 n}, \frac{L_{\max}\omega}{n}\right\}, \quad (49)$$

$\beta = \frac{1}{\omega+1}$, $\theta_{\min} = \frac{1}{4}\min\left\{1, \frac{\alpha}{p}, \frac{\tau}{p}, \frac{\beta}{p}\right\}$, $h_i^0 = \nabla f_i(z^0)$ *for all* $i \in [n]$, $w^0 = u^0$, $k^0 = \nabla f(z^0)$, $v^0 = \nabla f(z^0)$, *and* $\Gamma_0 \geq 1$. *Then Algorithm 1 guarantees that*

$$\mathbb{E}\left[f(z^T) - f(x^*)\right] + \frac{\mu}{2}\mathbb{E}\left[\left\|u^T - x^*\right\|^2\right] \leq 2\exp\left(-\frac{T}{Q}\right)\left((f(z^0) - f(x^*)) + \left(\frac{\bar{L}}{\Gamma_0} + \mu\right)\left\|u^0 - x^*\right\|^2\right), \quad (50)$$

*where*

$$Q := 2 \times \sqrt{660508}\times$$

$$\max\left\{\sqrt{\frac{L}{\alpha p\mu}}, \sqrt{\frac{L}{\alpha\tau\mu}}, \sqrt{\frac{\sqrt{LL_{\max}}(\omega+1)\sqrt{\omega\tau}}{\alpha\sqrt{n}\mu}}, \sqrt{\frac{\sqrt{LL_{\max}}\sqrt{\omega+1}\sqrt{\omega\tau}}{\alpha\sqrt{p}\sqrt{n}\mu}}, \sqrt{\frac{L_{\max}\omega(\omega+1)^2 p}{n\mu}}, \sqrt{\frac{L_{\max}\omega}{np\mu}},\right.$$

$$\left.\frac{1}{\alpha}, \frac{1}{\tau}, (\omega+1), \frac{1}{p}\right\}.$$

*Remark* E.13. Up to a constant factor of 2, one can see that the optimal $\Gamma_0$ in (50) from Theorem E.12 equals $\Gamma_0 = \bar{L}/\mu$. But the dependence on $\Gamma_0$ is under the logarithm, so if the dependence on the logarithm is not critical, one can take any $\Gamma_0 \geq 1$.

*Proof.* All conditions from Theorem E.9 are satisfied. Let us sum the inequality (45) for $t = 0$ to $T - 1$ :

$$\Gamma_T\left(\mathbb{E}\left[f(z^T) - f(x^*)\right] + \kappa\mathbb{E}\left[\frac{1}{n}\sum_{i=1}^n\left\|h_i^T - \nabla f_i(z^T)\right\|^2\right] + \nu_T\mathbb{E}\left[\left\|w^T - u^T\right\|^2\right]\right.$$

$$\left.+\rho\mathbb{E}\left[\left\|k^T - \nabla f(z^T)\right\|^2\right] + \lambda\mathbb{E}\left[\left\|v^T - \nabla f(z^T)\right\|^2\right]\right)$$

$$\leq \Gamma_0 \left( \mathbb{E}\left[ f(z^0) - f(x^*) \right] + \kappa \mathbb{E}\left[ \frac{1}{n} \sum_{i=1}^{n} \left\| h_i^0 - \nabla f_i(z^0) \right\|^2 \right] + \nu_0 \mathbb{E}\left[ \left\| w^0 - u^0 \right\|^2 \right] \right.$$

$$\left. + \rho \mathbb{E}\left[ \left\| k^0 - \nabla f(z^0) \right\|^2 \right] + \lambda \mathbb{E}\left[ \left\| v^0 - \nabla f(z^0) \right\|^2 \right] \right)$$

$$+ \left( \frac{\bar{L} + \Gamma_0 \mu}{2} \mathbb{E}\left[ \left\| u^0 - x^* \right\|^2 \right] - \frac{\bar{L} + \Gamma_T \mu}{2} \mathbb{E}\left[ \left\| u^T - x^* \right\|^2 \right] \right).$$

Using the initial conditions and the non-negativity of the terms, we get

$$\Gamma_T \mathbb{E}\left[ f(z^T) - f(x^*) \right] + \frac{\Gamma_T \mu}{2} \mathbb{E}\left[ \left\| u^T - x^* \right\|^2 \right] \leq \Gamma_0 \left( f(z^0) - f(x^*) \right) + \frac{\bar{L} + \Gamma_0 \mu}{2} \left\| u^0 - x^* \right\|^2.$$

Using Lemma E.1, we have

$$\mathbb{E}\left[ f(z^T) - f(x^*) \right] + \frac{\mu}{2} \mathbb{E}\left[ \left\| u^T - x^* \right\|^2 \right]$$

$$\leq \exp\left( -T \min\left\{ \sqrt{\frac{p\mu}{4\bar{L}}}, p\theta_{\min} \right\} \right) \left( 2 \left( f(z^0) - f(x^*) \right) + \left( \frac{\bar{L}}{\Gamma_0} + \mu \right) \left\| u^0 - x^* \right\|^2 \right).$$

It is left to use the definitions of $\bar{L}$ and $\theta_{\min}$. $\qquad\square$

## E.6 General convex case

Let us use Theorem E.9 to analyze the general convex case ($\mu$ can possibly be equal to zero):

**Theorem E.14.** *Suppose that Assumptions 1.2, 1.1, 1.3, 2.4 hold. Let*

$$\bar{L} = 660508 \times \max\left\{ \frac{L}{\alpha}, \frac{Lp}{\alpha\tau}, \frac{\sqrt{LL_{\max}}p\sqrt{\omega\tau}}{\alpha\beta\sqrt{n}}, \frac{\sqrt{LL_{\max}}\sqrt{p}\sqrt{\omega\tau}}{\alpha\sqrt{\beta}\sqrt{n}}, \frac{L_{\max}\omega p^2}{\beta^2 n}, \frac{L_{\max}\omega}{n} \right\}, \quad (51)$$

*$\beta = \frac{1}{\omega+1}$, $\theta_{\min} = \frac{1}{4} \min\left\{ 1, \frac{\alpha}{p}, \frac{\tau}{p}, \frac{\beta}{p} \right\}$, $h_i^0 = \nabla f_i(z^0)$ for all $i \in [n]$, $w^0 = u^0$, $k^0 = \nabla f(z^0)$, $v^0 = \nabla f(z^0)$, and $\Gamma_0 \in [1, \bar{L}/L]$. Then Algorithm 1 returns $\varepsilon$-solution, i.e., $\mathbb{E}\left[ f(z^T) \right] - f(x^*) \leq \varepsilon$, after*

$$T = \begin{cases} \Theta\left( \frac{1}{p\theta_{\min}} \log \frac{\bar{L}\|z^0 - x^*\|^2}{\Gamma_0 \varepsilon} \right), & \frac{\bar{L}\|z^0 - x^*\|^2}{\varepsilon} < \frac{1}{p\theta_{\min}^2} \\ \Theta\left( \max\left\{ \frac{1}{p\theta_{\min}} \log \frac{1}{\Gamma_0 p\theta_{\min}^2}, 0 \right\} + Q\sqrt{\frac{\|z^0 - x^*\|^2}{\varepsilon}} \right), & \text{otherwise} \end{cases} \quad (52)$$

*iterations, where*

$$Q := \Theta\left( \max\left\{ \sqrt{\frac{L}{\alpha p}}, \sqrt{\frac{L}{\alpha\tau}}, \sqrt{\frac{\sqrt{LL_{\max}}(\omega+1)\sqrt{\omega\tau}}{\alpha\sqrt{n}}}, \sqrt{\frac{\sqrt{LL_{\max}}\sqrt{\omega+1}\sqrt{\omega\tau}}{\alpha\sqrt{p}\sqrt{n}}}, \sqrt{\frac{L_{\max}\omega(\omega+1)^2 p}{n}}, \sqrt{\frac{L_{\max}\omega}{np}} \right\} \right).$$

*Remark* E.15. One can see that the optimal $\Gamma_0$ in (52) from Theorem E.14 equals $\Gamma_0 = \bar{L}/L$. But the dependence on $\Gamma_0$ is under the logarithm, so if the dependence on the logarithm is not critical, one can take any $\Gamma_0 \in [1, \bar{L}/L]$.

*Proof.* All conditions from Theorem E.9 are satisfied. Let us sum the inequality (45) for $t = 0$ to $T - 1$:

$$\Gamma_T \left( \mathbb{E}\left[ f(z^T) - f(x^*) \right] + \kappa \mathbb{E}\left[ \frac{1}{n} \sum_{i=1}^{n} \left\| h_i^T - \nabla f_i(z^T) \right\|^2 \right] + \nu_T \mathbb{E}\left[ \left\| w^T - u^T \right\|^2 \right] \right.$$

$$\left. + \rho \mathbb{E}\left[ \left\| k^T - \nabla f(z^T) \right\|^2 \right] + \lambda \mathbb{E}\left[ \left\| v^T - \nabla f(z^T) \right\|^2 \right] \right)$$

$$\leq \Gamma_0 \left( \mathbb{E}\left[ f(z^0) - f(x^*) \right] + \kappa \mathbb{E}\left[ \frac{1}{n} \sum_{i=1}^{n} \left\| h_i^0 - \nabla f_i(z^0) \right\|^2 \right] + \nu_0 \mathbb{E}\left[ \left\| w^0 - u^0 \right\|^2 \right] \right.$$

$$+ \rho \mathbb{E}\left[\left\|k^0 - \nabla f(z^0)\right\|^2\right] + \lambda \mathbb{E}\left[\left\|v^0 - \nabla f(z^0)\right\|^2\right]\Bigg)$$

$$+ \left(\frac{\bar{L} + \Gamma_0 \mu}{2}\mathbb{E}\left[\left\|u^0 - x^*\right\|^2\right] - \frac{\bar{L} + \Gamma_T \mu}{2}\mathbb{E}\left[\left\|u^T - x^*\right\|^2\right]\right).$$

Using the initial conditions, the non-negativity of the terms, and $\Gamma_0 \leq \bar{L}/L$, we get

$$\Gamma_T \mathbb{E}\left[f(z^T) - f(x^*)\right] \leq \Gamma_0\left(f(z^0) - f(x^*)\right) + \frac{\bar{L} + \Gamma_0 \mu}{2}\left\|u^0 - x^*\right\|^2$$

$$\leq \frac{\bar{L}}{L}\left(f(z^0) - f(x^*)\right) + \bar{L}\left\|z^0 - x^*\right\|^2.$$

Using the $L$–smoothness and $\mu \leq L \leq \bar{L}$, we have

$$\Gamma_T \mathbb{E}\left[f(z^T) - f(x^*)\right] \leq \bar{L}\left\|z^0 - x^*\right\|^2 + \bar{L}\left\|z^0 - x^*\right\|^2 \leq 2\bar{L}\left\|z^0 - x^*\right\|^2.$$

Using Lemma E.1, we have

$$\begin{cases} \frac{\Gamma_0}{2}\exp\left(tp\theta_{\min}\right)\mathbb{E}\left[f(z^T) - f(x^*)\right] \leq 2\bar{L}\left\|z^0 - x^*\right\|^2, & t < \bar{t} \\ \frac{p(t-\bar{t})^2}{16}\mathbb{E}\left[f(z^T) - f(x^*)\right] \leq 2\bar{L}\left\|z^0 - x^*\right\|^2, & t \geq \bar{t}, \end{cases}$$

where $\bar{t} := \max\left\{\left\lceil\frac{1}{p\theta_{\min}}\log\frac{1}{2\Gamma_0 p\theta_{\min}^2}\right\rceil, 0\right\}$. The last inequalities guarantees that Algorithm 1 returns $\varepsilon$-solution after (52) iterations.

$$\square$$

## E.7 Choosing optimal parameters

**Theorem 5.2.** *Choose* $r \in [0,1]$ *and let* $\mu_{\omega,\alpha}^r := \frac{rd}{(1-r)K_\omega + rK_\alpha}$. *In view of Theorem 5.1, the values* $p = \min\left\{\frac{1}{\omega+1}, \frac{1}{\mu_{\omega,\alpha}^r}\right\}$ *and* $\tau = \frac{p^{1/3}}{(\omega+1)^{2/3}}$ *minimize* $\max_{L_{\max} \in [L, nL]} \mathfrak{m}_{\text{new}}^r$. *This choice leads to the following number of communication rounds:*

$$T^{\text{realistic}} := \widetilde{\Theta}\left(\max\left\{\sqrt{\frac{L\max\{\omega+1, \mu_{\omega,\alpha}^r\}}{\alpha\mu}}, \sqrt{\frac{L_{\max}\omega\max\{\omega+1, \mu_{\omega,\alpha}^r\}}{n\mu}}, \frac{1}{\alpha}, (\omega+1), \mu_{\omega,\alpha}^r\right\}\right). \quad (13)$$

*The total communication complexity thus equals* $\mathfrak{m}_{\text{realistic}}^r = \widetilde{\Theta}\left(((1-r)K_\omega + rK_\alpha)T_{\text{realistic}} + d\right).$

*Proof.* We implicitly assume that $p \in (0,1]$ and $\tau \in (0,1]$. Using (12) and (4), we have

$$\arg\min_{p,\tau}\max_{L_{\max} \in [L,nL]}\mathfrak{m}_{\text{new}}^r = \arg\min_{p,\tau}\max_{L_{\max} \in [L,nL]}\widetilde{\Theta}\left((1-r)K_\omega T + r(K_\alpha + pd)T\right).$$

Note that only $T$ depends on $L_{\max}$ and $\tau$. We have

$$\min_{\tau}\max_{L_{\max} \in [L,nL]}T$$

$$\overset{(10)}{=}\min_{\tau}\max_{L_{\max} \in [L,nL]}\widetilde{\Theta}\left(\max\left\{\sqrt{\frac{L}{\alpha p\mu}}, \sqrt{\frac{L}{\alpha\tau\mu}}, \sqrt{\frac{\sqrt{LL_{\max}}(\omega+1)\sqrt{\omega\tau}}{\alpha\sqrt{n}\mu}},\right.\right.$$

$$\left.\left.\sqrt{\frac{\sqrt{LL_{\max}}\sqrt{\omega+1}\sqrt{\omega\tau}}{\alpha\sqrt{p}\sqrt{n}\mu}}, \sqrt{\frac{L_{\max}\omega(\omega+1)^2 p}{n\mu}}, \sqrt{\frac{L_{\max}\omega}{np\mu}}, \frac{1}{\alpha}, \frac{1}{\tau}, (\omega+1), \frac{1}{p}\right\}\right)$$

$$=\min_{\tau}\widetilde{\Theta}\left(\max\left\{\sqrt{\frac{L}{\alpha p\mu}}, \sqrt{\frac{L}{\alpha\tau\mu}}, \sqrt{\frac{L(\omega+1)\sqrt{\omega\tau}}{\alpha\mu}}, \sqrt{\frac{L\sqrt{\omega+1}\sqrt{\omega\tau}}{\alpha\sqrt{p}\mu}}, \sqrt{\frac{L\omega(\omega+1)^2 p}{\mu}}, \sqrt{\frac{L\omega}{p\mu}},\right.\right.$$

$$\left.\left.\frac{1}{\alpha}, \frac{1}{\tau}, (\omega+1), \frac{1}{p}\right\}\right)$$

The last term attains the minimum when

$$\tau = \min\left\{\frac{1}{\omega+1}, \frac{p^{1/3}}{(\omega+1)^{2/3}}\right\}. \tag{53}$$

Therefore, we get

$$T' := \min_{\tau} \max_{L_{\max}\in[L,nL]} T$$
$$= \widetilde{\Theta}\left(\max\left\{\sqrt{\frac{L}{\alpha p\mu}}, \sqrt{\frac{L(\omega+1)}{\alpha\mu}}, \sqrt{\frac{L(\omega+1)^{2/3}}{\alpha p^{1/3}\mu}}, \sqrt{\frac{L\omega(\omega+1)^2 p}{\mu}}, \sqrt{\frac{L\omega}{p\mu}}, \frac{1}{\alpha}, (\omega+1), \frac{1}{p}\right\}\right), \tag{54}$$

where we use that

$$\frac{1}{\tau} = \max\left\{\omega+1, \frac{(\omega+1)^{2/3}}{p^{1/3}}\right\} \le \max\left\{\omega+1, \frac{2}{3}(\omega+1) + \frac{1}{3p}\right\} = \Theta\left(\max\left\{(\omega+1), \frac{1}{p}\right\}\right).$$

It is left to find

$$\arg\min_{p}\left(\min_{\tau}\max_{L_{\max}\in[L,nL]}\mathfrak{m}_{\mathrm{new}}^{r}\right) = \arg\min_{p}\widetilde{\Theta}\left((1-r)K_{\omega}T' + r\left(K_{\alpha}+pd\right)T'\right)$$
$$= \arg\min_{p}\widetilde{\Theta}\left(A \times T' + B \times pT'\right), \tag{55}$$

where $A := (1-r)K_{\omega} + rK_{\alpha} \ge 0$ and $B := rd \ge 0$. Note that $A$ and $B$ do not depend on $p$. If $p \ge A/B$, then $\widetilde{\Theta}(AT' + BpT') = \widetilde{\Theta}(BpT')$. The term $pT'$ is non-decreasing function w.r.t. $p$. For $p \ge A/B$, it means that an optimal point $p = A/B$. Thus the argmin (55) is equivalent to

$$\arg\min_{p\in Q_0}\widetilde{\Theta}\left(A \times T' + B \times pT'\right) = \arg\min_{p\in Q_0}\widetilde{\Theta}\left(T'\right),$$

where $Q_0 := \left\{p\,\middle|\, p \le \frac{A}{B}\right\}$. Thus, we have

$$\arg\min_{p\in Q_0}\widetilde{\Theta}\left(\max\left\{\sqrt{\frac{L}{\alpha p\mu}}, \sqrt{\frac{L(\omega+1)}{\alpha\mu}}, \sqrt{\frac{L(\omega+1)^{2/3}}{\alpha p^{1/3}\mu}}, \sqrt{\frac{L\omega(\omega+1)^2 p}{\mu}}, \sqrt{\frac{L\omega}{p\mu}}, \frac{1}{\alpha}, (\omega+1), \frac{1}{p}\right\}\right).$$

The next observation is that this argmin is non-decreasing when $p \ge 1/\omega+1$. It means that the minimum attains at some point $p \in Q_1 := \{p\,|\,p \le 1/\omega+1, p \in Q_0\}$. Using this information, we can eliminate the redundant terms (for instance, $A\sqrt{\frac{L(\omega+1)}{\alpha\mu}} \le A\sqrt{\frac{L}{\alpha p\mu}}$ for $p \in Q_1$) and get an equivalent argmin

$$\arg\min_{p\in Q_1}\widetilde{\Theta}\left(\max\left\{\sqrt{\frac{L}{\alpha p\mu}}, \sqrt{\frac{L\omega}{p\mu}}, \frac{1}{\alpha}, \frac{1}{p}\right\}\right),$$

The last term attains the minimum at

$$p = \min\left\{\frac{1}{\omega+1}, \frac{A}{B}\right\}. \tag{56}$$

Using (56) and (53), an optimal $\tau$ is

$$\tau = \frac{p^{1/3}}{(\omega+1)^{2/3}}. \tag{57}$$

We substitute (56) and (57) to (10) and obtain (13). We use Lemma 1.4 to eliminate the redundant terms. Note that $\mu_{\omega,\alpha}^{r} := B/A$.

Using (12) and (56), we have

$$\mathfrak{m}_{\mathrm{realistic}}^{r} = (1-r)K_{\omega}T_{\mathrm{realistic}} + r\left(K_{\alpha}+pd\right)T_{\mathrm{realistic}} + d$$

$$\leq (1-r)K_\omega T_{\text{realistic}} + r\left(K_\alpha + \frac{(1-r)K_\omega + rK_\alpha}{rd}d\right)T_{\text{realistic}} + d$$

$$= 2(1-r)K_\omega T_{\text{realistic}} + 2rK_\alpha T_{\text{realistic}} + d.$$

Note that

$$\mathfrak{m}^r_{\text{realistic}} = (1-r)K_\omega T_{\text{realistic}} + r\left(K_\alpha + pd\right)T_{\text{realistic}} + d$$
$$\geq (1-r)K_\omega T_{\text{realistic}} + rK_\alpha T_{\text{realistic}} + d.$$

$\square$

**Theorem 5.4.** *Choose* $r \in [0,1]$, *and let* $\mu^r_{\omega,\alpha} := \frac{rd}{(1-r)K_\omega + rK_\alpha}$. *In view of Theorem 5.1, the values* $p$ *and* $\tau$ *given by* (63) *and* (58)*, respectively, minimize* $\mathfrak{m}^r_{\text{new}}$ *from* (10)*. This choice leads to the following number of communication rounds:*

$$T^{\text{optimistic}} = \widetilde{\Theta}\Bigg(\max\Bigg\{\sqrt{\frac{L\max\{1,\mu^r_{\omega,\alpha}\}}{\alpha\mu}}, \sqrt{\frac{L^{2/3}L^{1/3}_{\max}(\omega+1)}{\alpha n^{1/3}\mu}}, \sqrt{\frac{L^{1/2}L^{1/2}_{\max}(\omega+1)^{3/2}}{\sqrt{\alpha}n\mu}},$$
$$\sqrt{\frac{L_{\max}\omega\max\{\omega+1,\mu^r_{\omega,\alpha}\}}{n\mu}}, \frac{1}{\alpha}, (\omega+1), \mu^r_{\omega,\alpha}\Bigg\}\Bigg). \tag{14}$$

*The total communication complexity thus equals* $\mathfrak{m}^r_{\text{optimistic}} = \widetilde{\Theta}\left(((1-r)K_\omega + rK_\alpha)T_{\text{optimistic}} + d\right).$

*Proof.* We implicitly assume that $p \in (0,1]$ and $\tau \in (0,1]$. We start the proof as in Theorem 5.2. Using (12) and (4), we have

$$\arg\min_{p,\tau} \mathfrak{m}^r_{\text{new}} = \arg\min_{p,\tau} \widetilde{\Theta}\left((1-r)K_\omega T + r\left(K_\alpha + pd\right)T\right).$$

Unlike Theorem 5.2, we know the ratio $L_{\max}/L$, thus

$$\min_\tau T$$

$$\overset{(10)}{=} \min_\tau \widetilde{\Theta}\Bigg(\max\Bigg\{\sqrt{\frac{L}{\alpha p\mu}}, \sqrt{\frac{L}{\alpha\tau\mu}}, \sqrt{\frac{\sqrt{LL_{\max}}(\omega+1)\sqrt{\omega\tau}}{\alpha\sqrt{n}\mu}},$$
$$\sqrt{\frac{\sqrt{LL_{\max}}\sqrt{\omega+1}\sqrt{\omega\tau}}{\alpha\sqrt{p}\sqrt{n}\mu}}, \sqrt{\frac{L_{\max}\omega(\omega+1)^2 p}{n\mu}}, \sqrt{\frac{L_{\max}\omega}{np\mu}}, \frac{1}{\alpha}, \frac{1}{\tau}, (\omega+1), \frac{1}{p}\Bigg\}\Bigg)$$

The last term attains the minimum when

$$\tau = \min\left\{1, \left(\frac{Ln}{L_{\max}}\right)^{1/3}\min\left\{\frac{1}{\omega+1}, \frac{p^{1/3}}{(\omega+1)^{2/3}}\right\}\right\}. \tag{58}$$

Therefore, we get

$$T' := \min_\tau T$$

$$= \widetilde{\Theta}\Bigg(\max\Bigg\{\sqrt{\frac{L}{\alpha p\mu}}, \sqrt{\frac{L^{2/3}L^{1/3}_{\max}(\omega+1)}{\alpha n^{1/3}\mu}}, \sqrt{\frac{L^{2/3}L^{1/3}_{\max}(\omega+1)^{2/3}}{\alpha n^{1/3}p^{1/3}\mu}}, \sqrt{\frac{L_{\max}\omega(\omega+1)^2 p}{n\mu}}, \sqrt{\frac{L_{\max}\omega}{np\mu}}, \frac{1}{\alpha}, \omega+1, \frac{1}{p}\Bigg\}\Bigg). \tag{59}$$

It is left to find

$$\arg\min_p\left(\min_\tau \mathfrak{m}^r_{\text{new}}\right) = \arg\min_p \widetilde{\Theta}\left((1-r)K_\omega T' + r\left(K_\alpha + pd\right)T'\right)$$
$$= \arg\min_p \widetilde{\Theta}\left(A \times T' + B \times pT'\right), \tag{60}$$

where $A := (1-r)K_\omega + rK_\alpha \geq 0$ and $B := rd \geq 0$. Note that $A$ and $B$ do not depend on $p$. If $p \geq A/B$, then $\widetilde{\Theta}\left(AT' + BpT'\right) = \widetilde{\Theta}\left(BpT'\right)$. The term $pT'$ is non-decreasing function w.r.t. $p$. For $p \geq A/B$, it means that an optimal point $p = A/B$. Thus the argmin (60) is equivalent to

$$\arg\min_{p\in Q_0}\widetilde{\Theta}\left(A \times T' + B \times pT'\right) = \arg\min_{p\in Q_0}\widetilde{\Theta}\left(T'\right),$$

where $Q_0 := \left\{ p \,\middle|\, p \le \frac{A}{B} \right\}$. Next, we have

$$\underset{p \in Q_0}{\arg\min} \, \widetilde{\Theta}\left(T'\right)$$

$$= \underset{p \in Q_0}{\arg\min}$$

$$\widetilde{\Theta}\left( \max\left\{ \sqrt{\frac{L}{\alpha p \mu}}, \sqrt{\frac{L^{2/3}L_{\max}^{1/3}(\omega+1)}{\alpha n^{1/3}\mu}}, \sqrt{\frac{L^{2/3}L_{\max}^{1/3}(\omega+1)^{2/3}}{\alpha n^{1/3}p^{1/3}\mu}}, \sqrt{\frac{L_{\max}\omega(\omega+1)^2 p}{n\mu}}, \sqrt{\frac{L_{\max}\omega}{np\mu}}, \frac{1}{\alpha}, (\omega+1), \frac{1}{p} \right\} \right).$$
(61)

For $p \ge \left(\frac{Ln}{L_{\max}}\right)^{1/3}\frac{1}{\omega+1}$, using Lemma 1.4, we have $p \ge \frac{1}{\omega+1}$,

$$\sqrt{\frac{L}{\alpha p \mu}} \le \sqrt{\frac{L^{2/3}L_{\max}^{1/3}(\omega+1)}{\alpha n^{1/3}\mu}},$$

$$\sqrt{\frac{L^{2/3}L_{\max}^{1/3}(\omega+1)^{2/3}}{\alpha n^{1/3}p^{1/3}\mu}} \le \sqrt{\frac{L^{2/3}L_{\max}^{1/3}(\omega+1)}{\alpha n^{1/3}\mu}}$$

$$\sqrt{\frac{L_{\max}\omega}{np\mu}} \le \sqrt{\frac{L_{\max}\omega(\omega+1)^2 p}{n\mu}}$$

$$\frac{1}{p} \le \left(\frac{L_{\max}}{Ln}\right)^{1/3}(\omega+1) \le (\omega+1).$$

It means that for $p \ge \left(\frac{Ln}{L_{\max}}\right)^{1/3}\frac{1}{\omega+1}$, the argmin (61) is equivalent to

$$\underset{p \in Q_0}{\arg\min} \, \widetilde{\Theta}\left( \max\left\{ \sqrt{\frac{L^{2/3}L_{\max}^{1/3}(\omega+1)}{\alpha n^{1/3}\mu}}, \sqrt{\frac{L_{\max}\omega(\omega+1)^2 p}{n\mu}}, \frac{1}{\alpha}, (\omega+1) \right\} \right).$$

Since all terms are non-increasing functions of $p$, the minimum is attained at a point $p = \left(\frac{Ln}{L_{\max}}\right)^{1/3}\frac{1}{\omega+1}$ for all $p \ge \left(\frac{Ln}{L_{\max}}\right)^{1/3}\frac{1}{\omega+1}$. Let us define

$$Q_1 := \left\{ p \,\middle|\, p \le \left(\frac{Ln}{L_{\max}}\right)^{1/3}\frac{1}{\omega+1}, p \in Q_0 \right\}.$$

The last observation means that the argmin (61) is equivalent to

$$\underset{p \in Q_1}{\arg\min}$$

$$\widetilde{\Theta}\left( \max\left\{ \sqrt{\frac{L}{\alpha p \mu}}, \sqrt{\frac{L^{2/3}L_{\max}^{1/3}(\omega+1)}{\alpha n^{1/3}\mu}}, \sqrt{\frac{L^{2/3}L_{\max}^{1/3}(\omega+1)^{2/3}}{\alpha n^{1/3}p^{1/3}\mu}}, \sqrt{\frac{L_{\max}\omega(\omega+1)^2 p}{n\mu}}, \sqrt{\frac{L_{\max}\omega}{np\mu}}, \frac{1}{\alpha}, (\omega+1), \frac{1}{p} \right\} \right)$$

$$= \underset{p \in Q_1}{\arg\min} \, \widetilde{\Theta}\left( \max\left\{ \sqrt{\frac{L}{\alpha p \mu}}, \sqrt{\frac{L_{\max}\omega(\omega+1)^2 p}{n\mu}}, \sqrt{\frac{L_{\max}\omega}{np\mu}}, \frac{1}{\alpha}, (\omega+1), \frac{1}{p} \right\} \right),$$
(62)

where we eliminate the redundant terms using the additional information $p \le \left(\frac{Ln}{L_{\max}}\right)^{1/3}\frac{1}{\omega+1}$. In particular,

$$\sqrt{\frac{L^{2/3}L_{\max}^{1/3}(\omega+1)}{\alpha n^{1/3}\mu}} \le \sqrt{\frac{L}{\alpha p \mu}} \text{ and } \sqrt{\frac{L^{2/3}L_{\max}^{1/3}(\omega+1)^{2/3}}{\alpha n^{1/3}p^{1/3}\mu}} \le \sqrt{\frac{L}{\alpha p \mu}}.$$

for all $p \leq \left(\frac{Ln}{L_{\max}}\right)^{1/3} \frac{1}{\omega+1}$. Without the condition $p \in Q_1$, the argmin (62) attains the minimum at a point

$$p = \max\left\{\frac{1}{\omega+1}, \left(\frac{Ln}{L_{\max}}\right)^{1/2} \frac{1}{\sqrt{\alpha}(\omega+1)^{3/2}}\right\}.$$

Considering the condition $p \in Q_1$ and $\mu_{\omega,\alpha}^r := {}^B/_A$, we have

$$p = \min\left\{1, \frac{1}{\mu_{\omega,\alpha}^r}, \left(\frac{Ln}{L_{\max}}\right)^{1/3} \frac{1}{\omega+1}, \max\left\{\frac{1}{\omega+1}, \left(\frac{Ln}{L_{\max}}\right)^{1/2} \frac{1}{\sqrt{\alpha}(\omega+1)^{3/2}}\right\}\right\}. \quad (63)$$

It is left carefully to substitute (63) to

$$\widetilde{\Theta}\left(\max\left\{\sqrt{\frac{L}{\alpha p \mu}}, \sqrt{\frac{L_{\max}\omega}{n p \mu}}, \frac{1}{\alpha}, (\omega+1), \frac{1}{p}\right\}\right)$$

and obtain (14). The proof of $\mathfrak{m}_{\text{realistic}}^r = \widetilde{\Theta}\left(((1-r)K_\omega + rK_\alpha) T_{\text{realistic}}\right)$ is the same as in Theorem 5.2. $\qquad\square$

## E.8   Comparison with EF21 + DIANA

**Theorem 6.1.** *For all $r \in [0,1]$, $\mathfrak{m}_{\text{realistic}}^r = \widetilde{\mathcal{O}}\left(\mathfrak{m}_{\text{EF21-P + DIANA}}^r\right)$.*

*Proof.* Using the inequality of arithmetic and geometric means, i.e., $\sqrt{xy} \leq \frac{x+y}{2}$ for all $x, y \geq 0$, and $L \geq \mu$, we have

$$\mathfrak{m}_{\text{realistic}}^r = \widetilde{\Theta}\left(K_{\omega,\alpha}^r\left(\sqrt{\frac{L(\omega+1)}{\alpha\mu}} + \sqrt{\frac{L_{\max}\omega(\omega+1)}{n\mu}} + \sqrt{\frac{L\mu_{\omega,\alpha}^r}{\alpha\mu}} + \sqrt{\frac{L_{\max}\omega\mu_{\omega,\alpha}^r}{n\mu}} + \frac{1}{\alpha} + \omega + \mu_{\omega,\alpha}^r\right) + d\right)$$

$$= \widetilde{\mathcal{O}}\left(K_{\omega,\alpha}^r\left(\frac{L}{\alpha\mu} + \frac{L_{\max}\omega}{n\mu} + \omega + \sqrt{\frac{L\mu_{\omega,\alpha}^r}{\alpha\mu}} + \sqrt{\frac{L_{\max}\omega\mu_{\omega,\alpha}^r}{n\mu}} + \mu_{\omega,\alpha}^r\right) + d\right).$$

From the definition of $\mu_{\omega,\alpha}^r := {}^{rd}/_{K_{\omega,\alpha}^r}$, we get

$$\mathfrak{m}_{\text{realistic}}^r = \widetilde{\mathcal{O}}\left(K_{\omega,\alpha}^r\left(\frac{L}{\alpha\mu} + \frac{L_{\max}\omega}{n\mu} + \omega\right) + \left(\sqrt{\frac{LK_{\omega,\alpha}^r \times rd}{\alpha\mu}} + \sqrt{\frac{L_{\max}\omega K_{\omega,\alpha}^r \times rd}{n\mu}} + rd\right) + d\right).$$

Using the inequality of arithmetic and geometric means again and $r \leq 1$, we obtain

$$\mathfrak{m}_{\text{realistic}}^r = \widetilde{\mathcal{O}}\left(K_{\omega,\alpha}^r\left(\frac{L}{\alpha\mu} + \frac{L_{\max}\omega}{n\mu} + \omega\right) + K_{\omega,\alpha}^r\left(\frac{L}{\alpha\mu} + \frac{L_{\max}\omega}{n\mu}\right) + d\right).$$

The last equality means that $\mathfrak{m}_{\text{realistic}}^r = \widetilde{\mathcal{O}}\left(\mathfrak{m}_{\text{EF21-P + DIANA}}^r\right)$ for all $r \in [0,1]$. $\qquad\square$

## E.9   Comparison with AGD

**Theorem 6.2.** *For all $r \in [0,1]$ and for all $K \in [d]$, let us take the RandK and TopK compressors with the parameters (expected densities) i) $K_\omega = K$ and $K_\alpha = \min\{\lceil {}^{1-r}/_r K\rceil, d\}$ for $r \in [0, {}^1/_2]$, ii) $K_\omega = \min\{\lceil {}^r/_{1-r} K\rceil, d\}$ and $K_\alpha = K$ for $r \in ({}^1/_2, 1]$. Then we have $\mathfrak{m}_{\text{realistic}}^r = \widetilde{\mathcal{O}}\left(\mathfrak{m}_{\text{AGD}}\right)$.*

*Proof.* Consider that $r \in [0, 1/2]$, then $K_\omega = K$, $K_\alpha = \min\{\lceil {}^{1-r}/_r K\rceil, d\}$. Therefore, we have

$$K_{\omega,\alpha}^r := (1-r)K_\omega + rK_\alpha \leq (1-r)K + r\lceil {}^{1-r}/_r K\rceil \leq 3(1-r)K$$

and

$$K_{\omega,\alpha}^r := (1-r)K_\omega + rK_\alpha \geq (1-r)K.$$

Using this observation, we obtain $\mu_{\omega,\alpha}^r := \frac{rd}{K_{\omega,\alpha}^r} \le \frac{rd}{(1-r)K} \le \frac{d}{K}$. Note that $\alpha \ge \sqrt[K_\alpha]{d}$ and $\omega \le$ $\sqrt[d]{K_\omega} - 1$ for $\mathrm{Top}K$ and $\mathrm{Rand}K$. Thus $\alpha \ge \min\left\{\frac{(1-r)K}{rd}, 1\right\}$ and $\omega \le \frac{d}{K} - 1$. We substitute the bounds to (16) and obtain

$$
\mathfrak{m}_{\text{realistic}}^r = \widetilde{\mathcal{O}}\left((1-r)K\left(\sqrt{\frac{d}{K}}\sqrt{\frac{L}{\mu}} + \frac{d}{K}\sqrt{\frac{rL}{(1-r)\mu}} + \frac{d}{K}\sqrt{\frac{L_{\max}}{n\mu}} + \frac{rd}{(1-r)K} + \frac{d}{K} + \frac{rd}{(1-r)K}\right) + d\right).
$$

Since $r \in [0, 1/2]$ and $K \le d$, one can easily show that

$$
\mathfrak{m}_{\text{realistic}}^r = \widetilde{\mathcal{O}}\left(d\sqrt{\frac{L}{\mu}} + d\sqrt{\frac{L_{\max}}{n\mu}} + d\right).
$$

It is left to use Lemma 1.4, to get $\mathfrak{m}_{\text{realistic}}^r = \widetilde{\mathcal{O}}\left(d\sqrt{\frac{L}{\mu}}\right) = \widetilde{\mathcal{O}}\left(\mathfrak{m}_{\text{AGD}}\right)$ for all $r \in [0, 1/2]$.

Assume that $r \in (1/2, 1]$, then $K_\omega = \min\{\lceil r/1-rK\rceil, d\}$ and $K_\alpha = K$. Using the same reasoning, we have

$$
K_{\omega,\alpha}^r := (1-r)K_\omega + rK_\alpha \le (1-r)\lceil r/1-rK\rceil + rK \le 3rK,
$$

$$
K_{\omega,\alpha}^r := (1-r)K_\omega + rK_\alpha \ge rK,
$$

$$
\mu_{\omega,\alpha}^r := \frac{rd}{K_{\omega,\alpha}^r} \le \frac{d}{K},
$$

$$
\alpha \ge \frac{K}{d} \quad \text{and} \quad \omega \le \max\left\{\frac{(1-r)d}{rK}, 1\right\} - 1.
$$

By substituting these inequalities to (16), we obtain

$$
\mathfrak{m}_{\text{realistic}}^r = \widetilde{\mathcal{O}}\left(rK\left(\frac{d}{K}\sqrt{\frac{L}{\mu}} + \frac{d}{K}\sqrt{\frac{(1-r)L_{\max}\omega}{rn\mu}} + \frac{d}{K} + \frac{(1-r)d}{rK} + \frac{d}{K}\right) + d\right)
$$

Using Lemma 1.4, one can easily show that $\mathfrak{m}_{\text{realistic}}^r = \widetilde{\mathcal{O}}\left(d\sqrt{\frac{L}{\mu}}\right) = \widetilde{\mathcal{O}}\left(\mathfrak{m}_{\text{AGD}}\right)$ for all $r \in (1/2, 1]$.
$\qquad\qquad\qquad\qquad\qquad\qquad\qquad\qquad\qquad\qquad\qquad\qquad\qquad\qquad\qquad\qquad\qquad\quad\square$

## F  Auxillary Inequalities For $\bar{L}$

We now prove useful bounds for $\bar{L}$.

**Lemma F.1** (Auxillary Inequalities). *Assume that the constraint* (44) *hold, and a constant* $c = 660508$. *Then*

$$\bar{L} \ge c\frac{L_{\max}\omega p^2}{\beta^2 n} \quad (64) \qquad\qquad \bar{L} \ge c\frac{L_{\max}\omega p}{\beta n} \quad (65) \qquad\qquad \bar{L} \ge c\frac{L_{\max}\omega}{n} \quad (66)$$

$$\bar{L} \ge c\frac{\sqrt{LL_{\max}}p\sqrt{\omega\tau}}{\alpha\beta\sqrt{n}} \quad (67) \qquad \bar{L} \ge c\frac{\sqrt{LL_{\max}}\sqrt{p}\sqrt{\omega\tau}}{\alpha\sqrt{\beta}\sqrt{n}} \quad (68) \qquad \bar{L} \ge c\frac{\sqrt{LL_{\max}}\sqrt{p}\sqrt{\omega\tau}}{\alpha\sqrt{n}} \quad (69)$$

$$\bar{L} \ge c\frac{\widehat{L}p\sqrt{\omega\tau}}{\alpha\beta\sqrt{n}} \quad (70) \qquad\qquad \bar{L} \ge c\frac{\widehat{L}\sqrt{p}\sqrt{\omega\tau}}{\alpha\sqrt{\beta}\sqrt{n}} \quad (71) \qquad\qquad \bar{L} \ge c\frac{\widehat{L}\sqrt{p}\sqrt{\omega\tau}}{\alpha\sqrt{n}} \quad (72)$$

$$\bar{L} \ge c\frac{\widehat{L}p\sqrt{\omega}}{\sqrt{\alpha}\beta\sqrt{n}} \quad (73) \qquad\qquad \bar{L} \ge c\frac{\widehat{L}\sqrt{p}\sqrt{\omega}}{\sqrt{\alpha}\sqrt{\beta}\sqrt{n}} \quad (74)$$

$$\bar{L} \geq c\frac{\widehat{L}p\sqrt{\omega}}{\beta\sqrt{n}} \qquad (75) \qquad\qquad \bar{L} \geq c\frac{\widehat{L}\sqrt{p\omega}}{\sqrt{\beta n}} \qquad (76)$$

$$\bar{L} \geq c\frac{L}{\alpha} \qquad (77) \qquad\qquad \bar{L} \geq c\frac{Lp}{\alpha\tau} \qquad (78) \qquad\qquad \bar{L} \geq cL \qquad (79)$$

$$\bar{L} \geq c\left(\frac{L\widehat{L}^2\omega p^4}{\alpha^2\beta^2 n\tau^2}\right)^{1/3} \qquad (80) \qquad \bar{L} \geq c\left(\frac{L\widehat{L}^2\omega p^3}{\alpha^2\beta n\tau^2}\right)^{1/3} \qquad (81)$$

*Proof.* The inequalities (64) and (66) follow from (44). The inequality (65) follows from (64) and (66):

$$c\frac{L_{\max}\omega p}{\beta n} \leq c\frac{L_{\max}\omega}{n}\left(\frac{1}{2}\times\frac{p^2}{\beta^2} + \frac{1}{2}\times 1^2\right) = \frac{c}{2}\times\frac{L_{\max}\omega p^2}{\beta^2 n} + \frac{c}{2}\times\frac{L_{\max}\omega}{n} \leq \bar{L}.$$

The inequalities (67) and (68) follow from (44). The inequality (69) follows from (68) and $\beta \in (0,1]$:

$$\bar{L} \geq c\frac{\sqrt{LL_{\max}}\sqrt{p}\sqrt{\omega\tau}}{\alpha\sqrt{\beta}\sqrt{n}} \geq c\frac{\sqrt{LL_{\max}}\sqrt{p}\sqrt{\omega\tau}}{\alpha\sqrt{n}}.$$

Using Lemma 1.4, (67), (68), and (69), the inequalities (70), (71), and (72) follow from

$$\bar{L} \geq c\frac{\sqrt{LL_{\max}}p\sqrt{\omega\tau}}{\alpha\beta\sqrt{n}} \geq c\frac{\widehat{L}p\sqrt{\omega\tau}}{\alpha\beta\sqrt{n}},$$

$$\bar{L} \geq c\frac{\sqrt{LL_{\max}}\sqrt{p}\sqrt{\omega\tau}}{\alpha\sqrt{\beta}\sqrt{n}} \geq c\frac{\widehat{L}\sqrt{p}\sqrt{\omega\tau}}{\alpha\sqrt{\beta}\sqrt{n}},$$

$$\bar{L} \geq c\frac{\sqrt{LL_{\max}}\sqrt{p}\sqrt{\omega\tau}}{\alpha\sqrt{n}} \geq c\frac{\widehat{L}\sqrt{p}\sqrt{\omega\tau}}{\alpha\sqrt{n}}.$$

Next, using Lemma 1.4, and $\frac{x+y}{2} \geq \sqrt{xy}$ for all $x, y \geq 0$, the inequality (73) follows from

$$c\frac{\widehat{L}p\sqrt{\omega}}{\sqrt{\alpha}\beta\sqrt{n}} \leq c\frac{\sqrt{LL_{\max}}p\sqrt{\omega}}{\sqrt{\alpha}\beta\sqrt{n}} \leq \frac{c}{2}\times\frac{L}{\alpha} + \frac{c}{2}\times\frac{L_{\max}p^2\omega}{\beta^2 n} \overset{(44)}{\leq} \bar{L}.$$

The inequality (74) follows from

$$c\frac{\widehat{L}\sqrt{p}\sqrt{\omega}}{\sqrt{\alpha}\sqrt{\beta}\sqrt{n}} \leq c\frac{\sqrt{LL_{\max}}\sqrt{p}\sqrt{\omega}}{\sqrt{\alpha}\sqrt{\beta}\sqrt{n}} \leq \frac{c}{2}\times\frac{L}{\alpha} + \frac{c}{2}\times\frac{L_{\max}p\omega}{\beta n} \overset{(44),(65)}{\leq} \bar{L}.$$

The inequalities (75) and (76) follow from (73), (74), and $\alpha \in (0,1]$ :

$$\bar{L} \geq c\frac{\widehat{L}p\sqrt{\omega}}{\sqrt{\alpha}\beta\sqrt{n}} \geq c\frac{\widehat{L}p\sqrt{\omega}}{\beta\sqrt{n}},$$

$$\bar{L} \geq c\frac{\widehat{L}\sqrt{p}\sqrt{\omega}}{\sqrt{\alpha}\sqrt{\beta}\sqrt{n}} \geq c\frac{\widehat{L}\sqrt{p}\sqrt{\omega}}{\sqrt{\beta}\sqrt{n}}.$$

The inequalities (77) and (78) follow from (44), and (79) follows from (77) and $\alpha \in (0,1]$. Using Lemma 1.4, and $\frac{x+y+z}{3} \geq (xyz)^{1/3}$ for all $x, y, z \geq 0$, the inequalities (80) and (81) follows from

$$c\left(\frac{L\widehat{L}^2\omega p^4}{\alpha^2\beta^2 n\tau^2}\right)^{1/3} \leq c\left(\frac{L^2 L_{\max}\omega p^4}{\alpha^2\beta^2 n\tau^2}\right)^{1/3} \leq \frac{c}{3}\times\frac{Lp}{\alpha\tau} + \frac{c}{3}\times\frac{Lp}{\alpha\tau} + \frac{c}{3}\times\frac{L_{\max}\omega p^2}{\beta^2 n} \overset{(44)}{\leq} \bar{L},$$

$$c\left(\frac{L\widehat{L}^2\omega p^3}{\alpha^2\beta n\tau^2}\right)^{1/3} \leq c\left(\frac{L^2 L_{\max}\omega p^3}{\alpha^2\beta n\tau^2}\right)^{1/3} \leq \frac{c}{3}\times\frac{Lp}{\alpha\tau} + \frac{c}{3}\times\frac{Lp}{\alpha\tau} + \frac{c}{3}\times\frac{L_{\max}\omega p}{\beta n} \overset{(44),(65)}{\leq} \bar{L}.$$

$\square$

# G    Proof of Lemma E.10 (First Symbolically Computed)

We use the notations from the proof of Theorem E.9.

**Lemma E.10** (First Symbolically Computed). *Assume that for the parameter $\bar{L}$, the inequalities from Sections I and J hold. Then, for all $t \geq 0$, exists $\rho$ in (90), $\kappa$ in (88), $\lambda$ in (82), and $\nu_t$ in (83) such that (46) holds.*

*Proof.* The inequalities (46) are equivalent to

$$\frac{8\theta_{t+1}^2}{\alpha}\left(\frac{pL}{2} + \kappa 4p\left(1 + \frac{p}{\beta}\right)\widehat{L}^2 + \rho 4pL^2 + \lambda\left(2p\left(1 + \frac{2p}{\tau}\right)L^2 + \frac{4p\tau^2\omega\widehat{L}^2}{n}\right)\right) \leq \nu_t,$$

$$\nu_t\frac{8}{\alpha p}\left(\frac{\gamma_{t+1}}{\bar{L} + \Gamma_{t+1}\mu}\right)^2 \leq \rho,$$

$$\rho\frac{8p}{\tau} \leq \lambda,$$

$$\frac{8p\omega}{n\bar{L}\beta} + \nu_t\frac{8\omega}{n\beta}\left(\frac{\gamma_{t+1}}{\bar{L} + \Gamma_{t+1}\mu}\right)^2 + \lambda\frac{8\tau^2\omega}{n\beta} \leq \kappa.$$

Let us take

$$\lambda := \rho\frac{8p}{\tau} \tag{82}$$

to ensure that the third inequality holds. It left to find the parameters such that

$$\frac{8\theta_{t+1}^2}{\alpha}\left(\frac{pL}{2} + \kappa 4p\left(1 + \frac{p}{\beta}\right)\widehat{L}^2 + \rho 4pL^2 + \rho\frac{8p}{\tau}\left(2p\left(1 + \frac{2p}{\tau}\right)L^2 + \frac{4p\tau^2\omega\widehat{L}^2}{n}\right)\right) \leq \nu_t,$$

$$\nu_t\frac{8}{\alpha p}\left(\frac{\gamma_{t+1}}{\bar{L} + \Gamma_{t+1}\mu}\right)^2 \leq \rho,$$

$$\frac{8p\omega}{n\bar{L}\beta} + \nu_t\frac{8\omega}{n\beta}\left(\frac{\gamma_{t+1}}{\bar{L} + \Gamma_{t+1}\mu}\right)^2 + \rho\frac{8p}{\tau}\cdot\frac{8\tau^2\omega}{n\beta} \leq \kappa.$$

Let us take

$$\nu_t := \theta_{t+1}^2\widehat{\nu}(\kappa, \rho), \tag{83}$$

where we additionally define

$$\widehat{\nu} \equiv \widehat{\nu}(\kappa, \rho) := \frac{8}{\alpha}\left(\frac{pL}{2} + \kappa 4p\left(1 + \frac{p}{\beta}\right)\widehat{L}^2 + \rho 4pL^2 + \rho\frac{8p}{\tau}\left(2p\left(1 + \frac{2p}{\tau}\right)L^2 + \frac{4p\tau^2\omega\widehat{L}^2}{n}\right)\right), \tag{84}$$

to ensure that the first inequality holds. It left to find the parameters $\kappa$ and $\rho$ such that

$$\widehat{\nu}(\kappa, \rho)\frac{8}{\alpha p}\left(\frac{\gamma_{t+1}\theta_{t+1}}{\bar{L} + \Gamma_{t+1}\mu}\right)^2 \leq \rho,$$

$$\frac{8p\omega}{n\bar{L}\beta} + \widehat{\nu}(\kappa, \rho)\frac{8\omega}{n\beta}\left(\frac{\gamma_{t+1}\theta_{t+1}}{\bar{L} + \Gamma_{t+1}\mu}\right)^2 + \rho\frac{8p}{\tau}\cdot\frac{8\tau^2\omega}{n\beta} \leq \kappa.$$

Using Lemma E.1, we have $\frac{\gamma_{t+1}\theta_{t+1}}{\bar{L} + \Gamma_{t+1}\mu} \leq \frac{\gamma_{t+1}\theta_{t+1}}{\bar{L} + \Gamma_t\mu} \leq \frac{1}{\bar{L}}$, so it is sufficient to show that stronger inequalities hold:

$$\widehat{\nu}(\kappa, \rho)\frac{8}{\alpha p\bar{L}^2} \leq \rho, \tag{85}$$

$$\frac{8p\omega}{n\bar{L}\beta} + \widehat{\nu}(\kappa, \rho)\frac{8\omega}{n\beta\bar{L}^2} + \rho\frac{8p}{\tau}\cdot\frac{8\tau^2\omega}{n\beta} \leq \kappa. \tag{86}$$

From this point all formulas in this lemma are generated by the script from Section P (see Section 4 in Section P). We use the SymPy library (Meurer et al., 2017).

Using the definition of $\widehat{\nu}$, the left hand side of (86) equals

$$
\begin{aligned}
& \frac{2048L^2\omega p^3\rho}{\bar{L}^2\alpha\beta n\tau^2} + \frac{1024L^2\omega p^2\rho}{\bar{L}^2\alpha\beta n\tau} + \frac{256L^2\omega p\rho}{\bar{L}^2\alpha\beta n} + \frac{32L\omega p}{\bar{L}^2\alpha\beta n} \\
& + \kappa\left(\frac{256\hat{L}^2\omega p}{\bar{L}^2\alpha\beta n} + \frac{256\hat{L}^2\omega p^2}{\bar{L}^2\alpha\beta^2 n}\right) + \frac{64\omega p\rho\tau}{\beta n} + \frac{8\omega p}{\bar{L}\beta n} + \frac{2048\hat{L}^2\omega^2 p^2\rho\tau}{\bar{L}^2\alpha\beta n^2},
\end{aligned}
\tag{87}
$$

where we grouped the terms w.r.t. $\kappa$. Let us take $\bar{L}$ such that the bracket is less or equal to $1/2$. We define the constraints in Section I. Therefore, (86) holds if

$$
\kappa := \frac{4096L^2\omega p^3\rho}{\bar{L}^2\alpha\beta n\tau^2} + \frac{2048L^2\omega p^2\rho}{\bar{L}^2\alpha\beta n\tau} + \frac{512L^2\omega p\rho}{\bar{L}^2\alpha\beta n} + \frac{64L\omega p}{\bar{L}^2\alpha\beta n} + \frac{128\omega p\rho\tau}{\beta n} + \frac{16\omega p}{\bar{L}\beta n} + \frac{4096\hat{L}^2\omega^2 p^2\rho\tau}{\bar{L}^2\alpha\beta n^2}.
\tag{88}
$$

Using the definition of $\widehat{\nu}$ and $\kappa$, the left hand side of (85) equals

$$
\begin{aligned}
& \frac{32L}{\bar{L}^2\alpha^2} + \frac{16384L\hat{L}^2\omega p}{\bar{L}^4\alpha^3\beta n} + \frac{16384L\hat{L}^2\omega p^2}{\bar{L}^4\alpha^3\beta^2 n} \\
& + \rho\left(\frac{2048L^2 p^2}{\bar{L}^2\alpha^2\tau^2} + \frac{1024L^2 p}{\bar{L}^2\alpha^2\tau} + \frac{256L^2}{\bar{L}^2\alpha^2} + \frac{1048576L^2\hat{L}^2\omega p^3}{\bar{L}^4\alpha^3\beta n\tau^2} + \frac{524288L^2\hat{L}^2\omega p^2}{\bar{L}^4\alpha^3\beta n\tau}\right. \\
& + \frac{131072L^2\hat{L}^2\omega p}{\bar{L}^4\alpha^3\beta n} + \frac{1048576L^2\hat{L}^2\omega p^4}{\bar{L}^4\alpha^3\beta^2 n\tau^2} + \frac{524288L^2\hat{L}^2\omega p^3}{\bar{L}^4\alpha^3\beta^2 n\tau} \\
& + \frac{131072L^2\hat{L}^2\omega p^2}{\bar{L}^4\alpha^3\beta^2 n} + \frac{2048\hat{L}^2\omega p\tau}{\bar{L}^2\alpha^2 n} + \frac{32768\hat{L}^2\omega p\tau}{\bar{L}^2\alpha^2\beta n} + \frac{32768\hat{L}^2\omega p^2\tau}{\bar{L}^2\alpha^2\beta^2 n} \\
& \left. + \frac{1048576\hat{L}^4\omega^2 p^2\tau}{\bar{L}^4\alpha^3\beta n^2} + \frac{1048576\hat{L}^4\omega^2 p^3\tau}{\bar{L}^4\alpha^3\beta^2 n^2}\right) + \frac{4096\hat{L}^2\omega p}{\bar{L}^3\alpha^2\beta n} + \frac{4096\hat{L}^2\omega p^2}{\bar{L}^3\alpha^2\beta^2 n},
\end{aligned}
\tag{89}
$$

where we grouped the terms w.r.t. $\rho$. Let us take $\bar{L}$ such that the bracket is less or equal to $1/2$. We define the constraints in Section J. Therefore, (85) holds if

$$
\rho := \frac{64L}{\bar{L}^2\alpha^2} + \frac{32768L\hat{L}^2\omega p}{\bar{L}^4\alpha^3\beta n} + \frac{32768L\hat{L}^2\omega p^2}{\bar{L}^4\alpha^3\beta^2 n} + \frac{8192\hat{L}^2\omega p}{\bar{L}^3\alpha^2\beta n} + \frac{8192\hat{L}^2\omega p^2}{\bar{L}^3\alpha^2\beta^2 n}.
\tag{90}
$$

Finally, under the constraints from Sections I and J on $\bar{L}$, the choices of parameters (90), (88), (83) and (82) insure that (46) holds. $\qquad\square$

# H   Proof of Lemma E.11 (Second Symbolically Computed)

We use the notations from the proof of Theorem E.9.

**Lemma E.11** (Second Symbolically Computed). *Consider the parameters $\rho$, $\kappa$, $\lambda$, and $\nu_t$ from Lemma E.10. Assume that for the parameter $\bar{L}$, the inequalities from Sections L and N hold, and the step size $\theta_{t+1} \leq 1/4$ for all $t \geq 0$. Then, for all $t \geq 0$, the following inequalities are satisfied:*

$$
\begin{aligned}
p\left(\frac{4\omega L_{\max}}{n\bar{L}} + \kappa 8\left(1 + \frac{p}{\beta}\right)L_{\max} + \nu_t\left(\frac{\gamma_{t+1}}{\bar{L} + \Gamma_{t+1}\mu}\right)^2\left(\frac{4\omega L_{\max}}{pn} + \frac{8L}{p\alpha}\right) + \rho 8L + \\
+ \lambda\left(4\left(1 + \frac{2p}{\tau}\right)L + \frac{8\tau^2\omega L_{\max}}{n}\right) + \theta_{t+1} - 1\right)D_f(z^t, y^{t+1}) \leq 0
\end{aligned}
\tag{47}
$$

*and*

$$2\theta_{t+1}^2 \left( \frac{pL}{2} + \kappa 4p \left(1 + \frac{p}{\beta}\right) \widehat{L}^2 + \rho 4pL^2 + \lambda \left(2p\left(1 + \frac{2p}{\tau}\right) L^2 + \frac{4p\tau^2\omega\widehat{L}^2}{n}\right) \right) \mathbb{E}_t \left[ \left\| u^{t+1} - u^t \right\|^2 \right]$$

$$- \frac{p\theta_{t+1}^2 \bar{L}}{2} \mathbb{E}_t \left[ \left\| u^{t+1} - u^t \right\|^2 \right] \leq 0.$$

$$(48)$$

*Proof.* Since $p \geq 0$ and $D_f(z^t, y^{t+1}) \geq 0$ for all $t \geq 0$, the inequality (47) is satisfied if

$$\frac{4\omega L_{\max}}{n\bar{L}} + \kappa 8\left(1 + \frac{p}{\beta}\right) L_{\max} + \nu_t \left(\frac{\gamma_{t+1}}{\bar{L} + \Gamma_{t+1}\mu}\right)^2 \left(\frac{4\omega L_{\max}}{pn} + \frac{8L}{p\alpha}\right) + \rho 8L +$$

$$+ \lambda \left(4\left(1 + \frac{2p}{\tau}\right) L + \frac{8\tau^2\omega L_{\max}}{n}\right) + \theta_{t+1} - 1 \leq 0.$$

Note that $\theta_{t+1} \leq \frac{1}{4}$ for all $t \geq 0$. Therefore, it is sufficient to show that

$$\frac{4\omega L_{\max}}{n\bar{L}} + \kappa 8\left(1 + \frac{p}{\beta}\right) L_{\max} + \nu_t \left(\frac{\gamma_{t+1}}{\bar{L} + \Gamma_{t+1}\mu}\right)^2 \left(\frac{4\omega L_{\max}}{pn} + \frac{8L}{p\alpha}\right) + \rho 8L +$$

$$+ \lambda \left(4\left(1 + \frac{2p}{\tau}\right) L + \frac{8\tau^2\omega L_{\max}}{n}\right) \leq \frac{3}{4}.$$

In the view of (83), we have to show that

$$\frac{4\omega L_{\max}}{n\bar{L}} + \kappa 8\left(1 + \frac{p}{\beta}\right) L_{\max} + \widehat{\nu} \left(\frac{\gamma_{t+1}\theta_{t+1}}{\bar{L} + \Gamma_{t+1}\mu}\right)^2 \left(\frac{4\omega L_{\max}}{pn} + \frac{8L}{p\alpha}\right) + \rho 8L +$$

$$+ \lambda \left(4\left(1 + \frac{2p}{\tau}\right) L + \frac{8\tau^2\omega L_{\max}}{n}\right) \leq \frac{3}{4}.$$

Using Lemma E.1, we have $\frac{\gamma_{t+1}\theta_{t+1}}{\bar{L} + \Gamma_{t+1}\mu} \leq \frac{\gamma_{t+1}\theta_{t+1}}{\bar{L} + \Gamma_t\mu} \leq \frac{1}{\bar{L}}$, so it is sufficient to show that

$$\frac{4\omega L_{\max}}{n\bar{L}} + \kappa 8\left(1 + \frac{p}{\beta}\right) L_{\max} + \frac{\widehat{\nu}}{\bar{L}^2}\left(\frac{4\omega L_{\max}}{pn} + \frac{8L}{p\alpha}\right) + \rho 8L +$$

$$+ \lambda \left(4\left(1 + \frac{2p}{\tau}\right) L + \frac{8\tau^2\omega L_{\max}}{n}\right) \leq \frac{3}{4}.$$

$$(91)$$

From this point all formulas in this lemma are generated by the script from Section P (see Section 5 in Section P).

Let us substitute (90), (88), (82), and (84) to the last inequality and obtain the inequality from Section K. The conditions from Section L insure that the inequality from Section K holds. It left to prove (48). Since $p \geq 0$, $\mathbb{E}_t \left[ \left\| u^{t+1} - u^t \right\|^2 \right] \geq 0$ and $\theta_{t+1}^2 \geq 0$ for all $t \geq 0$, the inequality (48) holds if

$$\frac{4}{\bar{L}}\left(\frac{L}{2} + \kappa 4\left(1 + \frac{p}{\beta}\right) \widehat{L}^2 + \rho 4L^2 + \lambda \left(2\left(1 + \frac{2p}{\tau}\right) L^2 + \frac{4\tau^2\omega\widehat{L}^2}{n}\right) \right) \leq 1.$$

$$(92)$$

Let us substitute (90), (88) and (82) to the last inequality and obtain the inequality from Section M. The inequality from Section M holds if $\bar{L}$ satisfy the inequalities from Section N. $\qquad \square$

# I Symbolically Computed Constraints for $\bar{L}$ Such That The Term w.r.t. $\kappa$ is less or equal $1/2$ in (87)

$$\frac{256\hat{L}^2\omega p}{\bar{L}^2\alpha\beta n} \leq \frac{1}{4} \qquad (93) \qquad\qquad \frac{256\hat{L}^2\omega p^2}{\bar{L}^2\alpha\beta^2 n} \leq \frac{1}{4} \qquad (94)$$

## J Symbolically Computed Constraints for $\bar{L}$ Such That The Term w.r.t. $\rho$ is less or equal $^1/_2$ in (89)

$$\frac{256L^2}{\bar{L}^2\alpha^2} \le \frac{1}{28} \qquad (95)$$

$$\frac{1024L^2p}{\bar{L}^2\alpha^2\tau} \le \frac{1}{28} \qquad (96)$$

$$\frac{2048L^2p^2}{\bar{L}^2\alpha^2\tau^2} \le \frac{1}{28} \qquad (97)$$

$$\frac{2048\hat{L}^2\omega p\tau}{\bar{L}^2\alpha^2 n} \le \frac{1}{28} \qquad (98)$$

$$\frac{32768\hat{L}^2\omega p\tau}{\bar{L}^2\alpha^2\beta n} \le \frac{1}{28} \qquad (99)$$

$$\frac{32768\hat{L}^2\omega p^2\tau}{\bar{L}^2\alpha^2\beta^2 n} \le \frac{1}{28} \qquad (100)$$

$$\frac{131072L^2\hat{L}^2\omega p}{\bar{L}^4\alpha^3\beta n} \le \frac{1}{28} \qquad (101)$$

$$\frac{131072L^2\hat{L}^2\omega p^2}{\bar{L}^4\alpha^3\beta^2 n} \le \frac{1}{28} \qquad (102)$$

$$\frac{1048576\hat{L}^4\omega^2 p^2\tau}{\bar{L}^4\alpha^3\beta n^2} \le \frac{1}{28} \qquad (103)$$

$$\frac{1048576\hat{L}^4\omega^2 p^3\tau}{\bar{L}^4\alpha^3\beta^2 n^2} \le \frac{1}{28} \qquad (104)$$

$$\frac{524288L^2\hat{L}^2\omega p^2}{\bar{L}^4\alpha^3\beta n\tau} \le \frac{1}{28} \qquad (105)$$

$$\frac{524288L^2\hat{L}^2\omega p^3}{\bar{L}^4\alpha^3\beta^2 n\tau} \le \frac{1}{28} \qquad (106)$$

$$\frac{1048576L^2\hat{L}^2\omega p^3}{\bar{L}^4\alpha^3\beta n\tau^2} \le \frac{1}{28} \qquad (107)$$

$$\frac{1048576L^2\hat{L}^2\omega p^4}{\bar{L}^4\alpha^3\beta^2 n\tau^2} \le \frac{1}{28} \qquad (108)$$

## K Symbolically Computed Expression (91)

$$\frac{544L^2}{\bar{L}^2\alpha^2} + \frac{16384L^4}{\bar{L}^4\alpha^4} + \frac{4L_{\max}\omega}{\bar{L}n}$$
$$+ \frac{2048L^2p}{\bar{L}^2\alpha^2\tau} + \frac{4096L^2p^2}{\bar{L}^2\alpha^2\tau^2} + \frac{65536L^4p}{\bar{L}^4\alpha^4\tau}$$
$$+ \frac{131072L^4p^2}{\bar{L}^4\alpha^4\tau^2} + \frac{16LL_{\max}\omega}{\bar{L}^2\alpha n} + \frac{128L_{\max}\omega p}{\bar{L}\beta n}$$
$$+ \frac{128L_{\max}\omega p^2}{\bar{L}\beta^2 n} + \frac{8192L^3L_{\max}\omega}{\bar{L}^4\alpha^3 n} + \frac{512LL_{\max}\omega p}{\bar{L}^2\alpha\beta n}$$
$$+ \frac{512LL_{\max}\omega p^2}{\bar{L}^2\alpha\beta^2 n} + \frac{2048L_{\max}\hat{L}^2\omega^2 p}{\bar{L}^3\alpha\beta n^2} + \frac{2048L_{\max}\hat{L}^2\omega^2 p^2}{\bar{L}^3\alpha\beta^2 n^2}$$
$$+ \frac{4096LL_{\max}\omega p\tau}{\bar{L}^2\alpha^2 n} + \frac{32768L^3L_{\max}\omega p}{\bar{L}^4\alpha^3 n\tau} + \frac{65536L^3L_{\max}\omega p^2}{\bar{L}^4\alpha^3 n\tau^2}$$
$$+ \frac{69632L\hat{L}^2\omega p}{\bar{L}^3\alpha^2\beta n} + \frac{69632L\hat{L}^2\omega p^2}{\bar{L}^3\alpha^2\beta^2 n} + \frac{131072L^2\hat{L}^2\omega p\tau}{\bar{L}^4\alpha^4 n}$$
$$+ \frac{262144L^3L_{\max}\omega p}{\bar{L}^4\alpha^3\beta n} + \frac{262144L^3L_{\max}\omega p^2}{\bar{L}^4\alpha^3\beta^2 n} + \frac{278528L^2\hat{L}^2\omega p}{\bar{L}^4\alpha^3\beta n}$$
$$+ \frac{278528L^2\hat{L}^2\omega p^2}{\bar{L}^4\alpha^3\beta^2 n} + \frac{2097152L^3\hat{L}^2\omega p}{\bar{L}^5\alpha^4\beta n} + \frac{2097152L^3\hat{L}^2\omega p^2}{\bar{L}^5\alpha^4\beta^2 n}$$
$$+ \frac{16777216L^4\hat{L}^2\omega p}{\bar{L}^6\alpha^5\beta n} + \frac{16777216L^4\hat{L}^2\omega p^2}{\bar{L}^6\alpha^5\beta^2 n} + \frac{1073741824L^3\hat{L}^4\omega^2 p^4}{\bar{L}^7\alpha^5\beta^4 n^2}$$
$$+ \frac{1073741824L^3\hat{L}^4\omega^2 p^2}{\bar{L}^7\alpha^5\beta^2 n^2} + \frac{2147483648L^3\hat{L}^4\omega^2 p^3}{\bar{L}^7\alpha^5\beta^3 n^2} + \frac{4294967296L^4\hat{L}^4\omega^2 p^4}{\bar{L}^8\alpha^6\beta^4 n^2}$$

$$+ \frac{4294967296 L^4 \hat{L}^4 \omega^2 p^2}{\bar{L}^8 \alpha^6 \beta^2 n^2} + \frac{8589934592 L^4 \hat{L}^4 \omega^2 p^3}{\bar{L}^8 \alpha^6 \beta^3 n^2} + \frac{8192 L L_{\max} \hat{L}^2 \omega^2 p}{\bar{L}^4 \alpha^2 \beta n^2}$$

$$+ \frac{8192 L L_{\max} \hat{L}^2 \omega^2 p^2}{\bar{L}^4 \alpha^2 \beta^2 n^2} + \frac{65536 L L_{\max} \omega p \tau}{\bar{L}^2 \alpha^2 \beta n} + \frac{65536 L L_{\max} \omega p^2 \tau}{\bar{L}^2 \alpha^2 \beta^2 n}$$

$$+ \frac{65536 L L_{\max} \hat{L}^2 \omega^2 p \tau}{\bar{L}^4 \alpha^3 n^2} + \frac{262144 L \hat{L}^2 \omega p^2}{\bar{L}^3 \alpha^2 \beta n \tau} + \frac{262144 L \hat{L}^2 \omega p^3}{\bar{L}^3 \alpha^2 \beta^2 n \tau}$$

$$+ \frac{524288 L \hat{L}^2 \omega p^3}{\bar{L}^3 \alpha^2 \beta n \tau^2} + \frac{524288 L \hat{L}^2 \omega p^4}{\bar{L}^3 \alpha^2 \beta^2 n \tau^2} + \frac{524288 L_{\max} \hat{L}^2 \omega^2 p^2 \tau}{\bar{L}^3 \alpha^2 \beta n^2}$$

$$+ \frac{524288 L_{\max} \hat{L}^2 \omega^2 p^3 \tau}{\bar{L}^3 \alpha^2 \beta^2 n^2} + \frac{1048576 L^3 L_{\max} \omega p^2}{\bar{L}^4 \alpha^3 \beta n \tau} + \frac{1048576 L^3 L_{\max} \omega p^3}{\bar{L}^4 \alpha^3 \beta^2 n \tau}$$

$$+ \frac{1048576 L^2 \hat{L}^2 \omega p^2}{\bar{L}^4 \alpha^3 \beta n \tau} + \frac{1048576 L^2 \hat{L}^2 \omega p^3}{\bar{L}^4 \alpha^3 \beta^2 n \tau} + \frac{1048576 L^2 L_{\max} \hat{L}^2 \omega^2 p}{\bar{L}^5 \alpha^3 \beta n^2}$$

$$+ \frac{2097152 L^3 L_{\max} \omega p^3}{\bar{L}^4 \alpha^3 \beta n \tau^2} + \frac{2097152 L^3 L_{\max} \omega p^4}{\bar{L}^4 \alpha^3 \beta^2 n \tau^2} + \frac{2097152 L^2 \hat{L}^2 \omega p \tau}{\bar{L}^4 \alpha^4 \beta n}$$

$$+ \frac{2097152 L^2 \hat{L}^2 \omega p^2 \tau}{\bar{L}^4 \alpha^4 \beta^2 n} + \frac{2097152 L^2 \hat{L}^2 \omega p^3}{\bar{L}^4 \alpha^3 \beta n \tau^2} + \frac{2097152 L^2 \hat{L}^2 \omega p^4}{\bar{L}^4 \alpha^3 \beta^2 n \tau^2}$$

$$+ \frac{8388608 L^3 \hat{L}^2 \omega p^2}{\bar{L}^5 \alpha^4 \beta n \tau} + \frac{8388608 L^3 \hat{L}^2 \omega p^3}{\bar{L}^5 \alpha^4 \beta^2 n \tau} + \frac{8388608 L^3 L_{\max} \hat{L}^2 \omega^2 p}{\bar{L}^6 \alpha^4 \beta n^2}$$

$$+ \frac{8388608 L_{\max} \hat{L}^2 \omega^2 p^4 \tau}{\bar{L}^3 \alpha^2 \beta^4 n^2} + \frac{8388608 L_{\max} \hat{L}^2 \omega^2 p^2 \tau}{\bar{L}^3 \alpha^2 \beta^2 n^2} + \frac{8388608 L_{\max} \hat{L}^4 \omega^3 p^2 \tau}{\bar{L}^5 \alpha^3 \beta n^3}$$

$$+ \frac{16777216 L^3 \hat{L}^2 \omega p^3}{\bar{L}^5 \alpha^4 \beta n \tau^2} + \frac{16777216 L^3 \hat{L}^2 \omega p^4}{\bar{L}^5 \alpha^4 \beta^2 n \tau^2} + \frac{16777216 L \hat{L}^4 \omega^2 p^2 \tau}{\bar{L}^5 \alpha^4 \beta n^2}$$

$$+ \frac{16777216 L \hat{L}^4 \omega^2 p^3 \tau}{\bar{L}^5 \alpha^4 \beta^2 n^2} + \frac{16777216 L_{\max} \hat{L}^2 \omega^2 p^3 \tau}{\bar{L}^3 \alpha^2 \beta^3 n^2} + \frac{33554432 L^2 L_{\max} \hat{L}^2 \omega^2 p^4}{\bar{L}^5 \alpha^3 \beta^4 n^2}$$

$$+ \frac{34603008 L^2 L_{\max} \hat{L}^2 \omega^2 p^2}{\bar{L}^5 \alpha^3 \beta^2 n^2} + \frac{67108864 L^4 \hat{L}^2 \omega p^2}{\bar{L}^6 \alpha^5 \beta n \tau} + \frac{67108864 L^4 \hat{L}^2 \omega p^3}{\bar{L}^6 \alpha^5 \beta^2 n \tau}$$

$$+ \frac{67108864 L^2 L_{\max} \hat{L}^2 \omega^2 p^3}{\bar{L}^5 \alpha^3 \beta^3 n^2} + \frac{134217728 L^4 \hat{L}^2 \omega p^3}{\bar{L}^6 \alpha^5 \beta n \tau^2} + \frac{134217728 L^4 \hat{L}^2 \omega p^4}{\bar{L}^6 \alpha^5 \beta^2 n \tau^2}$$

$$+ \frac{134217728 L_{\max} \hat{L}^4 \omega^3 p^4 \tau}{\bar{L}^5 \alpha^3 \beta^4 n^3} + \frac{134217728 L_{\max} \hat{L}^4 \omega^3 p^2 \tau}{\bar{L}^5 \alpha^3 \beta^2 n^3} + \frac{134217728 L^3 L_{\max} \hat{L}^2 \omega^2 p^4}{\bar{L}^6 \alpha^4 \beta^4 n^2}$$

$$+ \frac{134217728 L^2 \hat{L}^4 \omega^2 p^2 \tau}{\bar{L}^6 \alpha^5 \beta n^2} + \frac{134217728 L^2 \hat{L}^4 \omega^2 p^3 \tau}{\bar{L}^6 \alpha^5 \beta^2 n^2} + \frac{142606336 L^3 L_{\max} \hat{L}^2 \omega^2 p^2}{\bar{L}^6 \alpha^4 \beta^2 n^2}$$

$$+ \frac{268435456 L \hat{L}^4 \omega^2 p^4 \tau}{\bar{L}^5 \alpha^4 \beta^4 n^2} + \frac{268435456 L \hat{L}^4 \omega^2 p^2 \tau}{\bar{L}^5 \alpha^4 \beta^2 n^2} + \frac{268435456 L_{\max} \hat{L}^4 \omega^3 p^5 \tau}{\bar{L}^5 \alpha^3 \beta^4 n^3}$$

$$+ \frac{268435456 L_{\max} \hat{L}^4 \omega^3 p^3 \tau}{\bar{L}^5 \alpha^3 \beta^3 n^3} + \frac{268435456 L^3 L_{\max} \hat{L}^2 \omega^2 p^3}{\bar{L}^6 \alpha^4 \beta^3 n^2} + \frac{276824064 L_{\max} \hat{L}^4 \omega^3 p^3 \tau}{\bar{L}^5 \alpha^3 \beta^2 n^3}$$

$$+ \frac{536870912 L \hat{L}^4 \omega^2 p^3 \tau}{\bar{L}^5 \alpha^4 \beta^3 n^2} + \frac{536870912 L_{\max} \hat{L}^4 \omega^3 p^4 \tau}{\bar{L}^5 \alpha^3 \beta^3 n^3} + \frac{536870912 L^2 L_{\max} \hat{L}^4 \omega^3 p^4}{\bar{L}^7 \alpha^4 \beta^4 n^3}$$

$$+ \frac{536870912 L^2 L_{\max} \hat{L}^4 \omega^3 p^2}{\bar{L}^7 \alpha^4 \beta^2 n^3} + \frac{1073741824 L^2 L_{\max} \hat{L}^4 \omega^3 p^3}{\bar{L}^7 \alpha^4 \beta^3 n^3} + \frac{1073741824 L^2 \hat{L}^4 \omega^2 p^4 \tau}{\bar{L}^6 \alpha^5 \beta^4 n^2}$$

$$+ \frac{1073741824 L^2 \hat{L}^4 \omega^2 p^2 \tau}{\bar{L}^6 \alpha^5 \beta^2 n^2} + \frac{2147483648 L^3 L_{\max} \hat{L}^4 \omega^3 p^4}{\bar{L}^8 \alpha^5 \beta^4 n^3} + \frac{2147483648 L^3 L_{\max} \hat{L}^4 \omega^3 p^2}{\bar{L}^8 \alpha^5 \beta^2 n^3}$$

$$+ \frac{2147483648 L^2 \hat{L}^4 \omega^2 p^3 \tau}{\bar{L}^6 \alpha^5 \beta^3 n^2} + \frac{4294967296 L_{\max} \hat{L}^6 \omega^4 p^5 \tau}{\bar{L}^7 \alpha^4 \beta^4 n^4} + \frac{4294967296 L_{\max} \hat{L}^6 \omega^4 p^3 \tau}{\bar{L}^7 \alpha^4 \beta^2 n^4}$$

$$+ \frac{4294967296 L^3 L_{\max} \hat{L}^4 \omega^3 p^3}{\bar{L}^8 \alpha^5 \beta^3 n^3} + \frac{4294967296 L^3 \hat{L}^4 \omega^2 p^5}{\bar{L}^7 \alpha^5 \beta^4 n^2 \tau} + \frac{4294967296 L^3 \hat{L}^4 \omega^2 p^3}{\bar{L}^7 \alpha^5 \beta^2 n^2 \tau}$$

$$+ \frac{8589934592 L \hat{L}^6 \omega^3 p^5 \tau}{\bar{L}^7 \alpha^5 \beta^4 n^3} + \frac{8589934592 L \hat{L}^6 \omega^3 p^3 \tau}{\bar{L}^7 \alpha^5 \beta^2 n^3} + \frac{8589934592 L_{\max} \hat{L}^6 \omega^4 p^4 \tau}{\bar{L}^7 \alpha^4 \beta^3 n^4}$$

$$+ \frac{8589934592 L^3 \hat{L}^4 \omega^2 p^6}{\bar{L}^7 \alpha^5 \beta^4 n^2 \tau^2} + \frac{8589934592 L^3 \hat{L}^4 \omega^2 p^4}{\bar{L}^7 \alpha^5 \beta^3 n^2 \tau} + \frac{8589934592 L^3 \hat{L}^4 \omega^2 p^4}{\bar{L}^7 \alpha^5 \beta^2 n^2 \tau^2}$$

$$+ \frac{17179869184 L \hat{L}^6 \omega^3 p^4 \tau}{\bar{L}^7 \alpha^5 \beta^3 n^3} + \frac{17179869184 L^3 \hat{L}^4 \omega^2 p^5}{\bar{L}^7 \alpha^5 \beta^3 n^2 \tau^2} + \frac{17179869184 L^4 \hat{L}^4 \omega^2 p^5}{\bar{L}^8 \alpha^6 \beta^4 n^2 \tau}$$

$$+ \frac{17179869184 L^4 \hat{L}^4 \omega^2 p^3}{\bar{L}^8 \alpha^6 \beta^2 n^2 \tau} + \frac{34359738368 L^2 \hat{L}^6 \omega^3 p^5 \tau}{\bar{L}^8 \alpha^6 \beta^4 n^3} + \frac{34359738368 L^2 \hat{L}^6 \omega^3 p^3 \tau}{\bar{L}^8 \alpha^6 \beta^2 n^3}$$

$$+ \frac{34359738368 L^4 \hat{L}^4 \omega^2 p^6}{\bar{L}^8 \alpha^6 \beta^4 n^2 \tau^2} + \frac{34359738368 L^4 \hat{L}^4 \omega^2 p^4}{\bar{L}^8 \alpha^6 \beta^3 n^2 \tau} + \frac{34359738368 L^4 \hat{L}^4 \omega^2 p^4}{\bar{L}^8 \alpha^6 \beta^2 n^2 \tau^2}$$

$$+ \frac{68719476736 L^2 \hat{L}^6 \omega^3 p^4 \tau}{\bar{L}^8 \alpha^6 \beta^3 n^3} + \frac{68719476736 L^4 \hat{L}^4 \omega^2 p^5}{\bar{L}^8 \alpha^6 \beta^3 n^2 \tau^2} + \frac{1048576 L L_{\max} \hat{L}^2 \omega^2 p \tau}{\bar{L}^4 \alpha^3 \beta n^2}$$

$$+ \frac{4194304 L L_{\max} \hat{L}^2 \omega^2 p^2 \tau}{\bar{L}^4 \alpha^3 \beta n^2} + \frac{4194304 L L_{\max} \hat{L}^2 \omega^2 p^3 \tau}{\bar{L}^4 \alpha^3 \beta^2 n^2} + \frac{4194304 L^2 L_{\max} \hat{L}^2 \omega^2 p^2}{\bar{L}^5 \alpha^3 \beta n^2 \tau}$$

$$+ \frac{8388608 L^2 L_{\max} \hat{L}^2 \omega^2 p^3}{\bar{L}^5 \alpha^3 \beta n^2 \tau^2} + \frac{33554432 L L_{\max} \hat{L}^2 \omega^2 p^4 \tau}{\bar{L}^4 \alpha^3 \beta^4 n^2} + \frac{33554432 L^3 L_{\max} \hat{L}^2 \omega^2 p^2}{\bar{L}^6 \alpha^4 \beta n^2 \tau}$$

$$+ \frac{34603008 L L_{\max} \hat{L}^2 \omega^2 p^2 \tau}{\bar{L}^4 \alpha^3 \beta^2 n^2} + \frac{67108864 L L_{\max} \hat{L}^2 \omega^2 p^3 \tau}{\bar{L}^4 \alpha^3 \beta^3 n^2} + \frac{67108864 L L_{\max} \hat{L}^4 \omega^3 p^2 \tau}{\bar{L}^6 \alpha^4 \beta n^3}$$

$$+ \frac{67108864 L^3 L_{\max} \hat{L}^2 \omega^2 p^3}{\bar{L}^6 \alpha^4 \beta n^2 \tau^2} + \frac{134217728 L^2 L_{\max} \hat{L}^2 \omega^2 p^5}{\bar{L}^5 \alpha^3 \beta^4 n^2 \tau} + \frac{138412032 L^2 L_{\max} \hat{L}^2 \omega^2 p^3}{\bar{L}^5 \alpha^3 \beta^2 n^2 \tau}$$

$$+ \frac{268435456 L^2 L_{\max} \hat{L}^2 \omega^2 p^6}{\bar{L}^5 \alpha^3 \beta^4 n^2 \tau^2} + \frac{268435456 L^2 L_{\max} \hat{L}^2 \omega^2 p^4}{\bar{L}^5 \alpha^3 \beta^3 n^2 \tau} + \frac{276824064 L^2 L_{\max} \hat{L}^2 \omega^2 p^4}{\bar{L}^5 \alpha^3 \beta^2 n^2 \tau^2}$$

$$+ \frac{536870912 L L_{\max} \hat{L}^4 \omega^3 p^4 \tau}{\bar{L}^6 \alpha^4 \beta^4 n^3} + \frac{536870912 L L_{\max} \hat{L}^4 \omega^3 p^2 \tau}{\bar{L}^6 \alpha^4 \beta^2 n^3} + \frac{536870912 L^2 L_{\max} \hat{L}^2 \omega^2 p^5}{\bar{L}^5 \alpha^3 \beta^3 n^2 \tau^2}$$

$$+ \frac{536870912 L^3 L_{\max} \hat{L}^2 \omega^2 p^5}{\bar{L}^6 \alpha^4 \beta^4 n^2 \tau} + \frac{570425344 L^3 L_{\max} \hat{L}^2 \omega^2 p^3}{\bar{L}^6 \alpha^4 \beta^2 n^2 \tau} + \frac{1073741824 L L_{\max} \hat{L}^4 \omega^3 p^5 \tau}{\bar{L}^6 \alpha^4 \beta^4 n^3}$$

$$+ \frac{1073741824 L L_{\max} \hat{L}^4 \omega^3 p^3 \tau}{\bar{L}^6 \alpha^4 \beta^3 n^3} + \frac{1073741824 L^3 L_{\max} \hat{L}^2 \omega^2 p^6}{\bar{L}^6 \alpha^4 \beta^4 n^2 \tau^2} + \frac{1073741824 L^3 L_{\max} \hat{L}^2 \omega^2 p^4}{\bar{L}^6 \alpha^4 \beta^3 n^2 \tau}$$

$$+ \frac{1140850688 L L_{\max} \hat{L}^4 \omega^3 p^3 \tau}{\bar{L}^6 \alpha^4 \beta^2 n^3} + \frac{1140850688 L^3 L_{\max} \hat{L}^2 \omega^2 p^4}{\bar{L}^6 \alpha^4 \beta^2 n^2 \tau^2} + \frac{2147483648 L L_{\max} \hat{L}^4 \omega^3 p^4 \tau}{\bar{L}^6 \alpha^4 \beta^3 n^3}$$

$$+ \frac{2147483648 L^3 L_{\max} \hat{L}^2 \omega^2 p^5}{\bar{L}^6 \alpha^4 \beta^3 n^2 \tau^2} + \frac{2147483648 L^2 L_{\max} \hat{L}^4 \omega^3 p^5}{\bar{L}^7 \alpha^4 \beta^4 n^3 \tau} + \frac{2147483648 L^2 L_{\max} \hat{L}^4 \omega^3 p^3}{\bar{L}^7 \alpha^4 \beta^2 n^3 \tau}$$

$$+ \frac{4294967296 L^2 L_{\max} \hat{L}^4 \omega^3 p^6}{\bar{L}^7 \alpha^4 \beta^4 n^3 \tau^2} + \frac{4294967296 L^2 L_{\max} \hat{L}^4 \omega^3 p^4}{\bar{L}^7 \alpha^4 \beta^3 n^3 \tau} + \frac{4294967296 L^2 L_{\max} \hat{L}^4 \omega^3 p^4}{\bar{L}^7 \alpha^4 \beta^2 n^3 \tau^2}$$

$$+ \frac{8589934592 L^2 L_{\max} \hat{L}^4 \omega^3 p^5}{\bar{L}^7 \alpha^4 \beta^3 n^3 \tau^2} + \frac{8589934592 L^3 L_{\max} \hat{L}^4 \omega^3 p^5}{\bar{L}^8 \alpha^5 \beta^4 n^3} + \frac{8589934592 L^3 L_{\max} \hat{L}^4 \omega^3 p^3}{\bar{L}^8 \alpha^5 \beta^2 n^3}$$

$$+ \frac{17179869184 L L_{\max} \hat{L}^6 \omega^4 p^5 \tau}{\bar{L}^8 \alpha^5 \beta^4 n^4} + \frac{17179869184 L L_{\max} \hat{L}^6 \omega^4 p^3 \tau}{\bar{L}^8 \alpha^5 \beta^2 n^4} + \frac{17179869184 L^3 L_{\max} \hat{L}^4 \omega^3 p^6}{\bar{L}^8 \alpha^5 \beta^4 n^3 \tau^2}$$

$$+ \frac{17179869184 L^3 L_{\max} \hat{L}^4 \omega^3 p^4}{\bar{L}^8 \alpha^5 \beta^3 n^3 \tau} + \frac{17179869184 L^3 L_{\max} \hat{L}^4 \omega^3 p^4}{\bar{L}^8 \alpha^5 \beta^2 n^3 \tau^2} + \frac{34359738368 L L_{\max} \hat{L}^6 \omega^4 p^4 \tau}{\bar{L}^8 \alpha^5 \beta^3 n^4}$$

$$+ \frac{34359738368 L^3 L_{\max} \hat{L}^4 \omega^3 p^5}{\bar{L}^8 \alpha^5 \beta^3 n^3 \tau^2} \leq \frac{3}{4}$$

# L    Symbolically Computed Constraints for $\bar{L}$ Such That The Inequality from Section K Holds

$$\frac{544L^2}{\bar{L}^2\alpha^2} \leq \frac{1}{326} \quad (109)$$

$$\frac{16384L^4}{\bar{L}^4\alpha^4} \leq \frac{1}{326} \quad (110)$$

$$\frac{4L_{\max}\omega}{\bar{L}n} \leq \frac{1}{326} \quad (111)$$

$$\frac{2048L^2p}{\bar{L}^2\alpha^2\tau} \leq \frac{1}{326} \quad (112)$$

$$\frac{4096L^2p^2}{\bar{L}^2\alpha^2\tau^2} \leq \frac{1}{326} \quad (113)$$

$$\frac{65536L^4p}{\bar{L}^4\alpha^4\tau} \leq \frac{1}{326} \quad (114)$$

$$\frac{131072L^4p^2}{\bar{L}^4\alpha^4\tau^2} \leq \frac{1}{326} \quad (115)$$

$$\frac{16LL_{\max}\omega}{\bar{L}^2\alpha n} \leq \frac{1}{326} \quad (116)$$

$$\frac{128L_{\max}\omega p}{\bar{L}\beta n} \leq \frac{1}{326} \quad (117)$$

$$\frac{128L_{\max}\omega p^2}{\bar{L}\beta^2 n} \leq \frac{1}{326} \quad (118)$$

$$\frac{8192L^3L_{\max}\omega}{\bar{L}^4\alpha^3 n} \leq \frac{1}{326} \quad (119)$$

$$\frac{512LL_{\max}\omega p}{\bar{L}^2\alpha\beta n} \leq \frac{1}{326} \quad (120)$$

$$\frac{512LL_{\max}\omega p^2}{\bar{L}^2\alpha\beta^2 n} \leq \frac{1}{326} \quad (121)$$

$$\frac{2048L_{\max}\hat{L}^2\omega^2 p}{\bar{L}^3\alpha\beta n^2} \leq \frac{1}{326} \quad (122)$$

$$\frac{2048L_{\max}\hat{L}^2\omega^2 p^2}{\bar{L}^3\alpha\beta^2 n^2} \leq \frac{1}{326} \quad (123)$$

$$\frac{4096LL_{\max}\omega p\tau}{\bar{L}^2\alpha^2 n} \leq \frac{1}{326} \quad (124)$$

$$\frac{32768L^3L_{\max}\omega p}{\bar{L}^4\alpha^3 n\tau} \leq \frac{1}{326} \quad (125)$$

$$\frac{65536L^3L_{\max}\omega p^2}{\bar{L}^4\alpha^3 n\tau^2} \leq \frac{1}{326} \quad (126)$$

$$\frac{69632L\hat{L}^2\omega p}{\bar{L}^3\alpha^2\beta n} \leq \frac{1}{326} \quad (127)$$

$$\frac{69632L\hat{L}^2\omega p^2}{\bar{L}^3\alpha^2\beta^2 n} \leq \frac{1}{326} \quad (128)$$

$$\frac{131072L^2\hat{L}^2\omega p\tau}{\bar{L}^4\alpha^4 n} \leq \frac{1}{326} \quad (129)$$

$$\frac{262144L^3L_{\max}\omega p}{\bar{L}^4\alpha^3\beta n} \leq \frac{1}{326} \quad (130)$$

$$\frac{262144L^3L_{\max}\omega p^2}{\bar{L}^4\alpha^3\beta^2 n} \leq \frac{1}{326} \quad (131)$$

$$\frac{278528L^2\hat{L}^2\omega p}{\bar{L}^4\alpha^3\beta n} \leq \frac{1}{326} \quad (132)$$

$$\frac{278528L^2\hat{L}^2\omega p^2}{\bar{L}^4\alpha^3\beta^2 n} \leq \frac{1}{326} \quad (133)$$

$$\frac{2097152L^3\hat{L}^2\omega p}{\bar{L}^5\alpha^4\beta n} \leq \frac{1}{326} \quad (134)$$

$$\frac{2097152L^3\hat{L}^2\omega p^2}{\bar{L}^5\alpha^4\beta^2 n} \leq \frac{1}{326} \quad (135)$$

$$\frac{16777216L^4\hat{L}^2\omega p}{\bar{L}^6\alpha^5\beta n} \leq \frac{1}{326} \quad (136)$$

$$\frac{16777216L^4\hat{L}^2\omega p^2}{\bar{L}^6\alpha^5\beta^2 n} \leq \frac{1}{326} \quad (137)$$

$$\frac{1073741824L^3\hat{L}^4\omega^2 p^4}{\bar{L}^7\alpha^5\beta^4 n^2} \leq \frac{1}{326} \quad (138)$$

$$\frac{1073741824L^3\hat{L}^4\omega^2 p^2}{\bar{L}^7\alpha^5\beta^2 n^2} \leq \frac{1}{326} \quad (139)$$

$$\frac{2147483648L^3\hat{L}^4\omega^2 p^3}{\bar{L}^7\alpha^5\beta^3 n^2} \leq \frac{1}{326} \quad (140)$$

$$\frac{4294967296L^4\hat{L}^4\omega^2 p^4}{\bar{L}^8\alpha^6\beta^4 n^2} \leq \frac{1}{326} \quad (141)$$

$$\frac{4294967296 L^4 \hat{L}^4 \omega^2 p^2}{\bar{L}^8 \alpha^6 \beta^2 n^2} \leq \frac{1}{326} \tag{142}$$

$$\frac{8589934592 L^4 \hat{L}^4 \omega^2 p^3}{\bar{L}^8 \alpha^6 \beta^3 n^2} \leq \frac{1}{326} \tag{143}$$

$$\frac{8192 L L_{\max} \hat{L}^2 \omega^2 p}{\bar{L}^4 \alpha^2 \beta n^2} \leq \frac{1}{326} \tag{144}$$

$$\frac{8192 L L_{\max} \hat{L}^2 \omega^2 p^2}{\bar{L}^4 \alpha^2 \beta^2 n^2} \leq \frac{1}{326} \tag{145}$$

$$\frac{65536 L L_{\max} \omega p \tau}{\bar{L}^2 \alpha^2 \beta n} \leq \frac{1}{326} \tag{146}$$

$$\frac{65536 L L_{\max} \omega p^2 \tau}{\bar{L}^2 \alpha^2 \beta^2 n} \leq \frac{1}{326} \tag{147}$$

$$\frac{65536 L L_{\max} \hat{L}^2 \omega^2 p \tau}{\bar{L}^4 \alpha^3 n^2} \leq \frac{1}{326} \tag{148}$$

$$\frac{262144 L \hat{L}^2 \omega p^2}{\bar{L}^3 \alpha^2 \beta n \tau} \leq \frac{1}{326} \tag{149}$$

$$\frac{262144 L \hat{L}^2 \omega p^3}{\bar{L}^3 \alpha^2 \beta^2 n \tau} \leq \frac{1}{326} \tag{150}$$

$$\frac{524288 L \hat{L}^2 \omega p^3}{\bar{L}^3 \alpha^2 \beta n \tau^2} \leq \frac{1}{326} \tag{151}$$

$$\frac{524288 L \hat{L}^2 \omega p^4}{\bar{L}^3 \alpha^2 \beta^2 n \tau^2} \leq \frac{1}{326} \tag{152}$$

$$\frac{524288 L_{\max} \hat{L}^2 \omega^2 p^2 \tau}{\bar{L}^3 \alpha^2 \beta n^2} \leq \frac{1}{326} \tag{153}$$

$$\frac{524288 L_{\max} \hat{L}^2 \omega^2 p^3 \tau}{\bar{L}^3 \alpha^2 \beta^2 n^2} \leq \frac{1}{326} \tag{154}$$

$$\frac{1048576 L^3 L_{\max} \omega p^2}{\bar{L}^4 \alpha^3 \beta n \tau} \leq \frac{1}{326} \tag{155}$$

$$\frac{1048576 L^3 L_{\max} \omega p^3}{\bar{L}^4 \alpha^3 \beta^2 n \tau} \leq \frac{1}{326} \tag{156}$$

$$\frac{1048576 L^2 \hat{L}^2 \omega p^2}{\bar{L}^4 \alpha^3 \beta n \tau} \leq \frac{1}{326} \tag{157}$$

$$\frac{1048576 L^2 \hat{L}^2 \omega p^3}{\bar{L}^4 \alpha^3 \beta^2 n \tau} \leq \frac{1}{326} \tag{158}$$

$$\frac{1048576 L^2 L_{\max} \hat{L}^2 \omega^2 p}{\bar{L}^5 \alpha^3 \beta n^2} \leq \frac{1}{326} \tag{159}$$

$$\frac{2097152 L^3 L_{\max} \omega p^3}{\bar{L}^4 \alpha^3 \beta n \tau^2} \leq \frac{1}{326} \tag{160}$$

$$\frac{2097152 L^3 L_{\max} \omega p^4}{\bar{L}^4 \alpha^3 \beta^2 n \tau^2} \leq \frac{1}{326} \tag{161}$$

$$\frac{2097152 L^2 \hat{L}^2 \omega p \tau}{\bar{L}^4 \alpha^4 \beta n} \leq \frac{1}{326} \tag{162}$$

$$\frac{2097152 L^2 \hat{L}^2 \omega p^2 \tau}{\bar{L}^4 \alpha^4 \beta^2 n} \leq \frac{1}{326} \tag{163}$$

$$\frac{2097152 L^2 \hat{L}^2 \omega p^3}{\bar{L}^4 \alpha^3 \beta n \tau^2} \leq \frac{1}{326} \tag{164}$$

$$\frac{2097152 L^2 \hat{L}^2 \omega p^4}{\bar{L}^4 \alpha^3 \beta^2 n \tau^2} \leq \frac{1}{326} \tag{165}$$

$$\frac{8388608 L^3 \hat{L}^2 \omega p^2}{\bar{L}^5 \alpha^4 \beta n \tau} \leq \frac{1}{326} \tag{166}$$

$$\frac{8388608 L^3 \hat{L}^2 \omega p^3}{\bar{L}^5 \alpha^4 \beta^2 n \tau} \leq \frac{1}{326} \tag{167}$$

$$\frac{8388608 L^3 L_{\max} \hat{L}^2 \omega^2 p}{\bar{L}^6 \alpha^4 \beta n^2} \leq \frac{1}{326} \tag{168}$$

$$\frac{8388608 L_{\max} \hat{L}^2 \omega^2 p^4 \tau}{\bar{L}^3 \alpha^2 \beta^4 n^2} \leq \frac{1}{326} \tag{169}$$

$$\frac{8388608 L_{\max} \hat{L}^2 \omega^2 p^2 \tau}{\bar{L}^3 \alpha^2 \beta^2 n^2} \leq \frac{1}{326} \tag{170}$$

$$\frac{8388608 L_{\max} \hat{L}^4 \omega^3 p^2 \tau}{\bar{L}^5 \alpha^3 \beta n^3} \leq \frac{1}{326} \tag{171}$$

$$\frac{16777216L^3\hat{L}^2\omega p^3}{\bar{L}^5\alpha^4\beta n\tau^2} \leq \frac{1}{326} \quad (172)$$

$$\frac{16777216L^3\hat{L}^2\omega p^4}{\bar{L}^5\alpha^4\beta^2 n\tau^2} \leq \frac{1}{326} \quad (173)$$

$$\frac{16777216L\hat{L}^4\omega^2 p^2\tau}{\bar{L}^5\alpha^4\beta n^2} \leq \frac{1}{326} \quad (174)$$

$$\frac{16777216L\hat{L}^4\omega^2 p^3\tau}{\bar{L}^5\alpha^4\beta^2 n^2} \leq \frac{1}{326} \quad (175)$$

$$\frac{16777216L_{\max}\hat{L}^2\omega^2 p^3\tau}{\bar{L}^3\alpha^2\beta^3 n^2} \leq \frac{1}{326} \quad (176)$$

$$\frac{33554432L^2 L_{\max}\hat{L}^2\omega^2 p^4}{\bar{L}^5\alpha^3\beta^4 n^2} \leq \frac{1}{326} \quad (177)$$

$$\frac{34603008L^2 L_{\max}\hat{L}^2\omega^2 p^2}{\bar{L}^5\alpha^3\beta^2 n^2} \leq \frac{1}{326} \quad (178)$$

$$\frac{67108864L^4\hat{L}^2\omega p^2}{\bar{L}^6\alpha^5\beta n\tau} \leq \frac{1}{326} \quad (179)$$

$$\frac{67108864L^4\hat{L}^2\omega p^3}{\bar{L}^6\alpha^5\beta^2 n\tau} \leq \frac{1}{326} \quad (180)$$

$$\frac{67108864L^2 L_{\max}\hat{L}^2\omega^2 p^3}{\bar{L}^5\alpha^3\beta^3 n^2} \leq \frac{1}{326} \quad (181)$$

$$\frac{134217728L^4\hat{L}^2\omega p^3}{\bar{L}^6\alpha^5\beta n\tau^2} \leq \frac{1}{326} \quad (182)$$

$$\frac{134217728L^4\hat{L}^2\omega p^4}{\bar{L}^6\alpha^5\beta^2 n\tau^2} \leq \frac{1}{326} \quad (183)$$

$$\frac{134217728L_{\max}\hat{L}^4\omega^3 p^4\tau}{\bar{L}^5\alpha^3\beta^4 n^3} \leq \frac{1}{326} \quad (184)$$

$$\frac{134217728L_{\max}\hat{L}^4\omega^3 p^2\tau}{\bar{L}^5\alpha^3\beta^2 n^3} \leq \frac{1}{326} \quad (185)$$

$$\frac{134217728L^3 L_{\max}\hat{L}^2\omega^2 p^4}{\bar{L}^6\alpha^4\beta^4 n^2} \leq \frac{1}{326} \quad (186)$$

$$\frac{134217728L^2\hat{L}^4\omega^2 p^2\tau}{\bar{L}^6\alpha^5\beta n^2} \leq \frac{1}{326} \quad (187)$$

$$\frac{134217728L^2\hat{L}^4\omega^2 p^3\tau}{\bar{L}^6\alpha^5\beta^2 n^2} \leq \frac{1}{326} \quad (188)$$

$$\frac{142606336L^3 L_{\max}\hat{L}^2\omega^2 p^2}{\bar{L}^6\alpha^4\beta^2 n^2} \leq \frac{1}{326} \quad (189)$$

$$\frac{268435456L\hat{L}^4\omega^2 p^4\tau}{\bar{L}^5\alpha^4\beta^4 n^2} \leq \frac{1}{326} \quad (190)$$

$$\frac{268435456L\hat{L}^4\omega^2 p^2\tau}{\bar{L}^5\alpha^4\beta^2 n^2} \leq \frac{1}{326} \quad (191)$$

$$\frac{268435456L_{\max}\hat{L}^4\omega^3 p^5\tau}{\bar{L}^5\alpha^3\beta^4 n^3} \leq \frac{1}{326} \quad (192)$$

$$\frac{268435456L_{\max}\hat{L}^4\omega^3 p^3\tau}{\bar{L}^5\alpha^3\beta^3 n^3} \leq \frac{1}{326} \quad (193)$$

$$\frac{268435456L^3 L_{\max}\hat{L}^2\omega^2 p^3}{\bar{L}^6\alpha^4\beta^3 n^2} \leq \frac{1}{326} \quad (194)$$

$$\frac{276824064L_{\max}\hat{L}^4\omega^3 p^3\tau}{\bar{L}^5\alpha^3\beta^2 n^3} \leq \frac{1}{326} \quad (195)$$

$$\frac{536870912L\hat{L}^4\omega^2 p^3\tau}{\bar{L}^5\alpha^4\beta^3 n^2} \leq \frac{1}{326} \quad (196)$$

$$\frac{536870912L_{\max}\hat{L}^4\omega^3 p^4\tau}{\bar{L}^5\alpha^3\beta^3 n^3} \leq \frac{1}{326} \quad (197)$$

$$\frac{536870912L^2 L_{\max}\hat{L}^4\omega^3 p^4}{\bar{L}^7\alpha^4\beta^4 n^3} \leq \frac{1}{326} \quad (198)$$

$$\frac{536870912L^2 L_{\max}\hat{L}^4\omega^3 p^2}{\bar{L}^7\alpha^4\beta^2 n^3} \leq \frac{1}{326} \quad (199)$$

$$\frac{1073741824L^2 L_{\max}\hat{L}^4\omega^3 p^3}{\bar{L}^7\alpha^4\beta^3 n^3} \leq \frac{1}{326} \quad (200)$$

$$\frac{1073741824L^2\hat{L}^4\omega^2 p^4\tau}{\bar{L}^6\alpha^5\beta^4 n^2} \leq \frac{1}{326} \quad (201)$$

$$\frac{1073741824L^2\hat{L}^4\omega^2p^2\tau}{\bar{L}^6\alpha^5\beta^2n^2} \leq \frac{1}{326}$$
(202)

$$\frac{2147483648L^3L_{\max}\hat{L}^4\omega^3p^4}{\bar{L}^8\alpha^5\beta^4n^3} \leq \frac{1}{326}$$
(203)

$$\frac{2147483648L^3L_{\max}\hat{L}^4\omega^3p^2}{\bar{L}^8\alpha^5\beta^2n^3} \leq \frac{1}{326}$$
(204)

$$\frac{2147483648L^2\hat{L}^4\omega^2p^3\tau}{\bar{L}^6\alpha^5\beta^3n^2} \leq \frac{1}{326}$$
(205)

$$\frac{4294967296L_{\max}\hat{L}^6\omega^4p^5\tau}{\bar{L}^7\alpha^4\beta^4n^4} \leq \frac{1}{326}$$
(206)

$$\frac{4294967296L_{\max}\hat{L}^6\omega^4p^3\tau}{\bar{L}^7\alpha^4\beta^2n^4} \leq \frac{1}{326}$$
(207)

$$\frac{4294967296L^3L_{\max}\hat{L}^4\omega^3p^3}{\bar{L}^8\alpha^5\beta^3n^3} \leq \frac{1}{326}$$
(208)

$$\frac{4294967296L^3\hat{L}^4\omega^2p^5}{\bar{L}^7\alpha^5\beta^4n^2\tau} \leq \frac{1}{326}$$
(209)

$$\frac{4294967296L^3\hat{L}^4\omega^2p^3}{\bar{L}^7\alpha^5\beta^2n^2\tau} \leq \frac{1}{326}$$
(210)

$$\frac{8589934592L\hat{L}^6\omega^3p^5\tau}{\bar{L}^7\alpha^5\beta^4n^3} \leq \frac{1}{326}$$
(211)

$$\frac{8589934592L\hat{L}^6\omega^3p^3\tau}{\bar{L}^7\alpha^5\beta^2n^3} \leq \frac{1}{326}$$
(212)

$$\frac{8589934592L_{\max}\hat{L}^6\omega^4p^4\tau}{\bar{L}^7\alpha^4\beta^3n^4} \leq \frac{1}{326}$$
(213)

$$\frac{8589934592L^3\hat{L}^4\omega^2p^6}{\bar{L}^7\alpha^5\beta^4n^2\tau^2} \leq \frac{1}{326}$$
(214)

$$\frac{8589934592L^3\hat{L}^4\omega^2p^4}{\bar{L}^7\alpha^5\beta^3n^2\tau} \leq \frac{1}{326}$$
(215)

$$\frac{8589934592L^3\hat{L}^4\omega^2p^4}{\bar{L}^7\alpha^5\beta^2n^2\tau^2} \leq \frac{1}{326}$$
(216)

$$\frac{17179869184L\hat{L}^6\omega^3p^4\tau}{\bar{L}^7\alpha^5\beta^3n^3} \leq \frac{1}{326}$$
(217)

$$\frac{17179869184L^3\hat{L}^4\omega^2p^5}{\bar{L}^7\alpha^5\beta^3n^2\tau^2} \leq \frac{1}{326}$$
(218)

$$\frac{17179869184L^4\hat{L}^4\omega^2p^5}{\bar{L}^8\alpha^6\beta^4n^2\tau} \leq \frac{1}{326}$$
(219)

$$\frac{17179869184L^4\hat{L}^4\omega^2p^3}{\bar{L}^8\alpha^6\beta^2n^2\tau} \leq \frac{1}{326}$$
(220)

$$\frac{34359738368L^2\hat{L}^6\omega^3p^5\tau}{\bar{L}^8\alpha^6\beta^4n^3} \leq \frac{1}{326}$$
(221)

$$\frac{34359738368L^2\hat{L}^6\omega^3p^3\tau}{\bar{L}^8\alpha^6\beta^2n^3} \leq \frac{1}{326}$$
(222)

$$\frac{34359738368L^4\hat{L}^4\omega^2p^6}{\bar{L}^8\alpha^6\beta^4n^2\tau^2} \leq \frac{1}{326}$$
(223)

$$\frac{34359738368L^4\hat{L}^4\omega^2p^4}{\bar{L}^8\alpha^6\beta^3n^2\tau} \leq \frac{1}{326}$$
(224)

$$\frac{34359738368L^4\hat{L}^4\omega^2p^4}{\bar{L}^8\alpha^6\beta^2n^2\tau^2} \leq \frac{1}{326}$$
(225)

$$\frac{68719476736L^2\hat{L}^6\omega^3p^4\tau}{\bar{L}^8\alpha^6\beta^3n^3} \leq \frac{1}{326}$$
(226)

$$\frac{68719476736L^4\hat{L}^4\omega^2p^5}{\bar{L}^8\alpha^6\beta^3n^2\tau^2} \leq \frac{1}{326}$$
(227)

$$\frac{1048576LL_{\max}\hat{L}^2\omega^2p\tau}{\bar{L}^4\alpha^3\beta n^2} \leq \frac{1}{326}$$
(228)

$$\frac{4194304LL_{\max}\hat{L}^2\omega^2p^2\tau}{\bar{L}^4\alpha^3\beta n^2} \leq \frac{1}{326}$$
(229)

$$\frac{4194304LL_{\max}\hat{L}^2\omega^2p^3\tau}{\bar{L}^4\alpha^3\beta^2n^2} \leq \frac{1}{326}$$
(230)

$$\frac{4194304L^2L_{\max}\hat{L}^2\omega^2p^2}{\bar{L}^5\alpha^3\beta n^2\tau} \leq \frac{1}{326}$$
(231)

$$\frac{8388608 L^2 L_{\max}\hat{L}^2\omega^2 p^3}{\bar{L}^5\alpha^3\beta n^2\tau^2} \leq \frac{1}{326}$$
(232)

$$\frac{33554432 L L_{\max}\hat{L}^2\omega^2 p^4\tau}{\bar{L}^4\alpha^3\beta^4 n^2} \leq \frac{1}{326}$$
(233)

$$\frac{33554432 L^3 L_{\max}\hat{L}^2\omega^2 p^2}{\bar{L}^6\alpha^4\beta n^2\tau} \leq \frac{1}{326}$$
(234)

$$\frac{34603008 L L_{\max}\hat{L}^2\omega^2 p^2\tau}{\bar{L}^4\alpha^3\beta^2 n^2} \leq \frac{1}{326}$$
(235)

$$\frac{67108864 L L_{\max}\hat{L}^2\omega^2 p^3\tau}{\bar{L}^4\alpha^3\beta^3 n^2} \leq \frac{1}{326}$$
(236)

$$\frac{67108864 L L_{\max}\hat{L}^4\omega^3 p^2\tau}{\bar{L}^6\alpha^4\beta n^3} \leq \frac{1}{326}$$
(237)

$$\frac{67108864 L^3 L_{\max}\hat{L}^2\omega^2 p^3}{\bar{L}^6\alpha^4\beta n^2\tau^2} \leq \frac{1}{326}$$
(238)

$$\frac{134217728 L^2 L_{\max}\hat{L}^2\omega^2 p^5}{\bar{L}^5\alpha^3\beta^4 n^2\tau} \leq \frac{1}{326}$$
(239)

$$\frac{138412032 L^2 L_{\max}\hat{L}^2\omega^2 p^3}{\bar{L}^5\alpha^3\beta^2 n^2\tau} \leq \frac{1}{326}$$
(240)

$$\frac{268435456 L^2 L_{\max}\hat{L}^2\omega^2 p^6}{\bar{L}^5\alpha^3\beta^4 n^2\tau^2} \leq \frac{1}{326}$$
(241)

$$\frac{268435456 L^2 L_{\max}\hat{L}^2\omega^2 p^4}{\bar{L}^5\alpha^3\beta^3 n^2\tau} \leq \frac{1}{326}$$
(242)

$$\frac{276824064 L^2 L_{\max}\hat{L}^2\omega^2 p^4}{\bar{L}^5\alpha^3\beta^2 n^2\tau^2} \leq \frac{1}{326}$$
(243)

$$\frac{536870912 L L_{\max}\hat{L}^4\omega^3 p^4\tau}{\bar{L}^6\alpha^4\beta^4 n^3} \leq \frac{1}{326}$$
(244)

$$\frac{536870912 L L_{\max}\hat{L}^4\omega^3 p^2\tau}{\bar{L}^6\alpha^4\beta^2 n^3} \leq \frac{1}{326}$$
(245)

$$\frac{536870912 L^2 L_{\max}\hat{L}^2\omega^2 p^5}{\bar{L}^5\alpha^3\beta^3 n^2\tau^2} \leq \frac{1}{326}$$
(246)

$$\frac{536870912 L^3 L_{\max}\hat{L}^2\omega^2 p^5}{\bar{L}^6\alpha^4\beta^4 n^2\tau} \leq \frac{1}{326}$$
(247)

$$\frac{570425344 L^3 L_{\max}\hat{L}^2\omega^2 p^3}{\bar{L}^6\alpha^4\beta^2 n^2\tau} \leq \frac{1}{326}$$
(248)

$$\frac{1073741824 L L_{\max}\hat{L}^4\omega^3 p^5\tau}{\bar{L}^6\alpha^4\beta^4 n^3} \leq \frac{1}{326}$$
(249)

$$\frac{1073741824 L L_{\max}\hat{L}^4\omega^3 p^3\tau}{\bar{L}^6\alpha^4\beta^3 n^3} \leq \frac{1}{326}$$
(250)

$$\frac{1073741824 L^3 L_{\max}\hat{L}^2\omega^2 p^6}{\bar{L}^6\alpha^4\beta^4 n^2\tau^2} \leq \frac{1}{326}$$
(251)

$$\frac{1073741824 L^3 L_{\max}\hat{L}^2\omega^2 p^4}{\bar{L}^6\alpha^4\beta^3 n^2\tau} \leq \frac{1}{326}$$
(252)

$$\frac{1140850688 L L_{\max}\hat{L}^4\omega^3 p^3\tau}{\bar{L}^6\alpha^4\beta^2 n^3} \leq \frac{1}{326}$$
(253)

$$\frac{1140850688 L^3 L_{\max}\hat{L}^2\omega^2 p^4}{\bar{L}^6\alpha^4\beta^2 n^2\tau^2} \leq \frac{1}{326}$$
(254)

$$\frac{2147483648 L L_{\max}\hat{L}^4\omega^3 p^4\tau}{\bar{L}^6\alpha^4\beta^3 n^3} \leq \frac{1}{326}$$
(255)

$$\frac{2147483648 L^3 L_{\max}\hat{L}^2\omega^2 p^5}{\bar{L}^6\alpha^4\beta^3 n^2\tau^2} \leq \frac{1}{326}$$
(256)

$$\frac{2147483648 L^2 L_{\max}\hat{L}^4\omega^3 p^5}{\bar{L}^7\alpha^4\beta^4 n^3\tau} \leq \frac{1}{326}$$
(257)

$$\frac{2147483648 L^2 L_{\max}\hat{L}^4\omega^3 p^3}{\bar{L}^7\alpha^4\beta^2 n^3\tau} \leq \frac{1}{326}$$
(258)

$$\frac{4294967296 L^2 L_{\max}\hat{L}^4\omega^3 p^6}{\bar{L}^7\alpha^4\beta^4 n^3\tau^2} \leq \frac{1}{326}$$
(259)

$$\frac{4294967296 L^2 L_{\max}\hat{L}^4\omega^3 p^4}{\bar{L}^7\alpha^4\beta^3 n^3\tau} \leq \frac{1}{326}$$
(260)

$$\frac{4294967296 L^2 L_{\max}\hat{L}^4\omega^3 p^4}{\bar{L}^7\alpha^4\beta^2 n^3\tau^2} \leq \frac{1}{326}$$
(261)

$$\frac{8589934592 L^2 L_{\max} \hat{L}^4 \omega^3 p^5}{\bar{L}^7 \alpha^4 \beta^3 n^3 \tau^2} \le \frac{1}{326}$$
$$(262)$$

$$\frac{8589934592 L^3 L_{\max} \hat{L}^4 \omega^3 p^5}{\bar{L}^8 \alpha^5 \beta^4 n^3 \tau} \le \frac{1}{326}$$
$$(263)$$

$$\frac{8589934592 L^3 L_{\max} \hat{L}^4 \omega^3 p^3}{\bar{L}^8 \alpha^5 \beta^2 n^3 \tau} \le \frac{1}{326}$$
$$(264)$$

$$\frac{17179869184 L L_{\max} \hat{L}^6 \omega^4 p^5 \tau}{\bar{L}^8 \alpha^5 \beta^4 n^4} \le \frac{1}{326}$$
$$(265)$$

$$\frac{17179869184 L L_{\max} \hat{L}^6 \omega^4 p^3 \tau}{\bar{L}^8 \alpha^5 \beta^2 n^4} \le \frac{1}{326}$$
$$(266)$$

$$\frac{17179869184 L^3 L_{\max} \hat{L}^4 \omega^3 p^6}{\bar{L}^8 \alpha^5 \beta^4 n^3 \tau^2} \le \frac{1}{326}$$
$$(267)$$

$$\frac{17179869184 L^3 L_{\max} \hat{L}^4 \omega^3 p^4}{\bar{L}^8 \alpha^5 \beta^3 n^3 \tau} \le \frac{1}{326}$$
$$(268)$$

$$\frac{17179869184 L^3 L_{\max} \hat{L}^4 \omega^3 p^4}{\bar{L}^8 \alpha^5 \beta^2 n^3 \tau^2} \le \frac{1}{326}$$
$$(269)$$

$$\frac{34359738368 L L_{\max} \hat{L}^6 \omega^4 p^4 \tau}{\bar{L}^8 \alpha^5 \beta^3 n^4} \le \frac{1}{326}$$
$$(270)$$

$$\frac{34359738368 L^3 L_{\max} \hat{L}^4 \omega^3 p^5}{\bar{L}^8 \alpha^5 \beta^3 n^3 \tau^2} \le \frac{1}{326}$$
$$(271)$$

# M Symbolically Computed Expression (92)

$$\frac{2L}{\bar{L}} + \frac{1024 L^3}{\bar{L}^3 \alpha^2} + \frac{4096 L^3 p}{\bar{L}^3 \alpha^2 \tau}$$
$$+ \frac{8192 L^3 p^2}{\bar{L}^3 \alpha^2 \tau^2} + \frac{256 \hat{L}^2 \omega p}{\bar{L}^2 \beta n} + \frac{256 \hat{L}^2 \omega p^2}{\bar{L}^2 \beta^2 n}$$
$$+ \frac{1024 L \hat{L}^2 \omega p}{\bar{L}^3 \alpha \beta n} + \frac{1024 L \hat{L}^2 \omega p^2}{\bar{L}^3 \alpha \beta^2 n} + \frac{8192 L \hat{L}^2 \omega p \tau}{\bar{L}^3 \alpha^2 n}$$
$$+ \frac{131072 L^2 \hat{L}^2 \omega p}{\bar{L}^4 \alpha^2 \beta n} + \frac{131072 L^2 \hat{L}^2 \omega p^2}{\bar{L}^4 \alpha^2 \beta^2 n} + \frac{1048576 L^3 \hat{L}^2 \omega p}{\bar{L}^5 \alpha^3 \beta n}$$
$$+ \frac{1048576 L^3 \hat{L}^2 \omega p^2}{\bar{L}^5 \alpha^3 \beta^2 n} + \frac{1048576 \hat{L}^4 \omega^2 p^2 \tau}{\bar{L}^4 \alpha^2 \beta n^2} + \frac{1048576 \hat{L}^4 \omega^2 p^3 \tau}{\bar{L}^4 \alpha^2 \beta^2 n^2}$$
$$+ \frac{16777216 \hat{L}^4 \omega^2 p^4 \tau}{\bar{L}^4 \alpha^2 \beta^4 n^2} + \frac{16777216 \hat{L}^4 \omega^2 p^2 \tau}{\bar{L}^4 \alpha^2 \beta^2 n^2} + \frac{33554432 \hat{L}^4 \omega^2 p^3 \tau}{\bar{L}^4 \alpha^2 \beta^3 n^2}$$
$$+ \frac{67108864 L^2 \hat{L}^4 \omega^2 p^4}{\bar{L}^6 \alpha^3 \beta^4 n^2} + \frac{67108864 L^2 \hat{L}^4 \omega^2 p^2}{\bar{L}^6 \alpha^3 \beta^2 n^2} + \frac{134217728 L^2 \hat{L}^4 \omega^2 p^3}{\bar{L}^6 \alpha^3 \beta^3 n^2}$$
$$+ \frac{268435456 L^3 \hat{L}^4 \omega^2 p^4}{\bar{L}^7 \alpha^4 \beta^4 n^2} + \frac{268435456 L^3 \hat{L}^4 \omega^2 p^2}{\bar{L}^7 \alpha^4 \beta^2 n^2} + \frac{536870912 \hat{L}^6 \omega^3 p^5 \tau}{\bar{L}^6 \alpha^3 \beta^4 n^3}$$
$$+ \frac{536870912 \hat{L}^6 \omega^3 p^3 \tau}{\bar{L}^6 \alpha^3 \beta^2 n^3} + \frac{536870912 L^3 \hat{L}^4 \omega^2 p^3}{\bar{L}^7 \alpha^4 \beta^3 n^2} + \frac{1073741824 \hat{L}^6 \omega^3 p^4 \tau}{\bar{L}^6 \alpha^3 \beta^3 n^3}$$
$$+ \frac{131072 L \hat{L}^2 \omega p \tau}{\bar{L}^3 \alpha^2 \beta n} + \frac{131072 L \hat{L}^2 \omega p^2 \tau}{\bar{L}^3 \alpha^2 \beta^2 n} + \frac{524288 L^2 \hat{L}^2 \omega p^2}{\bar{L}^4 \alpha^2 \beta n \tau}$$
$$+ \frac{524288 L^2 \hat{L}^2 \omega p^3}{\bar{L}^4 \alpha^2 \beta^2 n \tau} + \frac{1048576 L^2 \hat{L}^2 \omega p^3}{\bar{L}^4 \alpha^2 \beta n \tau^2} + \frac{1048576 L^2 \hat{L}^2 \omega p^4}{\bar{L}^4 \alpha^2 \beta^2 n \tau^2}$$
$$+ \frac{4194304 L^3 \hat{L}^2 \omega p^2}{\bar{L}^5 \alpha^3 \beta n \tau} + \frac{4194304 L^3 \hat{L}^2 \omega p^3}{\bar{L}^5 \alpha^3 \beta^2 n \tau} + \frac{8388608 L^3 \hat{L}^2 \omega p^3}{\bar{L}^5 \alpha^3 \beta n \tau^2}$$

$$+ \frac{8388608 L^3 \hat{L}^2 \omega p^4}{\bar{L}^5 \alpha^3 \beta^2 n \tau^2} + \frac{8388608 L \hat{L}^4 \omega^2 p^2 \tau}{\bar{L}^5 \alpha^3 \beta n^2} + \frac{8388608 L \hat{L}^4 \omega^2 p^3 \tau}{\bar{L}^5 \alpha^3 \beta^2 n^2}$$

$$+ \frac{67108864 L \hat{L}^4 \omega^2 p^4 \tau}{\bar{L}^5 \alpha^3 \beta^4 n^2} + \frac{67108864 L \hat{L}^4 \omega^2 p^2 \tau}{\bar{L}^5 \alpha^3 \beta^2 n^2} + \frac{134217728 L \hat{L}^4 \omega^2 p^3 \tau}{\bar{L}^5 \alpha^3 \beta^3 n^2}$$

$$+ \frac{268435456 L^2 \hat{L}^4 \omega^2 p^5}{\bar{L}^6 \alpha^3 \beta^4 n^2 \tau} + \frac{268435456 L^2 \hat{L}^4 \omega^2 p^3}{\bar{L}^6 \alpha^3 \beta^2 n^2 \tau} + \frac{536870912 L^2 \hat{L}^4 \omega^2 p^6}{\bar{L}^6 \alpha^3 \beta^4 n^2 \tau^2}$$

$$+ \frac{536870912 L^2 \hat{L}^4 \omega^2 p^4}{\bar{L}^6 \alpha^3 \beta^3 n^2 \tau} + \frac{536870912 L^2 \hat{L}^4 \omega^2 p^4}{\bar{L}^6 \alpha^3 \beta^2 n^2 \tau^2} + \frac{1073741824 L^2 \hat{L}^4 \omega^2 p^5}{\bar{L}^6 \alpha^3 \beta^3 n^2 \tau^2}$$

$$+ \frac{1073741824 L^3 \hat{L}^4 \omega^2 p^5}{\bar{L}^7 \alpha^4 \beta^4 n^2 \tau} + \frac{1073741824 L^3 \hat{L}^4 \omega^2 p^3}{\bar{L}^7 \alpha^4 \beta^2 n^2 \tau} + \frac{2147483648 L \hat{L}^6 \omega^3 p^5 \tau}{\bar{L}^7 \alpha^4 \beta^4 n^3}$$

$$+ \frac{2147483648 L \hat{L}^6 \omega^3 p^3 \tau}{\bar{L}^7 \alpha^4 \beta^2 n^3} + \frac{2147483648 L^3 \hat{L}^4 \omega^2 p^6}{\bar{L}^7 \alpha^4 \beta^4 n^2 \tau^2} + \frac{2147483648 L^3 \hat{L}^4 \omega^2 p^4}{\bar{L}^7 \alpha^4 \beta^3 n^2 \tau}$$

$$+ \frac{2147483648 L^3 \hat{L}^4 \omega^2 p^4}{\bar{L}^7 \alpha^4 \beta^2 n^2 \tau^2} + \frac{4294967296 L \hat{L}^6 \omega^3 p^4 \tau}{\bar{L}^7 \alpha^4 \beta^3 n^3} + \frac{4294967296 L^3 \hat{L}^4 \omega^2 p^5}{\bar{L}^7 \alpha^4 \beta^3 n^2 \tau^2} \le 1$$

# N  Symbolically Computed Constraints for $\bar{L}$ Such That The Inequality from Section M Holds

$$\frac{2L}{\bar{L}} \le \frac{1}{114} \qquad (272) \qquad \frac{1024 L^3}{\bar{L}^3 \alpha^2} \le \frac{1}{114} \qquad (273) \qquad \frac{4096 L^3 p}{\bar{L}^3 \alpha^2 \tau} \le \frac{1}{114} \qquad (274)$$

$$\frac{8192 L^3 p^2}{\bar{L}^3 \alpha^2 \tau^2} \le \frac{1}{114} \qquad (275) \qquad \frac{256 \hat{L}^2 \omega p}{\bar{L}^2 \beta n} \le \frac{1}{114} \qquad (276) \qquad \frac{256 \hat{L}^2 \omega p^2}{\bar{L}^2 \beta^2 n} \le \frac{1}{114} \qquad (277)$$

$$\frac{1024 L \hat{L}^2 \omega p}{\bar{L}^3 \alpha \beta n} \le \frac{1}{114} \qquad (278) \qquad \frac{1024 L \hat{L}^2 \omega p^2}{\bar{L}^3 \alpha \beta^2 n} \le \frac{1}{114} \qquad (279) \qquad \frac{8192 L \hat{L}^2 \omega p \tau}{\bar{L}^3 \alpha^2 n} \le \frac{1}{114} \qquad (280)$$

$$\frac{131072 L^2 \hat{L}^2 \omega p}{\bar{L}^4 \alpha^2 \beta n} \le \frac{1}{114} \quad (281) \qquad \frac{131072 L^2 \hat{L}^2 \omega p^2}{\bar{L}^4 \alpha^2 \beta^2 n} \le \frac{1}{114} \quad (282) \qquad \frac{1048576 L^3 \hat{L}^2 \omega p}{\bar{L}^5 \alpha^3 \beta n} \le \frac{1}{114} \quad (283)$$

$$\frac{1048576 L^3 \hat{L}^2 \omega p^2}{\bar{L}^5 \alpha^3 \beta^2 n} \le \frac{1}{114} \quad (284) \qquad \frac{1048576 \hat{L}^4 \omega^2 p^2 \tau}{\bar{L}^4 \alpha^2 \beta n^2} \le \frac{1}{114} \quad (285) \qquad \frac{1048576 \hat{L}^4 \omega^2 p^3 \tau}{\bar{L}^4 \alpha^2 \beta^2 n^2} \le \frac{1}{114} \quad (286)$$

$$\frac{16777216 \hat{L}^4 \omega^2 p^4 \tau}{\bar{L}^4 \alpha^2 \beta^4 n^2} \le \frac{1}{114} \quad (287) \qquad \frac{16777216 \hat{L}^4 \omega^2 p^2 \tau}{\bar{L}^4 \alpha^2 \beta^2 n^2} \le \frac{1}{114} \quad (288) \qquad \frac{33554432 \hat{L}^4 \omega^2 p^3 \tau}{\bar{L}^4 \alpha^2 \beta^3 n^2} \le \frac{1}{114} \quad (289)$$

$$\frac{67108864 L^2 \hat{L}^4 \omega^2 p^4}{\bar{L}^6 \alpha^3 \beta^4 n^2} \le \frac{1}{114} \qquad \frac{67108864 L^2 \hat{L}^4 \omega^2 p^2}{\bar{L}^6 \alpha^3 \beta^2 n^2} \le \frac{1}{114} \qquad \frac{134217728 L^2 \hat{L}^4 \omega^2 p^3}{\bar{L}^6 \alpha^3 \beta^3 n^2} \le \frac{1}{114}$$
$$(290) \qquad\qquad\qquad\qquad (291) \qquad\qquad\qquad\qquad (292)$$

$$\frac{268435456L^3\hat{L}^4\omega^2p^4}{\bar{L}^7\alpha^4\beta^4n^2} \le \frac{1}{114}$$
(293)

$$\frac{268435456L^3\hat{L}^4\omega^2p^2}{\bar{L}^7\alpha^4\beta^2n^2} \le \frac{1}{114}$$
(294)

$$\frac{536870912\hat{L}^6\omega^3p^5\tau}{\bar{L}^6\alpha^3\beta^4n^3} \le \frac{1}{114}$$
(295)

$$\frac{536870912\hat{L}^6\omega^3p^3\tau}{\bar{L}^6\alpha^3\beta^2n^3} \le \frac{1}{114}$$
(296)

$$\frac{536870912L^3\hat{L}^4\omega^2p^3}{\bar{L}^7\alpha^4\beta^3n^2} \le \frac{1}{114}$$
(297)

$$\frac{1073741824\hat{L}^6\omega^3p^4\tau}{\bar{L}^6\alpha^3\beta^3n^3} \le \frac{1}{114}$$
(298)

$$\frac{131072L\hat{L}^2\omega p\tau}{\bar{L}^3\alpha^2\beta n} \le \frac{1}{114}$$
(299)

$$\frac{131072L\hat{L}^2\omega p^2\tau}{\bar{L}^3\alpha^2\beta^2 n} \le \frac{1}{114}$$
(300)

$$\frac{524288L^2\hat{L}^2\omega p^2}{\bar{L}^4\alpha^2\beta n\tau} \le \frac{1}{114}$$
(301)

$$\frac{524288L^2\hat{L}^2\omega p^3}{\bar{L}^4\alpha^2\beta^2 n\tau} \le \frac{1}{114}$$
(302)

$$\frac{1048576L^2\hat{L}^2\omega p^3}{\bar{L}^4\alpha^2\beta n\tau^2} \le \frac{1}{114}$$
(303)

$$\frac{1048576L^2\hat{L}^2\omega p^4}{\bar{L}^4\alpha^2\beta^2 n\tau^2} \le \frac{1}{114}$$
(304)

$$\frac{4194304L^3\hat{L}^2\omega p^2}{\bar{L}^5\alpha^3\beta n\tau} \le \frac{1}{114}$$
(305)

$$\frac{4194304L^3\hat{L}^2\omega p^3}{\bar{L}^5\alpha^3\beta^2 n\tau} \le \frac{1}{114}$$
(306)

$$\frac{8388608L^3\hat{L}^2\omega p^3}{\bar{L}^5\alpha^3\beta n\tau^2} \le \frac{1}{114}$$
(307)

$$\frac{8388608L^3\hat{L}^2\omega p^4}{\bar{L}^5\alpha^3\beta^2 n\tau^2} \le \frac{1}{114}$$
(308)

$$\frac{8388608L\hat{L}^4\omega^2p^2\tau}{\bar{L}^5\alpha^3\beta n^2} \le \frac{1}{114}$$
(309)

$$\frac{8388608L\hat{L}^4\omega^2p^3\tau}{\bar{L}^5\alpha^3\beta^2 n^2} \le \frac{1}{114}$$
(310)

$$\frac{67108864L\hat{L}^4\omega^2p^4\tau}{\bar{L}^5\alpha^3\beta^4n^2} \le \frac{1}{114}$$
(311)

$$\frac{67108864L\hat{L}^4\omega^2p^2\tau}{\bar{L}^5\alpha^3\beta^2n^2} \le \frac{1}{114}$$
(312)

$$\frac{134217728L\hat{L}^4\omega^2p^3\tau}{\bar{L}^5\alpha^3\beta^3n^2} \le \frac{1}{114}$$
(313)

$$\frac{268435456L^2\hat{L}^4\omega^2p^5}{\bar{L}^6\alpha^3\beta^4n^2\tau} \le \frac{1}{114}$$
(314)

$$\frac{268435456L^2\hat{L}^4\omega^2p^3}{\bar{L}^6\alpha^3\beta^2n^2\tau} \le \frac{1}{114}$$
(315)

$$\frac{536870912L^2\hat{L}^4\omega^2p^6}{\bar{L}^6\alpha^3\beta^4n^2\tau^2} \le \frac{1}{114}$$
(316)

$$\frac{536870912L^2\hat{L}^4\omega^2p^4}{\bar{L}^6\alpha^3\beta^3n^2\tau} \le \frac{1}{114}$$
(317)

$$\frac{536870912L^2\hat{L}^4\omega^2p^4}{\bar{L}^6\alpha^3\beta^2n^2\tau^2} \le \frac{1}{114}$$
(318)

$$\frac{1073741824L^2\hat{L}^4\omega^2p^5}{\bar{L}^6\alpha^3\beta^3n^2\tau^2} \le \frac{1}{114}$$
(319)

$$\frac{1073741824L^3\hat{L}^4\omega^2p^5}{\bar{L}^7\alpha^4\beta^4n^2\tau} \le \frac{1}{114}$$
(320)

$$\frac{1073741824L^3\hat{L}^4\omega^2p^3}{\bar{L}^7\alpha^4\beta^2n^2\tau} \le \frac{1}{114}$$
(321)

$$\frac{2147483648L\hat{L}^6\omega^3p^5\tau}{\bar{L}^7\alpha^4\beta^4n^3} \le \frac{1}{114}$$
(322)

$$\frac{2147483648 L \hat{L}^6 \omega^3 p^3 \tau}{\bar{L}^7 \alpha^4 \beta^2 n^3} \le \frac{1}{114} \qquad (323)$$

$$\frac{2147483648 L^3 \hat{L}^4 \omega^2 p^6}{\bar{L}^7 \alpha^4 \beta^4 n^2 \tau^2} \le \frac{1}{114} \qquad (324)$$

$$\frac{2147483648 L^3 \hat{L}^4 \omega^2 p^4}{\bar{L}^7 \alpha^4 \beta^3 n^2 \tau} \le \frac{1}{114} \qquad (325)$$

$$\frac{2147483648 L^3 \hat{L}^4 \omega^2 p^4}{\bar{L}^7 \alpha^4 \beta^2 n^2 \tau^2} \le \frac{1}{114} \qquad (326)$$

$$\frac{4294967296 L \hat{L}^6 \omega^3 p^4 \tau}{\bar{L}^7 \alpha^4 \beta^3 n^3} \le \frac{1}{114} \qquad (327)$$

$$\frac{4294967296 L^3 \hat{L}^4 \omega^2 p^5}{\bar{L}^7 \alpha^4 \beta^3 n^2 \tau^2} \le \frac{1}{114} \qquad (328)$$

## O  Symbolical Check That The Constraints from Sections I, J, L and N Follow From The Constraint (44)

Note that the inequalities from Lemma F.1 follow from (44). Therefore, the inequalities from Sections I, J, L and N follow from (44), if they follow from the inequalities from Lemma F.1. We now present it[7]. These results are checked and generated using the script in Section P (see Section 6 in Section P)

(93) follows from (74), (74).
(94) follows from (73), (73).
(95) follows from (77), (77).
(96) follows from (78), (77).
(97) follows from (78), (78).
(98) follows from (72), (72).
(99) follows from (71), (71).
(100) follows from (70), (70).
(101) follows from (77), (77), (74), (74).
(102) follows from (77), (77), (73), (73).
(103) follows from (74), (74), (72), (72).
(104) follows from (73), (73), (72), (72).
(105) follows from (78), (77), (74), (74).
(106) follows from (78), (77), (73), (73).
(107) follows from (78), (78), (74), (74).
(108) follows from (78), (78), (73), (73).
(109) follows from (77), (77).
(110) follows from (77), (77), (77), (77).
(111) follows from (66).
(112) follows from (78), (77).
(113) follows from (78), (78).
(114) follows from (78), (77), (77), (77).
(115) follows from (78), (78), (77), (77).
(116) follows from (77), (66).
(117) follows from (65).
(118) follows from (64).
(119) follows from (77), (77), (77), (66).

---

[7] "(a) follows from (b),(c)" means that one should use (b) and (c) to get (a). It is possible that $b = c$, thus one should apply $b\ (= c)$ two times.

(120) follows from (77), (65).

(121) follows from (77), (64).

(122) follows from (74), (74), (66).

(123) follows from (74), (74), (65).

(124) follows from (69), (69).

(125) follows from (78), (77), (77), (66).

(126) follows from (78), (78), (77), (66).

(127) follows from (77), (74), (74).

(128) follows from (77), (73), (73).

(129) follows from (77), (77), (72), (72).

(130) follows from (77), (77), (77), (65).

(131) follows from (77), (77), (77), (64).

(132) follows from (77), (77), (74), (74).

(133) follows from (77), (77), (73), (73).

(134) follows from (77), (77), (77), (74), (74).

(135) follows from (77), (77), (77), (73), (73).

(136) follows from (77), (77), (77), (77), (74), (74).

(137) follows from (77), (77), (77), (77), (73), (73).

(138) follows from (77), (77), (77), (73), (73), (73), (73).

(139) follows from (77), (77), (77), (74), (74), (74), (74).

(140) follows from (77), (77), (77), (74), (74), (73), (73).

(141) follows from (77), (77), (77), (77), (73), (73), (73), (73).

(142) follows from (77), (77), (77), (77), (74), (74), (74), (74).

(143) follows from (77), (77), (77), (77), (74), (74), (73), (73).

(144) follows from (77), (74), (74), (66).

(145) follows from (77), (74), (74), (65).

(146) follows from (68), (68).

(147) follows from (67), (67).

(148) follows from (77), (72), (72), (66).

(149) follows from (78), (74), (74).

(150) follows from (78), (73), (73).

(151) follows from (81), (81), (81).

(152) follows from (80), (80), (80).

(153) follows from (72), (72), (65).

(154) follows from (72), (72), (64).

(155) follows from (78), (77), (77), (65).

(156) follows from (78), (77), (77), (64).

(157) follows from (78), (77), (74), (74).

(158) follows from (78), (77), (73), (73).

(159) follows from (77), (77), (74), (74), (66).

(160) follows from (78), (78), (77), (65).

(161) follows from (78), (78), (77), (64).

(162) follows from (77), (77), (71), (71).

(163) follows from (77), (77), (70), (70).

(164) follows from (78), (78), (74), (74).

(165) follows from (78), (78), (73), (73).

(166) follows from (78), (77), (77), (74), (74).

(167) follows from (78), (77), (77), (73), (73).

(168) follows from (77), (77), (77), (74), (74), (66).

(169) follows from (70), (70), (64).

(170) follows from (71), (71), (65).

(171) follows from (74), (74), (72), (72), (66).

(172) follows from (78), (78), (77), (74), (74).

(173) follows from (78), (78), (77), (73), (73).

(174) follows from (77), (74), (74), (72), (72).

(175) follows from (77), (73), (73), (72), (72).

(176) follows from (71), (71), (64).

(177) follows from (77), (77), (73), (73), (64).

(178) follows from (77), (77), (74), (74), (65).

(179) follows from (78), (77), (77), (77), (74), (74).

(180) follows from (78), (77), (77), (77), (73), (73).

(181) follows from (77), (77), (74), (74), (64).

(182) follows from (78), (78), (77), (77), (74), (74).

(183) follows from (78), (78), (77), (77), (73), (73).

(184) follows from (74), (74), (71), (71), (64).

(185) follows from (74), (74), (71), (71), (66).

(186) follows from (77), (77), (77), (73), (73), (64).

(187) follows from (77), (77), (74), (74), (72), (72).

(188) follows from (79), (78), (71), (71), (71), (71).

(189) follows from (77), (77), (77), (74), (74), (65).

(190) follows from (77), (73), (73), (70), (70).

(191) follows from (77), (74), (74), (71), (71).

(192) follows from (73), (73), (72), (72), (64).

(193) follows from (74), (74), (71), (71), (65).

(194) follows from (77), (77), (77), (74), (74), (64).

(195) follows from (74), (74), (72), (72), (65).

(196) follows from (77), (74), (74), (70), (70).

(197) follows from (74), (74), (72), (72), (64).

(198) follows from (77), (77), (74), (74), (74), (74), (64).

(199) follows from (77), (77), (74), (74), (74), (74), (66).

(200) follows from (77), (77), (74), (74), (74), (74), (65).

(201) follows from (77), (77), (73), (73), (70), (70).

(202) follows from (77), (77), (74), (74), (71), (71).

(203) follows from (77), (77), (77), (74), (74), (74), (74), (64).

(204) follows from (77), (77), (77), (74), (74), (74), (74), (66).

(205) follows from (77), (77), (74), (74), (70), (70).

(206) follows from (74), (74), (74), (74), (72), (72), (64).

(207) follows from (74), (74), (74), (74), (72), (72), (66).

(208) follows from (77), (77), (77), (74), (74), (74), (74), (65).

(209) follows from (78), (77), (77), (73), (73), (73), (73).

(210) follows from (78), (77), (77), (74), (74), (74), (74).

(211) follows from (78), (76), (76), (71), (71), (70), (70).

(212) follows from (77), (74), (74), (74), (74), (72), (72).

(213) follows from (74), (74), (74), (74), (72), (72), (65).

(214) follows from (78), (78), (77), (73), (73), (73), (73).

(215) follows from (78), (77), (77), (74), (74), (73), (73).

(216) follows from (78), (78), (77), (74), (74), (74), (74).

(217) follows from (78), (76), (76), (71), (71), (71), (71).

(218) follows from (78), (78), (77), (74), (74), (73), (73).

(219) follows from (78), (77), (77), (77), (73), (73), (73), (73).

(220) follows from (78), (77), (77), (77), (74), (74), (74), (74).

(221) follows from (79), (78), (74), (74), (71), (71), (70), (70).

(222) follows from (77), (77), (74), (74), (74), (74), (72), (72).

(223) follows from (78), (78), (77), (77), (73), (73), (73), (73).

(224) follows from (78), (77), (77), (77), (74), (74), (73), (73).

(225) follows from (78), (78), (77), (77), (74), (74), (74), (74).

(226) follows from (79), (78), (74), (74), (71), (71), (71), (71).

(227) follows from (78), (78), (77), (77), (74), (74), (73), (73).

(228) follows from (77), (71), (71), (66).

(229) follows from (77), (72), (72), (65).

(230) follows from (77), (72), (72), (64).

(231) follows from (78), (77), (74), (74), (66).

(232) follows from (78), (78), (74), (74), (66).

(233) follows from (77), (70), (70), (64).

(234) follows from (78), (77), (77), (74), (74), (66).

(235) follows from (77), (71), (71), (65).

(236) follows from (77), (71), (71), (64).

(237) follows from (77), (74), (74), (72), (72), (66).

(238) follows from (78), (78), (77), (74), (74), (66).

(239) follows from (78), (77), (73), (73), (64).

(240) follows from (78), (77), (74), (74), (65).

(241) follows from (78), (78), (73), (73), (64).

(242) follows from (78), (77), (74), (74), (64).

(243) follows from (78), (78), (74), (74), (65).

(244) follows from (77), (74), (74), (71), (71), (64).

(245) follows from (77), (74), (74), (71), (71), (66).

(246) follows from (78), (78), (74), (74), (64).

(247) follows from (78), (77), (77), (73), (73), (64).

(248) follows from (78), (77), (77), (74), (74), (65).

(249) follows from (77), (73), (73), (72), (72), (64).

(250) follows from (77), (74), (74), (71), (71), (65).

(251) follows from (78), (78), (77), (73), (73), (64).

(252) follows from (78), (77), (77), (74), (74), (64).

(253) follows from (77), (74), (74), (72), (72), (65).

(254) follows from (78), (78), (77), (74), (74), (65).

(255) follows from (77), (74), (74), (72), (72), (64).

(256) follows from (78), (78), (77), (74), (74), (64).

(257) follows from (78), (77), (74), (74), (74), (74), (64).

(258) follows from (78), (77), (74), (74), (74), (74), (66).

(259) follows from (78), (78), (74), (74), (74), (74), (64).

(260) follows from (78), (77), (74), (74), (74), (74), (65).

(261) follows from (78), (78), (74), (74), (74), (74), (66).

(262) follows from (78), (78), (74), (74), (74), (74), (65).

(263) follows from (78), (77), (77), (74), (74), (74), (74), (64).

(264) follows from (78), (77), (77), (74), (74), (74), (74), (66).

(265) follows from (77), (74), (74), (74), (74), (72), (72), (64).

(266) follows from (77), (74), (74), (74), (74), (72), (72), (66).

(267) follows from (78), (78), (77), (74), (74), (74), (74), (64).

(268) follows from (78), (77), (77), (74), (74), (74), (74), (65).

(269) follows from (78), (78), (77), (74), (74), (74), (74), (66).

(270) follows from (77), (74), (74), (74), (74), (72), (72), (65).

(271) follows from (78), (78), (77), (74), (74), (74), (74), (65).

(272) follows from (79).

(273) follows from (79), (77), (77).

(274) follows from (79), (78), (77).

(275) follows from (79), (78), (78).

(276) follows from (76), (76).

(277) follows from (75), (75).

(278) follows from (79), (74), (74).

(279) follows from (79), (73), (73).

(280) follows from (79), (72), (72).

(281) follows from (79), (77), (74), (74).

(282) follows from (79), (77), (73), (73).

(283) follows from (79), (77), (77), (74), (74).

(284) follows from (79), (77), (77), (73), (73).

(285) follows from (76), (76), (72), (72).

(286) follows from (75), (75), (72), (72).

(287) follows from (75), (75), (70), (70).

(288) follows from (76), (76), (71), (71).

(289) follows from (76), (76), (70), (70).

(290) follows from (79), (77), (73), (73), (73), (73).

(291) follows from (79), (77), (74), (74), (74), (74).

(292) follows from (79), (77), (74), (74), (73), (73).

(293) follows from (79), (77), (77), (73), (73), (73), (73).

(294) follows from (79), (77), (77), (74), (74), (74), (74).

(295) follows from (75), (75), (73), (73), (72), (72).

(296) follows from (76), (76), (74), (74), (72), (72).

(297) follows from (79), (77), (77), (74), (74), (73), (73).

(298) follows from (76), (76), (73), (73), (72), (72).

(299) follows from (79), (71), (71).

(300) follows from (79), (70), (70).

(301) follows from (79), (78), (74), (74).

(302) follows from (79), (78), (73), (73).

(303) follows from (78), (78), (76), (76).

(304) follows from (78), (78), (75), (75).

(305) follows from (79), (78), (77), (74), (74).

(306) follows from (79), (78), (77), (73), (73).

(307) follows from (79), (78), (78), (74), (74).

(308) follows from (79), (78), (78), (73), (73).

(309) follows from (79), (74), (74), (72), (72).

(310) follows from (79), (73), (73), (72), (72).

(311) follows from (79), (73), (73), (70), (70).

(312) follows from (79), (74), (74), (71), (71).

(313) follows from (79), (74), (74), (70), (70).

(314) follows from (79), (78), (73), (73), (73), (73).

(315) follows from (79), (78), (74), (74), (74), (74).

(316) follows from (78), (78), (75), (75), (73), (73).

(317) follows from (79), (78), (74), (74), (73), (73).

(318) follows from (78), (78), (76), (76), (74), (74).

(319) follows from (78), (78), (76), (76), (73), (73).

(320) follows from (79), (78), (77), (73), (73), (73), (73).

(321) follows from (79), (78), (77), (74), (74), (74), (74).

(322) follows from (79), (73), (73), (73), (73), (72), (72).

(323) follows from (79), (74), (74), (74), (74), (72), (72).

(324) follows from (79), (78), (78), (73), (73), (73), (73).

(325) follows from (79), (78), (77), (74), (74), (73), (73).

(326) follows from (79), (78), (78), (74), (74), (74), (74).

(327) follows from (79), (74), (74), (73), (73), (72), (72).

(328) follows from (79), (78), (78), (74), (74), (73), (73).

# Jupyter Notebook for Symbolic Computations

## 1  Import Necessary Libraries

```python
import os

from IPython.display import display

import sympy
from sympy import Symbol

from utils import FileWriter, ConstraintsAggregator
from utils import get_factors, get_term, search_factors,␣
 ↪latex_repres_of_inequality
```

## 2  Initialize a File for Results

```python
file_path = '../paper/result.txt'
if os.path.exists(file_path):
    os.remove(file_path)
fw = FileWriter(file_path)
```

## 3  Initialize Symbols From the Paper

```python
rho = Symbol('rho', nonnegative=True)
kappa = Symbol('kappa', nonnegative=True)
lmbda = Symbol('lambda', nonnegative=True)

omega = Symbol('omega', nonnegative=True)
p = Symbol('p', positive=True)
tau = Symbol('tau', positive=True)
beta = Symbol('beta', positive=True)
alpha = Symbol('alpha', positive=True)
n = Symbol('n', positive=True)

L = Symbol('L', positive=True)
L_hat = Symbol(r'\hat{L}', positive=True)
```

```
L_max = Symbol(r'L_{\max}', positive=True)

L_bar = Symbol(r'\bar{L}', positive=True)
```

## 4 Assistant for "First Symbolically Computed" Lemma

### 4.1 Calculate Expressions

```
[ ]: nu_hat_exp = (8 / alpha) * (p * L / 2
                            + kappa * 4 * p * (1 + p / beta) * L_hat**2
                            + rho * 4 * p * L**2
                            + rho * (8 * p / tau) * (2 * p * (1 + 2 * p / tau) *␣
    ↪L**2
                                                    + 4 * p * tau**2 * omega *␣
    ↪L_hat**2 / n))
    nu_hat = Symbol(r'\hat{\nu}', nonnegative=True)
    # The left hand sides of the inequalities
    rho_lhs = nu_hat * (8 / (alpha * p * L_bar ** 2))
    kappa_lhs = ((8 * p * omega) / (n * L_bar * beta)
                + nu_hat * (8 * omega) / (n * beta * L_bar**2)
                + rho * (8 * p) / tau * (8 * tau ** 2 * omega) / (n * beta))
```

### 4.2 Display Them

```
[ ]: display(nu_hat_exp)
    display(rho_lhs)
    display(kappa_lhs)
```

### 4.3 Symbolically Calculate The Steps From The Proof

```
[ ]: constraints_agg = ConstraintsAggregator()

    rho_lhs = rho_lhs.subs(nu_hat, nu_hat_exp)
    kappa_lhs = kappa_lhs.subs(nu_hat, nu_hat_exp)

    # Group Terms w.r.t. kappa
    kappa_lhs_poly = sympy.poly(kappa_lhs, kappa)
    kappa_coef = kappa_lhs_poly.all_coeffs()
    kappa_lhs = kappa_lhs.expand().collect(kappa)
    fw.write(sympy.latex(kappa_lhs) + ", \label{eq:kappa_expand}")

    # Find Conditions When The Coefficients Near Kappa <= 1/2
    terms = sympy.expand(kappa_coef[0]).args
    latex_string = constraints_agg.add_constraints(terms)
    fw.write(latex_string)
```

```python
# Define kappa
kappa_solution = (2 * kappa_coef[1]).expand()
fw.write("\kappa \eqdef " + sympy.latex(kappa_solution) + ". \label{eq:
 ↪kappa_sol}")

# Group Terms w.r.t. rho
rho_lhs = rho_lhs.subs(kappa, kappa_solution)
rho_lhs = rho_lhs.expand().collect(rho)
fw.write(sympy.latex(rho_lhs) + ", \label{eq:rho_expand}")

# Find Conditions When The Coefficients Near rho <= 1/2
rho_lhs_poly = sympy.poly(rho_lhs, rho)
rho_coef = rho_lhs_poly.all_coeffs()
terms = sympy.expand(rho_coef[0]).args
latex_string = constraints_agg.add_constraints(terms)
fw.write(latex_string)

# Define rho
rho_solution = (2 * rho_coef[1]).expand()
fw.write("\\rho \eqdef " + sympy.latex(rho_solution) + ". \label{eq:rho_sol}")
```

## 5 Assistant for "Second Symbolically Computed" Lemma

### 5.1 First Inequality

```python
[ ]: bregman_coef = ((4 * omega * L_max) / (n * L_bar)
                + kappa * 8 * (1 + p / beta) * L_max
                + (nu_hat / L_bar**2) * (4 * omega * L_max / (p * n) + 8 * L /
 ↪(p * alpha))
                + rho * 8 * L
                + lmbda * (4 * (1 + (2 * p) / tau) * L + 8 * tau**2 * omega *
 ↪L_max / n))
display(bregman_coef)

lmbda_solution = rho * 8 * p / tau

# Substitue all known expressions to the term
bregman_coef = \
    bregman_coef.subs(nu_hat, nu_hat_exp).subs(lmbda, lmbda_solution).
 ↪subs(kappa, kappa_solution).subs(rho, rho_solution)
bregman_coef = bregman_coef.expand().simplify().expand()
latex_string = latex_repres_of_inequality(bregman_coef)
fw.write(latex_string)

# Find Conditions When bregman_coef is Less or Equal <= 1/2
terms = bregman_coef.args
```

```
latex_string = constraints_agg.add_constraints(terms)
fw.write(latex_string)
```

## 5.2   Second Inequality

```
[ ]: dist_coef = (4 / L_bar) * (L / 2
                              + kappa * 4 * (1 + p / beta) * L_hat ** 2
                              + rho * 4 * L**2
                              + lmbda * (2 * (1 + 2 * p / tau) * L**2 + 4 * tau**2 *⊔
      ↪omega * L_hat**2 / n))
     display(dist_coef)

     dist_coef = \
         dist_coef.subs(lmbda, lmbda_solution).subs(kappa, kappa_solution).subs(rho,⊔
      ↪rho_solution)
     dist_coef = dist_coef.expand().simplify().expand()
     latex_string = latex_repres_of_inequality(dist_coef, rhs="1")
     fw.write(latex_string)
     terms = dist_coef.args
     latex_string = constraints_agg.add_constraints(terms)
     fw.write(latex_string)
```

# 6   Check That The Constraints Follow From The Inequalities from The "Auxillary Inequalities" Lemma.

## 6.1   The Inequalities from The "Auxillary Inequalities" Lemma:

```
[ ]: from collections import OrderedDict
     _contstant = 1
     proposals = OrderedDict([
         ("eq:lipt:max_2",         (L_max * omega * p**2) / (beta**2 * n)),
         ("eq:lipt:max_1",         (L_max * omega * p) / (beta * n)),
         ("eq:lipt:max",           (L_max * omega) / (n)),
         ("eq:lipt:l_l_max_2",     (sympy.sqrt(L * L_max) * p * sympy.sqrt(omega *⊔
      ↪tau)) / (alpha * beta * sympy.sqrt(n))),
         ("eq:lipt:l_l_max_1",     (sympy.sqrt(L * L_max) * sympy.sqrt(p * omega *⊔
      ↪tau)) / (alpha * sympy.sqrt(beta * n))),
         ("eq:lipt:l_l_max_p",     (sympy.sqrt(L * L_max) * sympy.sqrt(p * omega *⊔
      ↪tau)) / (alpha * sympy.sqrt(n))),
         ("eq:lipt:hat_2",         (L_hat * p * sympy.sqrt(omega * tau)) / (alpha *⊔
      ↪beta * sympy.sqrt(n))),
         ("eq:lipt:hat_1",         (L_hat * sympy.sqrt(p * omega * tau)) / (alpha *⊔
      ↪sympy.sqrt(beta * n))),
         ("eq:lipt:hat_p",         (L_hat * sympy.sqrt(p * omega * tau)) / (alpha *⊔
      ↪sympy.sqrt(n))),
```

```python
    ("eq:lipt:hat_alpha_2",   (L_hat * p * sympy.sqrt(omega)) / (beta * sympy.
 ↪sqrt(alpha * n))),
    ("eq:lipt:hat_alpha_1",   (L_hat * sympy.sqrt(p * omega)) / (sympy.sqrt(beta
 ↪* alpha * n))),
    ("eq:lipt:hat_no_alpha_2",(L_hat * p * sympy.sqrt(omega)) / (beta * sympy.
 ↪sqrt(n))),
    ("eq:lipt:hat_no_alpha_1",(L_hat * sympy.sqrt(p * omega)) / (sympy.sqrt(beta
 ↪* n))),
    ("eq:lipt:plain",          (L) / (alpha)),
    ("eq:lipt:plain_p_alpha", (L * p) / (alpha * tau)),
    ("eq:lipt:plain_no_alpha",(L)),
    ("eq:lipt:double_lipt_2", sympy.cbrt((L * L_hat**2 * omega * p**4) /
 ↪(alpha**2 * beta**2 * n * tau**2))),
    ("eq:lipt:double_lipt_1", sympy.cbrt((L * L_hat**2 * omega * p**3) /
 ↪(alpha**2 * beta * n * tau**2))),
])
const = 660508
for k, proposal in proposals.items():
    proposals[k] = (const * proposal / L_bar).expand()
proposals_factors = [get_factors(proposal) for _, proposal in proposals.items()]
```

## 6.2   Search The Right Inequalities for Each Constraint

```python
# Takes ~1 hour on a laptop
import tqdm
constraints = constraints_agg.get_constraints()
constraints_inequalities = []
for i, factor in enumerate(tqdm.tqdm(constraints)):
    num_of_base_factors_to_use = -int(factor[L_bar])
    path = search_factors(factor, num_of_base_factors_to_use, proposals_factors,
                          pos_hints=[L, L_max, L_hat, omega, p],
                          neg_hints=[alpha, n])
    constraints_inequalities.append(path)
    assert path is not None
```

```python
text = constraints_agg.prepare_text_from_proposals(proposals,
 ↪constraints_inequalities)
fw.write(text)
```

### P.1 File utils.py

```
 1  import os
 2  from collections import defaultdict
 3  from copy import copy
 4
 5  import sympy
 6
 7
 8  class _defaultdictwithconst(defaultdict):
 9      def __init__(self, *args, **kwargs):
10          super(_defaultdictwithconst, self).__init__(*args, **kwargs)
11          self.const = None
12
13      def __copy__(self):
14          obj = super(_defaultdictwithconst, self).__copy__()
15          obj.const = self.const
16          return obj
17
18
19  def get_factors(term):
20      """ Converts a sympy expression with the sympy.Mul type to a dictionary with
      factors.
21          Ex: A^2 / B^(1/2) -> {A: 2, B: -1/2}
22      Args:
23          param1: sympy expression (sympy.Mul).
24      """
25      factors = _defaultdictwithconst(int)
26      factors.const = 1
27      const_assigned = False
28      assert isinstance(term, sympy.Mul)
29      for el in term.args:
30          free_symbols = list(el.free_symbols)
31          if len(free_symbols) == 0:
32              assert not const_assigned
33              const_assigned = True
34              factors.const = int(el)
35              continue
36          assert len(free_symbols) == 1
37          power = el.as_powers_dict()[free_symbols[0]]
38          assert isinstance(power, sympy.Rational)
39          factors[free_symbols[0]] = power
40      return factors
41
42
43  def get_term(factors):
44      """ The inverse function to the get_factors function: converts factors to a term
45          Ex: {A: 2, B: -1/2} -> A^2 / B^(1/2)
46      Args:
47          param1: dictionary with factors
48      """
49      result = factors.const
50      for k, v in factors.items():
51          result = result * k ** v
52      return result
53
54
55  def search_factors(factors, num_of_base_factors_to_use, base_factors, pos_hints=[],
      neg_hints=[]):
56      """ Checks if it possible to decompose 'factors' using num_of_base_factors
      factors from 'base_factors'
57      Args:
58          param1: dictionary with factors
59          param2: number of base factors to use
60          param3: list of dictionaries with factors
```

```
61          param4 and param5: hints to the algorithm that helps to improve performance
62      Returns:
63          A list with indices (with repetitions) of 'base_factors' that compose '
    factors'.
64          If it not possible, then the function returns None.
65      """
66      def _check_hints(factors_):
67          for k in pos_hints:
68              assert factors_[k] >= 0
69          for k in neg_hints:
70              assert factors_[k] <= 0
71      _check_hints(factors)
72      for base in base_factors:
73          _check_hints(base)
74      def _search_factors(factors, num_of_base_factors_to_use, path, base_factors):
75          if num_of_base_factors_to_use == 0:
76              return factors.const <= 1 and all([factors[k] == 0 for k in factors])
77          for index, choice in enumerate(base_factors):
78              factor_choice = copy(factors)
79              factor_choice.const = factor_choice.const / choice.const
80              for k, v in choice.items():
81                  factor_choice[k] = factor_choice[k] - v
82              skip = False
83              for k in pos_hints:
84                  if factor_choice[k] < 0:
85                      skip = True
86                      break
87              for k in neg_hints:
88                  if factor_choice[k] > 0:
89                      skip = True
90                      break
91              if not skip and _search_factors(factor_choice, num_of_base_factors_to_use
    - 1, path, base_factors):
92                  path.append(index)
93                  return True
94          return False
95      path = []
96      if _search_factors(factors, num_of_base_factors_to_use, path, base_factors):
97          return path
98      else:
99          return None
100
101
102 class FileWriter(object):
103     def __init__(self, file_path):
104         self._file_path = file_path
105         assert not os.path.exists(file_path)
106
107     def write(self, text):
108         with open(self._file_path, "a") as fd:
109             fd.write("{}\n".format(text))
110
111
112 class ConstraintsAggregator(object):
113     def __init__(self):
114         self._constraints = []
115
116     def get_constraints(self):
117         return self._constraints
118
119     def add_constraints(self, terms):
120         denom_constant = self._denom_constant(terms)
121         terms_global_index = []
122         for term in terms:
123             terms_global_index.append(len(self._constraints))
```

```python
124                self._constraints.append(get_factors(term * denom_constant))
125            return self._prepare_text(terms, terms_global_index)
126
127        def prepare_text_from_proposals(self, proposals, constraints_inequalities):
128            # assert len(constraints_inequalities) == len(self._constraints)
129            text = ''
130            proposals_labels = list(proposals.keys())
131            for global_index, ineq in enumerate(constraints_inequalities):
132                constraint_label = self._get_label(global_index)
133                text += "& \eqref{{{}}}".format(constraint_label) + " \\textnormal{
        follows from } "
134                for i, index_proposal in enumerate(ineq):
135                    text += "\eqref{{{}}}".format(proposals_labels[index_proposal])
136                    if i == len(ineq) - 1:
137                        text += "."
138                    else:
139                        text += ","
140                text += "\\\\"
141            return text
142
143        def _denom_constant(self, terms):
144            return 2 * len(terms)
145
146        def _get_label(self, global_index):
147            return "eq:sympy:constraints:{}".format(global_index)
148
149        def _prepare_text(self, terms, terms_global_index):
150            text = ''
151            denom_constant = self._denom_constant(terms)
152            num_per_column = 3
153            for i, (global_index, term) in enumerate(zip(terms_global_index, terms)):
154                if i % num_per_column == 0:
155                    text += r"\begin{tabularx}{1.2\linewidth}{XXX}"
156                text += r"\begin{equation}"
157                text += sympy.latex(term) + " \leq \\frac{{1}}{{{}}}".format(
        denom_constant)
158                text += "\label{{{}}}".format(self._get_label(global_index))
159                text += r"\end{equation}"
160                if i % num_per_column == (num_per_column - 1):
161                    text += r"\end{tabularx} "
162                else:
163                    text += "&"
164            def ceildiv(a, b):
165                return -(a // -b)
166            for i in range(len(terms), ceildiv(len(terms), num_per_column) *
        num_per_column):
167                if i % num_per_column == (num_per_column - 1):
168                    text += r"\end{tabularx} "
169                else:
170                    text += "&"
171            return text
172
173
174    def latex_repres_of_inequality(coef, rhs="\\frac{{3}}{{4}}"):
175        latex_string = "&"
176        for i, term in enumerate(coef.args):
177            latex_string = latex_string + sympy.latex(term)
178            if i == len(coef.args) - 1:
179                latex_string += " \leq " + rhs
180            else:
181                if i % 3 == 2:
182                    latex_string += " \\\\ & +"
183                else:
184                    latex_string += " + "
185        return latex_string
```

## Q Experiments

### Q.1 Setup

We now conduct experiments on the practical logistic regression task with LIBSVM datasets (Chang and Lin, 2011) (under the 3-clause BSD license). The experiments were implemented in Python 3.7.9. The distributed environment was emulated on machines with Intel(R) Xeon(R) Gold 6248 CPU @ 2.50GHz. In each plot we show the relations between the total number of coordinates transmitted from and to the server and function values. The parameters of the algorithms are taken as suggested by the corresponding theory, except for the stepsizes that we fine-tune from a set $\{2^i \mid i \in [-20, 20]\}$. For 2Direction, we use parameters from Theorem 5.2 and finetune the step size $L$.

We solve the logistic regression problem:

$$f_i(x_1, \ldots, x_c) := -\frac{1}{m} \sum_{j=1}^{m} \log \left( \frac{\exp\left(a_{ij}^\top x_{y_{ij}}\right)}{\sum_{y=1}^{c} \exp\left(a_{ij}^\top x_y\right)} \right),$$

where $x_1, \ldots, x_c \in \mathbb{R}^d$, $c$ is the number of unique labels, $a_{ij} \in \mathbb{R}^d$ is a feature of a sample on the $i^{\text{th}}$ worker, $y_{ij}$ is a corresponding label and $m$ is the number of samples located on the $i^{\text{th}}$ worker. The Rand$K$ compressor is used to compress information from the workers to the server, the Top$K$ compressor is used to compress information from the server to the workers. The performance of algorithms is compared on CIFAR10 (Krizhevsky et al., 2009) (# of features $= 3072$, # of samples equals $50,000$), and *real-sim* (# of features $= 20958$, # of samples equals $72,309$) datasets.

### Q.2 Results

In Figure 2, 3 and 4 we provide empirical communication complexities of our experiments. For each algorithm, we show the three best experiments. The experiments are collaborative with our theory. 2Direction enjoys faster convergence rates than EF21-P + DIANA and AGD.

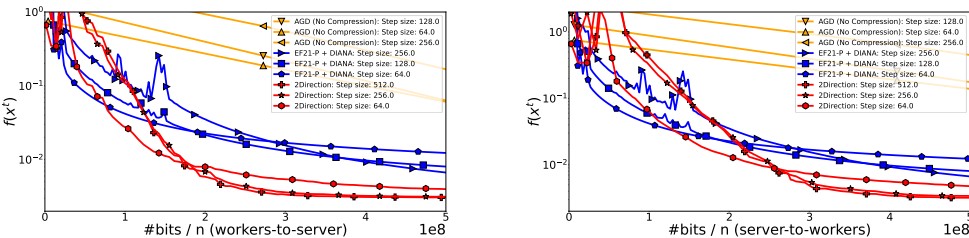

Figure 2: Logistic Regression with *real-sim* dataset. # of workers $n = 100$. $K = 1000$ in all compressors.

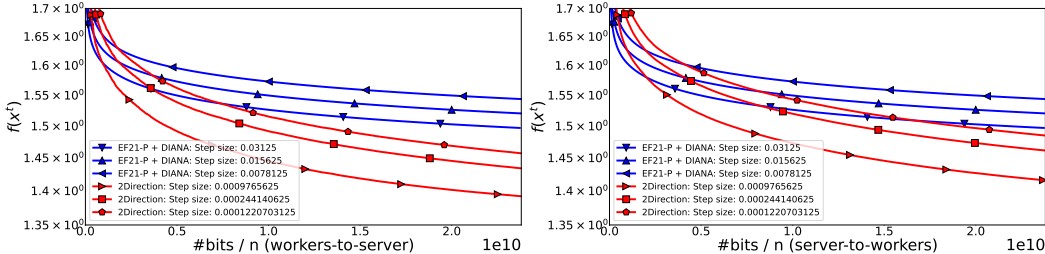

Figure 3: Logistic Regression with *CIFAR10* dataset. # of workers $n = 10$. $K = 1000$ in all compressors.

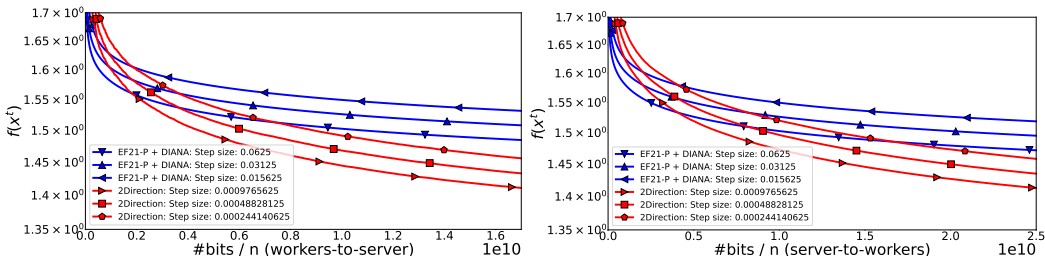

Figure 4: Logistic Regression with *CIFAR10* dataset. # of workers $n = 100$. $K = 1000$ in all compressors.

## R  Convergence Rate of CANITA obtained by Li and Richtárik (2021)

In their Equation (54), Li and Richtárik (2021) derive the following bound for their CANITA method:

$$\mathbb{E}\left[F^{T+1}\right] \leq \mathcal{O}\left(\max\left\{\frac{(1+\omega)^3}{T^3}, \frac{(1+b)(\beta+3/2)L}{T^2}\right\}\right).$$

In the regime when $\omega \geq n$, choosing $b = \omega$ and $\beta = \Theta\left(\frac{\omega}{n}\right)$ in their Equation (10) gives

$$\mathbb{E}\left[F^{T+1}\right] \leq \mathcal{O}\left(\max\left\{\frac{(1+\omega)^3}{T^3}, \frac{(1+b)(\beta+3/2)L}{T^2}\right\}\right)$$

$$= \mathcal{O}\left(\max\left\{\frac{(1+\omega)^3}{T^3}, \frac{\omega(\omega/n+3/2)L}{T^2}\right\}\right)$$

$$= \mathcal{O}\left(\max\left\{\frac{(1+\omega)^3}{T^3}, \frac{\omega^2 L}{nT^2}\right\}\right).$$

This means that the correct convergence rate of the CANITA method (Li and Richtárik, 2021) is

$$T = \begin{cases} \Theta\left(\frac{\omega}{\varepsilon^{1/3}} + \frac{\omega}{\sqrt{n}}\sqrt{\frac{L}{\varepsilon}}\right), & \omega \geq n, \\ \Theta\left(\frac{\omega}{\varepsilon^{1/3}} + \left(1 + \frac{\omega^{3/4}}{n^{1/4}}\right)\sqrt{\frac{L}{\varepsilon}}\right), & \omega < n. \end{cases} \tag{329}$$

Comparing this result with our Theorem E.14 describing the convergence of our method 2Direction, one can see that in the low accuracy regimes (in particular, when $\frac{\omega}{\varepsilon^{1/3}}$ dominates in (329)), our result improves $\Theta\left(\frac{1}{\varepsilon^{1/3}}\right)$ to at least $\Theta\left(\log\frac{1}{\varepsilon}\right)$. However, the dependence $\Theta\left(\log\frac{1}{\varepsilon}\right)$ should not be overly surprising as it was observed by Lan et al. (2019) already, albeit in a somewhat different context.

## S  Comparison with ADIANA

We now want to check that our rate (14) restores the rate from (Li et al., 2020). Since ADIANA only compresses from the workers to the server, let us take $r = 0$, the identity compressor operator $\mathcal{C}^P(x) = x$ for all $x \in \mathbb{R}^d$, which does not perform compression, and, as in (Li et al., 2020), consider the optimistic case, when $L_{\max} = L$. For this compressor, we have $\alpha = 1$ in (2). Note that $\mu^r_{\omega,\alpha} = 0$. Thus the iteration complexity (14) equals

$$T^{\text{optimistic}} = \widetilde{\Theta}\left(\max\left\{\sqrt{\frac{L}{\mu}}, \sqrt{\frac{L(\omega+1)}{n^{1/3}\mu}}, \sqrt{\frac{L(\omega+1)^{3/2}}{\sqrt{n}\mu}}, \sqrt{\frac{L\omega(\omega+1)}{n\mu}}, (\omega+1)\right\}\right)$$

$$= \widetilde{\Theta}\left(\max\left\{\sqrt{\frac{L}{\mu}}, \sqrt{\frac{L(\omega+1)^{3/2}}{\sqrt{n}\mu}}, \sqrt{\frac{L\omega(\omega+1)}{n\mu}}, (\omega+1)\right\}\right), \tag{330}$$

where we use Young's inequality: $\sqrt{\frac{L}{\mu}}\sqrt{\frac{(\omega+1)}{n^{1/3}}} \leq \sqrt{\frac{L}{\mu}}\sqrt{\frac{1}{3} \times 1^3 + \frac{2}{3}\frac{(\omega+1)^{3/2}}{\sqrt{n}}}$. Without the server-to-worker compression, Algorithm 1 has the same iteration (330) and communication complexity as (Li et al., 2020).

