# OpenReview forum: "2Direction: Theoretically Faster Distributed Training with Bidirectional Communication Compression"
_NeurIPS.cc/2023/Conference — NeurIPS 2023 poster_

### Official Review · Reviewer_AqDA · 2023-06-26

**Soundness:** 3 good
**Presentation:** 2 fair
**Contribution:** 3 good
**Rating:** 6
**Confidence:** 3

**Summary:**

This paper proposes 2Direction, a new GD-based distributed training algorithm for convex optimization problems. The improvement of 2Direction over previous works that target communication complexity in the convex case is that 2Direction is both accelerated and allows for bi-directional compression. 2Direction is proven to offer better communication complexity than previous works in certain regimes and not being worse otherwise.

**Strengths:**

1.	The scope of the paper and the main problem are well-defined and clear.
2.    The paper considers bi-directional compression. SO far, bi-directional compression received less attention in the literature that mostly considers only compressing uplink communication.
2.	The paper’s theoretical analysis and results are mostly clear and solid. The paper theoretically compares 2Direction to several SOTA alternatives.
3.	The design choices of 2Direction are well articulated.

**Weaknesses:**

1.	The communication complexity analysis and its optimization are not fully clear. Specifically, there is dependency between the round cost and the number of communication rounds (i.e., via K and w).
2.	2Direction (Algorithm 1) requires more memory and compute resources in comparison to some previous methods such as DG, AGD and EF21-P + DIANA  (e.g., 2Direction requires arrays for storing h, v and solving (11)-(12)-(23)). This overhead and its consequences, especially in federated learning contexts, are not discussed.
3.	The experimental part is somewhat weak (appears in the supplementary material and contains a single logistic regression problem with two datasets).
4. The presentation can be improved. In particular, the paper is not self contained and relies on the supplementary material to convey main ideas. Notably: (i) Section 4 points to Sections B and C to articulate design choices; (ii) Section 5.2 points to equations 58 and 63

**Questions:**

1.	Can the authors comment on weaknesses (1)(2)(3)?
2.	Can the authors shed more light on why with small probability exact values must be sent? Clearly the proof requires this, but more intuition on whether this is inherent or just an artifact of the proof technique would be helpful.
3.	The comparison/analysis is done with TopK (downlink) and RandK (uplink) as example compressors. However, these compressors have weaker worst-case MSE guarantees in comparison to SOTA compression techniques that provide asymptotically better worst-case tradeoffs between communication and MSE. Can the authors shed light on that? In particular, if compressors with stronger guarantees are used, can that affect the communication complexity analysis? E.g., since line 10 in algorithm 1 can be viewed as an instance of the distributed mean estimation problem, can the usage of techniques such as [1]-[3] lead to better communication complexity or alter the conclusions? (this also relates to weakness (1))
4.	Assumption 2.4 – while the independence between the workers in the uplink direction is indeed desired, why independence with the downlink direction is important?

[1] Suresh, Ananda Theertha, et al. "Distributed mean estimation with limited communication." International conference on machine learning. PMLR, 2017.

[2] Davies, Peter, et al. "New Bounds For Distributed Mean Estimation and Variance Reduction." International Conference on Learning Representations, 2021.

[3] Vargaftik, Shay, et al. "Eden: Communication-efficient and robust distributed mean estimation for federated learning." International Conference on Machine Learning. PMLR, 2022.

**Limitations:**

The authors clearly stated a limitation of their work in Section 8. I do have few additional suggestions about the memory/computational efficiency aspect. Moreover, since federated learning (FL) is mentioned as motivation, it worth discussing whether 2Direction can be potentially extended to common FL scenarios with partial participation and stochastic gradients.

I did not recognized potential negative broader impact.

---

> ### Author Rebuttal · Authors · 2023-08-05
>
> Thank you for the review!
>
> > The communication complexity analysis and its optimization are not fully clear. Specifically, there is dependency between the round cost and the number of communication rounds (i.e., via K and w).
>
> * The rounds costs and the number of communication rounds depend on $K$ ($K_{\omega}$ and $K_{\alpha}$) and $\omega$ in all methods (see Table 1). The parameters $K$ and $\omega$ are defined by a chosen compressor. For instance, in the case of Rand$K$, $K_{\omega} = K$ and $\omega = \frac{d}{K} - 1.$ In the case of Top$K$, $K_{\alpha} = K$ and $\alpha = \frac{K}{d}.$
> * One can find the communication complexities in (16) or in Table 1 (Comm. Compl = # Communication Rounds × Round Cost) as functions of $\omega,$ $\alpha$ and $K.$ For the Rand$K$ and Top$K$ compressors, we can substitute the parameters to Table 1 [2Direction] or (16) and obtain the communication complexity
> Round Cost × \# Communication Rounds = $K \times \left(\sqrt{\frac{L}{\mu}} \frac{d}{K} + \sqrt{\frac{L_{\max}}{n \mu}} \frac{d}{K} + \frac{d}{K}\right) = d \sqrt{\frac{L}{\mu}} + d \sqrt{\frac{L_{\max}}{n \mu}} + d.$
> Note that this communication complexity is never worse than the communication complexity of AGD. At the same it is very pessimistic because when we were deriving it we used that $\alpha = \frac{K}{d}$ in Top$K.$ However, in practice, $\alpha \gg \frac{K}{d}.$
>
> * Let us illustrate that. For simplicity, let us take Top$K$ with $K = 1.$ Then, one can show that  $||C(x) - x||^{2} = ||x||^2_2 -||x||^2_{\infty}.$ One can see that the "effective" $\alpha = ||x||^2_{\infty} / ||x||^2_2.$ It depends on the distribution of coordinates $x$. Yes, we can be unlucky, and in the case when all coordinates are equal, we get the worst case $\alpha = 1 / d.$ But in practice \[6, 7\], the distribution of coordinates is very non-equal to each other.
>
> > 2Direction (Algorithm 1) requires more memory and compute resources in comparison to some previous methods such as DG, AGD and EF21-P + DIANA.
>
> * The vanilla AGD method also requires additional memory compared to the vanilla GD method due to the fact that AGD uses the acceleration momentum technique. For instance, look at the implementation of AGD in \[1, p.64\]. It also has additional variables compared to GD.
> * The same reasoning applies to 2Direction and EF21-P + DIANA. Indeed, 2Direction requires more memory. However, it requires more memory by a **multiplicative constant factor**. In other words, the memory complexity of 2Direction, EF21-P + DIANA, AGD, and GD is $O(d),$ where the $d$ is the dimension of the problem. All methods have the same  $O(d)$ memory complexity. But we agree that the constant factors that the Big-O notation hides are different.
>
> > The experimental part is somewhat weak...
>
> Our work is theoretical, and we wish our work to be judged as such. We agree that the experiments are important to test the theoretical bounds. One can see that the behavior of methods in experiments in Section Q supports the theory. We analyze the logistic regression problem, which is the most popular convex optimization task in machine learning. Our method is designed only for convex problems, so we do not provide experiments with neural networks. Note that we provide the new theoretical SOTA communication complexities in the bidirectional convex setting.
>
> > Can the authors shed more light on why with small probability exact values must be sent? ...
>
> We believe that this is the nature of all *variance-reduced accelerated* methods. If you look at all other methods \[1, 2, 3, 4\] that accelerate variance-reduced methods, they all require intermittent computations via double loop or probabilistic switching (as it was done in our paper)
>
> > The comparison/analysis is done with TopK (downlink) and RandK (uplink) as example compressors. ...
>
> One can use any compressor in line 7 of Alg. 1 if it is unbiased (see Def 2.3). Also, one can use any compressor in line 13 of Alg. 1 if it is *biased* (see Def 2.2). The theory will hold for any choice of them. One can take his/her favorite compressor and use it our method. It is only necessary to find $K$ and $\omega.$ The parameters $K$ is the number of bits/coordinates that a compressor preserve. The parameters $\omega$ and $\alpha$ usually can be also easily estimated. For instance, Rand$K$ has $\omega = \frac{d}{K} - 1.$ Note that there are many more compressors \[5\]. The choice of TopK and RandK (as examples) is motivated by the fact that these compressors are the most popular in the literature, and they can be easily explained to a reader who is not from the community of distributed compressed methods.
>
> > Assumption 2.4 – while the independence between the workers in the uplink direction is indeed desired, why independence with the downlink direction is important?
>
> The independence of the uplink and the downlink directions is important. Also, it is important that the compression operators are independent across different iterations. Otherwise, we can not use the conditional expectation trick in the proofs. For instance, we use independence in Lines 528 and 519.
>
> **We believe that we addressed all questions. We kindly ask the reviewer to reconsider the score.**
>
> \[1\]: Lan G. First-order and stochastic optimization methods for machine learning
>
> \[2\] Lan G. et al  A unified variance-reduced accelerated gradient method for convex optimization
>
> \[3\] Zeyuan A. Katyusha: The First Direct Acceleration of Stochastic Gradient Methods
>
> \[4\] Kovalev D et al Don’t Jump Through Hoops and Remove Those Loops: SVRG and Katyusha are Better Without the Outer Loop
>
> \[5\] Xu H. GRACE: A Compressed Communication Framework for Distributed Machine Learning
>
> \[6\]: Beznosikov, A., Horvath, S., Richtarik, P., and Safaryan, M. On biased compression for distributed learning
>
> \[7\]: PowerSGD: Practical Low-Rank Gradient Compression for Distributed Optimization Thijs Vogels, Sai Praneeth Karimireddy, Martin Jaggi

---

> > ### Comment · Reviewer_AqDA · 2023-08-13
> >
> > Thank you for your answers.
> >
> > *W1:* based on your answer, is it correct that the reduction in communication complexity comes at the expense of more training rounds? For example, is it right that if K=1, 2Direction requires asymptotically more rounds (and by that gradient computations) than AGD? If this is the case, this should be discussed and conveyed as a legitimate tradeoff (at least for some reasonable choices of K).
> >
> > *W2:* Having previous works doing the same is evidence of a common difficulty in these sorts of algorithms, but it does not shed any light on whether this is an artifact of the proof method or the problem itself.
> >
> > *W3:* How to properly choose K or how different compressors with different guarantees affect the converge rate (and not only the communication complexity) is not discussed but left for the reader to deduce. This also relates to *W1*.
> >
> > *W4:* If I understand your answer correctly, is this assumption required to complete the proof? In that case, is there a simple example where dependence only in the downlink direction prevents convergence or asymptotically slows it down?

---

> > > ### Author Response · Authors · 2023-08-13
> > >
> > > > W1: based on your answer, is it correct that the reduction in communication complexity comes at the expense of more training rounds? For example, is it right that if K=1, 2Direction requires asymptotically more rounds (and by that gradient computations) than AGD? If this is the case, this should be discussed and conveyed as a legitimate tradeoff (at least for some reasonable choices of K).
> > >
> > > > W3: How to properly choose K or how different compressors with different guarantees affect the converge rate (and not only the communication complexity) is not discussed but left for the reader to deduce. This also relates to W1.
> > >
> > > Yes, if we take $K = 1,$ then our method will theoretically require more communication rounds than if $K$ was equal to $d$. **However, this is the case of all methods that reduce communication complexity,** just like every analysis of SGD with minibatch size equal to 1 will require more iterations than the full batch case. This is normal, and to be expected. What matters is the total complexity. In our case, it is the product of the number of communication rounds and the bits transferred in each round. We discuss this at length in the paper.
> > >
> > > Indeed, it turns out that $K = 1$ may not the best choice if we *also* care about the number of gradient computations. We can ask ourselves what is the maximum $K$ we can take that preserves the best complexity. Let us consider our complexity with the Rand$K$ and Top$K$ compressors. Then the communication complexity equals
> > > $$K \times \left(\sqrt{\frac{d}{K}}\sqrt{\frac{L}{\mu \alpha}} + \frac{d}{K}\sqrt{\frac{L_{max}}{\mu n}} + \frac{1}{\alpha} + \omega\right) = O\left(\sqrt{d K}\sqrt{\frac{L}{\mu \alpha}} + d\sqrt{\frac{L_{max}}{\mu n}} + d\right).$$ From the last formula, it is clear that we should take
> > > $K \leq d \alpha \min\left\[\frac{L_{max}}{L n}, \frac{\mu}{L}\right\]$ in order to preserve the best communication complexity. So instead of $K = 1,$ one can take $K = d \alpha \min\left\[\frac{L_{max}}{L n}, \frac{\mu}{L}\right\].$ We agree that it is not obvious how to calculate this formula in practice. However, we can try to find some reasonable bound for it. For instance, we can know that our problem is well-conditioned: $L / \mu \ll n.$ Thus we can take $K = \frac{d \alpha}{n}.$ In practice, the parameter $\alpha \approx 1$ in the Top$K$ compressor, thus the reasonable choice is $K = \frac{d}{n}.$ With this choice, the number of rounds equals $$\left(\sqrt{\frac{d}{K}}\sqrt{\frac{L}{\mu \alpha}} + \frac{d}{K}\sqrt{\frac{L_{max}}{\mu n}} + \frac{1}{\alpha} + \omega\right) = \sqrt{\frac{n L}{\mu \alpha}} + \sqrt{\frac{L_{max}}{\mu}} + \frac{1}{\alpha} + n.$$
> > > Assume that $\alpha \approx 1,$ then the number of rounds is not greater than $\sqrt{\frac{n L}{\mu}} + n.$ Unfortunattely, this number is greater than the number of rounds $\sqrt{\frac{L}{\mu}}$ of AGD (In all previous methods from Table 1, this gap is even worse). However, our method reduces the communication complexity! The same derivation one can do with other compressors instead of Rand$K$ and Top$K$.
> > >
> > > We believe that we answered the reviewer's question. We agree that this is an interesting discussion; we will be happy to add it to the paper. However, the main scope of our paper is communication complexity.
> > >
> > > > W2: Having previous works doing the same is evidence of a common difficulty in these sorts of algorithms, but it does not shed any light on whether this is an artifact of the proof method or the problem itself.
> > >
> > > If this comment related to "Can the authors shed more light on why with small probability exact values must be sent?", then we tried to explain it in as much detail as possible. The same problem was in many related works; the problem appears in our work. This is just the nature of accelerated variance-reduced methods. **Note that we show that this does not have an adverse effect on the communication complexity.** This is clearly not a weakness of our paper.
> > >
> > > > W4: If I understand your answer correctly, is this assumption required to complete the proof? In that case, is there a simple example where dependence only in the downlink direction prevents convergence or asymptotically slows it down?
> > >
> > > Notice that this assumption was considered in all previous papers on bidirection compression. We do not assume anything additional here. This is a standard assumption in the literature. Returning back to the question, the reviewer can look at Lines 528-529. In order to use Definition 2.2, we have to use that the fact $C^P$ is independent of $u^{t+1}$ and $q^{t+1}.$ For instance, the vector $u^{t+1}$ depends on $C^{D,y}_i$ and on the realization of $C^P$ from the previous iteration. In this place, the independence of compressors helps us to finish the proof.
> > >
> > > If you have more questions, we will happy to answer them.

---

> > > > ### Comment · Reviewer_AqDA · 2023-08-15
> > > >
> > > > I respectfully disagree with crucial parts of the response as described below.
> > > >
> > > > **W1+W3:**  The analogy to GD vs. SGD is misleading. SGD often exhibits many benefits in practice, such as faster convergence, noise regularization, memory efficiency, and the ability to handle large datasets. 2Direction was not shown to provide such benefits. On the contrary, in a balanced setup where networking and compute resources exhibit balanced utilization (which is desired), 2Direction may severely degrade the convergence time; 2Direction may force the participants to a significantly longer training procedure even though the total communication complexity is reduced. Moreover, the new analysis shows that even for the best choice of K proposed by the authors, 2Direction requires a factor of $\sqrt n$ more training rounds than AGD (this increases with n!).
> > > >
> > > > **W2+W4:** The reviewer understands that the same ingredients are present in previous works and does not consider them a weakness. It is, however, somewhat weakening not having evidence or intuition as to whether these are an artifact of the proof methods or inherently required to improve communication complexity (both outcomes are fine).
> > > >
> > > > The reviewer appreciates the difficulty in constructing the proof and the careful design of the algorithm that allows for that proof and understands that the contribution of the paper is theoretical. But the paper draws motivation from distributed/federated learning. Thus, there is an expectation that the theory will reflect inspiration from realistic setups and that the solution to the theoretical problem can be used to design better solutions for such setups. On that front, the paper needs evidence that 2Direction can be used for such a purpose. This makes it hard to appreciate the significance of the theoretical result.
> > > >
> > > > I strongly encourage the authors to present evidence (e.g., more evaluation) that 2Direction can improve convergence time or clearly and explicitly address/discuss all aspects of the work, including its practical limitations.

---

> > > > > ### Author Response · Authors · 2023-08-15
> > > > > **Reply to "Official Comment by Reviewer AqDA", part 1**
> > > > >
> > > > > > W1+W3: The analogy to GD vs. SGD is misleading.
> > > > >
> > > > > We believe the analogy is in fact, nearly perfect. Indeed, in W1 you wrote "W1: based on your answer, is it correct that the reduction in communication complexity comes at the expense of more training rounds? For example, is it right that if K=1, 2Direction requires asymptotically more rounds (and by that gradient computations) than AGD? If this is the case, this should be discussed and conveyed as a legitimate tradeoff (at least for some reasonable choices of K)."
> > > > >
> > > > > Based on our knowledge, just like
> > > > > - all variants of SGD we know of require more iterations than GD in the same regime, but have cheaper per-iteration cost (= computation cost of forming the gradient estimator), and the product of the two (= gradient evaluation complexity) is what matters,
> > > > > - all variants of compressed gradient methods we know of require more iterations (= communications), but have cheaper per-iteration cost (= cost of communication, which is equal to w2s communication in the one-directional setup, and w2s + s2w communication in the bi-directional setup; also notice that virtually all of these works assume the computation cost is zero when providing bounds), and the product of the two (= communication complexity) is what matters.
> > > > >
> > > > > This analogy is indeed nearly perfect in our view. Notice that many variants of compressed gradients methods can be formally be viewed as variants of SGD in which compression replaces subsampling as the source of stochasticity; further strengthening this analogy; see [P1]
> > > > >
> > > > > [P1] Gorbunov et al, A unified theory of SGD: variance reduction, sampling, quantization and coordinate descent, AISTATS 2020.
> > > > >
> > > > > As all analogies go, our also needs to be understood in the limits in which it was intended to apply. When you subsequently say that "The analogy to GD vs. SGD is misleading. SGD often exhibits many benefits in practice, such as faster convergence, noise regularization, memory efficiency, and the ability to handle large datasets. 2Direction was not shown to provide such benefits.", you bring new considerations into the discussion which were not intended to be covered by our analogy. We think that this is not fair. Our brief response to these new considerations follows:
> > > > > - In theory, "faster convergence" of SGD refers to the product of the number of iterations and the minibatch size. Indeed, SGD is faster than GD in this sense; this is well known, and can be seen very clearly in, e.g., Gower et al, SGD: general analysis and improved rates, ICML 2019. However, in a very precise sense (in terms of communication complexity defined above), unidirectionally compressed gradient descent methods were shown to be faster than GD (see the complexity of DIANA in [P1], and in the original paper it refers to, for example).
> > > > > - Compressed gradient methods also enjoy some level of noise regularization -- from the noise inherent in the compressors. This is well known.
> > > > > - Unlike traditional SGD, which is designed to work in the single-machine regime, compressed gradient descent methods are designed to work in the distributed regime, where communication efficiency is more typically seen as more important than memory efficiency. Having said that, there are variants of SGD (e.g., SAG, SAGA, SVRG) which work with control variates which require more memory, and in the same manner there are variants of compressed gradient methods (e.g., DIANA, ADIANA, PAGE, MARINA) which require more memory for the exact same reason. Just likes these methods, our 2Direction method requires a bit of extra memory. However, what we gain is a method with vastly improved communication complexity.
> > > > > - Re "ability to handle large datasets" - we study the distributed setup precisely because single node methods are unable to do so. So, 2Direction is in principle better suited to handling large datasets.
> > > > >
> > > > > Because of what we say above, we do not see how what we said was misleading.

---

> > > > > ### Author Response · Authors · 2023-08-15
> > > > > **Reply to "Official Comment by Reviewer AqDA", part 2**
> > > > >
> > > > > > On the contrary, in a balanced setup where networking and compute resources exhibit balanced utilization (which is desired), 2Direction may severely degrade the convergence time; 2Direction may force the participants to a significantly longer training procedure even though the total communication complexity is reduced.
> > > > >
> > > > > In virtually all theoretical works on communication efficient distributed optimization, communication complexity is the key quantity of interest, and local computation time is not taken into consideration (in the associated theory). We know about a couple of exceptions only, including
> > > > >
> > > > > [P2] Malinovsky et al, Variance reduced ProxSkip: algorithm, theory and application to federated learning, NeurIPS 2022 (see Section 5).
> > > > >
> > > > > This is because these works assume that the system is communication bound, i.e., communication is much more expensive than computation, and the algorithms in this genre, including 2Direction, cater to this setup. Therefore, your criticism "in a balanced setup where networking and compute resources exhibit balanced utilization" does not apply to our work - our method is not designed to deal with this regime since indeed in this case compressing communication does not make sense. This is the situation with all methods employing communication compression.
> > > > >
> > > > > So, your criticism is generic and applies to the whole field of compressed communication, and not merely to our paper. However, it is also trivial since these methods are not designed to handle this balanced regime. So, we believe that this criticism is not appropriate. Having said that, we are most happy to include a discussion of this (obvious) limitation of all works on compressed communication.
> > > > >
> > > > > > Moreover, the new analysis shows that even for the best choice of K proposed by the authors, 2Direction requires...
> > > > >
> > > > > In communication-bound settings, communication complexity is what matters. *All* methods that perform communication compression try to achieve this by an optimal balance of the number of communication rounds (which is a nondecreasing function of the compression ratio), and the compression ratio -- optimal in the sense of communication complexity. As we have remarked before, in practice it is often possible to obtain a certain limited compression ratio without the need to increase the number of comm. rounds at all. Once this limit is reached and exceeded, the number of required comm rounds increases, but the comm complexity may still decrease, up to some optimal compression ratio, after which deterioration of comm complexity starts.
> > > > >
> > > > > > W2+W4: The reviewer understands that the same ingredients are present in previous works and does not consider them a weakness. It is, however, somewhat weakening not having evidence or intuition as to whether these are an artifact of the proof methods or inherently required to improve communication complexity (both outcomes are fine).
> > > > >
> > > > > We do not know if such an increase in memory is necessary to obtain our superior comm complexity bounds, but we certainly think so, since the kinds of bounds we obtain seem to require the use of control variates (this was the case in all previous research in other domains we know of). We do not have a proof that a different method can't be constructed whose comm complexity would match that of 2Direction, but one that would need less memory. This can be a difficult problem to solve on its own if a definitive answer is sought. Having said that, we believe the mere fact that we obtain a new SOTA theoretical comm complexity (the first that always matches AGD and can beat it in some scenarios) for optimization with bi-directional communication compression is a breakthrough on its own, in a strongly contested area of research.
> > > > >
> > > > > > The reviewer appreciates the difficulty in constructing the proof and the careful design of the algorithm that allows for that proof and understands that the contribution of the paper is theoretical.
> > > > >
> > > > > Thank you!
> > > > >
> > > > > > But the paper draws motivation from distributed/federated learning. Thus, there is an expectation that the theory will reflect inspiration from realistic setups and that the solution to the theoretical problem can be used to design better solutions for such setups. On that front, the paper needs evidence that 2Direction can be used for such a purpose. This makes it hard to appreciate the significance of the theoretical result.
> > > > >
> > > > > We believe that our theory stands on its own - if one believes that theory has intrinsic value. Having said that, we perform some illustrative experiments which indicate that the theoretical benefits indeed translate to benefits in practice. Of course, more work needs to be done
> > > > > in this direction. We view our work as pioneering in this direction, and hence do not expect it's reasonable to expect that a single work will  resolve all these issues.

---

> > > > > ### Author Response · Authors · 2023-08-15
> > > > > **Reply to "Official Comment by Reviewer AqDA", part 3**
> > > > >
> > > > > > I strongly encourage the authors to present evidence (e.g., more evaluation) that 2Direction can improve convergence time or clearly and explicitly address/discuss all aspects of the work, including its practical limitations.
> > > > >
> > > > > We can do so; this is easy to do. But we believe that our work is a theoretical breakthrough as it is, despite the fact that it can be somewhat strengthened by the inclusion of more experiments, and a more detailed account of limitations. We take these suggestions seriously, but we view them as minor.
> > > > >
> > > > > For all of the above reasons, we certainly think the score 4 is unjustified.
> > > > >
> > > > > **If you agree that we designed a new method which attains the current SOTA theoretical communication complexity rate in the important field of communication efficient distributed training with bi-directional compression, why would such a low score be appropriate? Empirical works with SOTA practical results routinely obtain very high scores despite not including any theory. We believe that theory and practice should be valued equally.**

---

> > > > > > ### Comment · Reviewer_AqDA · 2023-08-21
> > > > > >
> > > > > > My concerns, which I repeatedly point out, are about the **practicality and applicability** of 2Direction rather than about the validity of the theoretical results or the assumptions.
> > > > > >
> > > > > > 1. Take, for example, our discussion about the analogy of SGD vs. DG. In that regard, the authors made convincing arguments from a theoretical point of view, where my concerns are pointed out from a practical point of view where this analogy is far from perfect. SGD is known to offer practical improvements in metrics of interest, such as training time and cost. No such improvements were shown to be achieved by 2Direction; moreover, there is no evidence that they can be achieved or that 2Direction can lead to a similar solution that can.
> > > > > >
> > > > > > 2. Regarding the assumption that computation has zero cost, then why does limiting the algorithm to a single local gradient step make sense? Why even use first-order approximation and not, e.g., Neuton's method? Clearly, in any practical setup, this cost is not zero. Let me emphasize this point further. There are three main setups the paper may apply for (i) distributed learning in a cluster or cross-silo FL. Here the zero computation cost is not justified, as all entities are expected to have a lot of data. (ii) Cross-device FL. Here partial participation is not addressed. Again, I am not suggesting that 2Direction should address all these considerations, but without clarity about the assumptions and what they actually mean (i.e., the resulting limitations), it is difficult to appreciate the importance of the result.
> > > > > >
> > > > > > 3. Stating that communication compression makes no sense in a balanced system is wrong and goes against standard practice. Making more than a single local gradient step or increasing batch size (e.g., via gradient accumulation) to produce higher-quality updates that essentially require **less** communication rounds to converge in practice is one of the motivations behind algorithms such as federated averaging. Namely, one can achieve balance by making more local computations and compressing communication - in other words, compression is an important ingredient in achieving this.
> > > > > >
> > > > > > 4. I strongly agree that theory has intrinsic value. However, the fact that something is hard to prove does not mean it is important or interesting. I am not suggesting that 2Direction is not important or interesting, just that the paper does an insufficient job of making that clear or being explicit about what was actually achieved that has practical value or offers new high-level insights that were missed by previous works.
> > > > > >
> > > > > > 5. I do not find my criticism generic. Generally, the fact that previous works assumed something does not mean that these assumptions should be accepted without any hesitation or criticism today or in the future. In fact, reexamining assumptions is at the very heart of progress. Therefore, stating that something critical is assumed because it was assumed by previous works and not convincing why this assumption makes sense today can be viewed as generic.
> > > > > >
> > > > > > 6. To conclude, the authors stated in their response:
> > > > > >
> > > > > > **.. So, your criticism is generic and applies to the whole field of compressed communication, and not merely to our paper. However, it is also trivial since these methods are not designed to handle this balanced regime. So, we believe that this criticism is not appropriate. Having said that, we are most happy to include a discussion of this (obvious) limitation of all works on compressed communication. **
> > > > > >
> > > > > > I find everything in this sentence to be wrong: (1) I have concerns regarding 2Direction. I have not reviewed the entire field of compressed communication, and I do not think that 2Direciton represents the entire field. (2) There is nothing trivial about the zero-computation assumption, and there are papers that address the practically common balanced regime where an improvement of even a few percent in networking or compute time may be important (e.g., [1] and reference therein). (3) The limitations are not obvious and are easy to miss, as the connection to practical scenarios in the paper is vague.
> > > > > >
> > > > > > To conclude, I appreciate the time the authors spent addressing my concerns. Considering **all** the arguments made by the authors (where with some of them I disagree), I am raising my score from 4 to 6 based on my understanding that the theoretical result achieved in this paper is important and can encourage further progress and lead to more practical solutions with reduced communication complexity.
> > > > > >
> > > > > > I hope that the authors will discuss explicitly in the paper the assumptions and their resulting practical limitations and encourage future research to address these limitations.
> > > > > >
> > > > > > [1] Wang, Z., Lin, H., Zhu, Y. and Ng, T.E., 2023, May. Hi-Speed DNN Training with Espresso: Unleashing the Full Potential of Gradient Compression with Near-Optimal Usage Strategies. In Proceedings of the Eighteenth European Conference on Computer Systems (pp. 867-882).

---

> > > > > > > ### Author Response · Authors · 2023-08-21
> > > > > > > **Thank you!**
> > > > > > >
> > > > > > > Thank you!

---

### Official Review · Reviewer_EaSZ · 2023-07-05

**Soundness:** 2 fair
**Presentation:** 2 fair
**Contribution:** 2 fair
**Rating:** 4
**Confidence:** 3

**Summary:**

This paper proposes a new distributed convex optimization problems with bidirectional compression. This algorithm, named as 2Direction, is claimed to be the first that improves upon the communication complexity of the vanilla accelerated gradient descent.

**Strengths:**

1. The 2Direction method seems novel
2. The studied problem is well-motivated

**Weaknesses:**

1. The proof of the algorithm is rather complicated. I cannot verify the solidness of the established theorem given the very tight review timeline. The appendix I, J, K, L and O are very scary given the symbolically computed expressions. It is basically a computer-aided proof. I will discuss with other reviewers and chairs on how to check this proof.

2. In table 2, some terms are associated with $\Omega$ while the others are not. Please clarify the reason.

3. 2Direction uses a unbiased compressor and a contractive compressor in algorithm design. Can 2Direction use unbiaed compressors for both w2s and s2w compression? Can 2Direction use contractive compressors for both w2s and s2w compression?

4. Does 2Direction have the smallest communication complexity when r = 0 compared with existing uni-directional compression algorithms?

Since I cannot verify the correctness of the established theorem, I would recommend borderline reject at this stage.

**Questions:**

See above

---

> ### Author Rebuttal · Authors · 2023-08-05
>
> Thank you for the comments!
>
> > The proof of the algorithm is rather complicated. I cannot verify the solidness of the established theorem given the very tight review timeline. The appendix I, J, K, L and O are very scary given the symbolically computed expressions. It is basically a computer-aided proof. I will discuss with other reviewers and chairs on how to check this proof.
>
> > Since I cannot verify the correctness of the established theorem, I would recommend borderline reject at this stage.
>
> We really appreciate your time and effort, thank you! **But it is not fair to leave us with the comment saying you would recommend rejection because you "cannot verify the correctness of the established theorem." We know the proofs and results are correct.** **Please ask us questions regarding the proof. We will be happy to answer them.** The computer-aided part is only listed starting from page 45. Before that, do you have any concerns regarding our proofs? Note that we give a brief overview of our proof in Section 7.
>
>  * We are not asking to check the generated formulas. It is sufficient to check that we generated the formulas correctly. So it is sufficient to check and understand the code on pages 64-71. The formulas are only listed for reproducibility purposes.
>
> * **The computer-aided proofs are very popular and becoming an essential instrument in modern optimization methods. Recent breakthroughs weren't possible without them. See, for instance, the celebrated SAG paper \[2\] (the authors were awarded the Lagrange prize in optimization for this work)! The modern era of variance-reduced methods (SVRG, Katyusha, SAGA, ...) would not be possible without this paper; as this work started this subfield.** We can list many more papers: \[3,4\]. Besides this, formal theorem provers are becoming popular as well, e.g., the Lean theorem prover developed by Leonardo de Moura.
>
> * Let us briefly give additional details why we need symbolic computation in a part of the proof. The first time we use this is on Line 704 (formula (87)). We need it there to substitute $\nu$ in (86). In this place, we agree that the computation assistant is not necessary. But it is becoming clear that it is necessary when we get (89). In order to get (89), we have to multiply a large number of fractions. The situation with the number of fractions, when we substitute the parameters to (91), is even worse, and the number of fractions that we have to multiply is even bigger. The symbolic computations are simply used to multiply brackets for us. Note that we could do it without the assistant, but the probability of making mistakes is much higher. Another place where we use symbolic comutations is when we check assertions in Section O. There, we also could it do it by hand, but it would be more difficult without a computation assistant.
>
> * It is clear that the method *requires* a computation assistant, and there is nothing that we can do here. **We could write the proof without the computer assistant, but it would make the proof even larger and less reader-friendly. The probability of making a mistake would be higher without the assistant.**
>
> > In table 2, some terms are associated with $\Omega$ while the others are not. Please clarify the reason.
>
> Because in some papers, the *full* convergence rate is not transparent. For instance, take a look at Theorem 1 from \[1\]. Just to be on the safe side, for all the methods with $\Omega$, we say that the complexity is not better than $\Omega(...).$ We agree that we should leave a footnote to explain that.
>
> > 2Direction uses a unbiased compressor and a contractive compressor in algorithm design. Can 2Direction use unbiaed compressors for both w2s and s2w compression?
>
> Yes, it can! Because one can get the biased compressor from an unbiased one using the following transformation: $\frac{1}{
> \omega + 1} \times C$ (see Lines 75-76)
>
> > Can 2Direction use contractive compressors for both w2s and s2w compression?
>
> No, it can't. In the proofs, it is important that clients use an unbiassed compressor. The use of contractive compressors for both w2s and s2w compression is a very challenging task. The known methods that work with contractive compressors in both directions cannot even match the communication complexity guarantees of the vanilla GD method (e.g., see the EF21-BC method and its complexity \[5\]). The obtained communication complexities of such methods are much worse than the same methods but using the identity compressor (i.e., variants that do not compress anything). Before our work, even with unbiased compressors an improvement over baselines not performing any compression was not achieved, and the vanilla AGD method had the best communication complexity in the convex setting.
>
> > Does 2Direction have the smallest communication complexity when r = 0 compared with existing uni-directional compression algorithms?
>
> Yes, it is! We explain it Lines 240-241 and proof in Section S. we show that the communication complexity of 2Direction is not worse than the communication complexity of ADIANA which is the current SOTA method in the uni-directional (w2s) setting!
>
> **If you have any additional questions, please ask us. We believe that we addressed all questions. We kindly ask the reviewer to reconsider the score.**
>
> \[1\] Liu X. A Double Residual Compression Algorithm for Efficient Distributed Learning
>
> \[2\] Schmidt M. et al Minimizing Finite Sums with the Stochastic Average Gradient
>
> \[3\] Drori Y.  et al Performance of first-order methods for smooth convex minimization: a novel approach.
>
> \[4\] Grimmer B. Provably Faster Gradient Descent via Long Steps
>
> \[5\] Fatkhullin I. et al. EF21 with bells & whistles: practical algorithmic extensions of modern error feedback

---

> ### Comment · Area_Chair_zZCP · 2023-08-14
> **Let us have some deep discussion here**
>
> To Reviewer: Thanks for the questions. Could you check if the author answered your concerns?
>
> To Authors: a followup question for 3. Can you explain why the biased compression cannot be allowed, given it is allowed for analyzing the stochastic algorithms such as DoubleSqueeze [Tang et. al. 2019]?

---

> > ### Author Response · Authors · 2023-08-14
> >
> > > To Authors: a followup question for 3. Can you explain why the biased compression cannot be allowed, given it is allowed for analyzing the stochastic algorithms such as DoubleSqueeze [Tang et. al. 2019]?
> >
> > Note that the biased compression *is allowed* on the *server's side*.
> >
> > Let us discuss the compression on the *workers' side*. First, notice that our goal was to design a method that will get the communication complexities guarantees that are not worse than AGD (that does not compress). Our work showed that it is possible to accomplish with unbiased compressors on the *workers' side*.
> >
> > However, the design of a method that will use biased compressors both on the *server's side* and on the *workers' side* is a very challenging task if we care about obtaining strong theoretical guarantees. Indeed, the theoretical communication guarantees of such methods are much worse than the communication guarantees of AGD (that does not compress). In particular, consider DoubleSqueeze [Tang et. al. 2019]. It requires stronger assumptions (Assumption 1.3 or the bounded gradients assumption) than our work and AGD. Next, the worst-case theoretical communication complexity of this work is $\frac{d^2}{K} \times \frac{1}{\varepsilon^3}$ to find an $\varepsilon$-stationary point with Top$K.$ Another recent work \[1\] with only biased compressors guarantees the communication complexity $\frac{d^2}{K} \times \frac{L_{\max}}{\mu} \log\frac{1}{\varepsilon}.$ The same reasoning applies to Dore \[2\]. The communication complexities of such methods are clearly worse than the communication complexity $d \times \sqrt{\frac{L}{\mu}} \log \frac{1}{\varepsilon}$ of AGD (that does not compress!).
> >
> > In other words, we do not allow biased compression on the workers' side because we do not know how to design a method that will have communication guarantees not worse than in AGD. As far as we know, nobody knows how to solve this challenging task. Our work can be a starting point to solve it.
> >
> > Another orthogonal motivation behind using the *unbiased compressors* is their scaling with the number of nodes $n$. The larger $n,$ the smaller the effect of the noises from the *unbiased compressors.* It can be seen in the complexities of EF21-P + DIANA and 2Direction in Table 1. The noise from the *biased compressors* does *not* decrease with the number of nodes $n.$
> >
> > \[1\]: Fatkhullin I. EF21 with Bells & Whistles: Practical Algorithmic Extensions of Modern Error Feedback
> >
> > \[2\]: Liu et al A double residual compression algorithm for efficient distributed learning, 2019

---

### Official Review · Reviewer_M4eW · 2023-07-06

**Soundness:** 3 good
**Presentation:** 3 good
**Contribution:** 3 good
**Rating:** 7
**Confidence:** 3

**Summary:**

This paper studies 2Direction, a new method for compressed communications for the centralized distributed convex optimization problem, in the case where both worker-to-server and server-to-worker communications are compressed. Compared to previous work EF21-P+DIANA, it  guarantees a total communication complexity no worse than Nesterov's accelerated gradient methods.

**Strengths:**

* The paper introduces a new objective of minimizing the *total communication complexity*, which is well defined through equation 4 and clearly explained with the problematic lines 118-122.
* To alleviate some of the burden of understanding the seemingly cumbersome algorithm and proof, efforts were made to describe some of the steps of the research/thought process that led to the final method in Section 4, and to provide a sketch of the proof in Section 7.
* For reproducibility purposes, I appreciated the presence of the SymPy code used for the symbolic computation part of the proof.
* The final result is original, seems to have required more than a *"small tinkering"* of previous methods (Section 4), and displays both theoretical and practical improvements over previous state of the art algorithms.

**Weaknesses:**

* While efforts were made to ease the understanding of the general ideas behind the algorithm and proof, they seem so cluttered that it deters from delving into the technical details.

**Questions:**

* line 130 claims a *"new error feedback mechanism"* but only compares it to the EF21-P method. What is the difference between equation 9 and other mechanisms such as MEM-SGD [1] ?
* In the experiments part (appendix Q), why use a single value of $K$ ? Would it be possible to display the evolution of the convergence rate at different degrees of compression ?
* line 1033, the stepsize $\bar L$ is said to be finetuned. However, if we assume $L = L_{\max}$ (which we can find for logistic regression), wouldn't equation 44 give a closed form expression of $\bar L$ ?

**Typos :**
* The order of Equations 6 and 7 seems to have been reversed.
* line 195: a *"and"* seems to have been left between *"general analysis"*.


**References :**

[1] Sebastian U Stich, Jean-Baptiste Cordonnier, and Martin Jaggi. *Sparsified SGD with memory*. In NeurIPS, 2018.

**Limitations:**

*

---

> ### Author Rebuttal · Authors · 2023-08-05
>
> Thank you for the positive evaluation and comments!
>
> > While efforts were made to ease the understanding of the general ideas behind the algorithm and proof, they seem so cluttered that it deters from delving into the technical details.
>
> Unfortunately, this is the nature of all accelerated methods. In our method, everything is overcomplicated by the fact that we have bidirectional compression. This is the first method that has better communication guarantees than the vanilla accelerated method. We tried to explain the proof idea in Section 7.
>
> > line 130 claims a "new error feedback mechanism" but only compares it to the EF21-P method. What is the difference between equation 9 and other mechanisms such as MEM-SGD [1] ?
>
> If we understood their work correctly, MEM-SGD is simply the EF mechanism from (Seide et al. (2014)). Their contribution seems to be not on the algorithmic part, but in theory (Seide et al (2014) did not provide any theory). In the abstract and in Lines 127-128, we explain that EF21-P is a reparameterization of the celebrated EF mechanism (see details in \[1, Section 2\]). So the difference between (9) and EF (MEM-SGD) is the same as the difference between (9) and EF21-P (under re-parameterization of variables). Note, however, that we had to use a new and modified version of the EF21-P mechanism (see Lines 167-176 where we explain the difference), and that unlike in [1], where classical EF is used on the workers' side, we use our new EF mechanism at the server side.
>
> > In the experiments part (appendix Q), why use a single value of $K$? Would it be possible to display the evolution of the convergence rate at different degrees of compression?
>
> There are many parameters, including $K,$ # of workers $n,$ dataset type... It is practically infeasible to experiment all possible parameters. In our experiments, we took several key parameters to check that the dependences from the theory holds in practice as well (it does). Note that the acceleration nature of the method is independent of the chosen $K.$ But we can easily also add experiments where we study the choice of $K$. We do not think though that such experiments would add much value.
>
> > line 1033, the stepsize $bar{L}$ is said to be fine-tuned...
>
> In general, for logistic regression, we can only find an *upper bound* for $L_{\max}.$ Thus, it is still not tight. Moreover, the reviewer can see that (44) is proportional to the large constant which is just an artifact of the proof. It is possible to get a better constant, but the proof will be even much larger. Note that the step size in EF21-P + DIANA is also finetuned, so the comparison is fair.
>
> \[1\] Gruntkowska K. et al EF21-P and Friends: Improved Theoretical Communication Complexity for Distributed Optimization with Bidirectional Compression

---

### Official Review · Reviewer_gU3b · 2023-07-29

**Soundness:** 3 good
**Presentation:** 3 good
**Contribution:** 3 good
**Rating:** 6
**Confidence:** 3

**Summary:**

The authors considered a distributed convex optimization setting where both uplink and downlink communications should be considered. In this setting, the authors proposed an accelerated method called 2Direction which generalizes AGD into the bidirectional compressed communication setting and prove its convergence rate and communication complexity. Finally, empirical results are provided.

**Strengths:**

For this new setting, the authors propose new strategies to make the adaption of AGD possible, which is nontrivial. Meanwhile, the bounds seem OK to me and can be applicable in the smooth transition from uplink to downlink.

**Weaknesses:**

Such a generalization for the bidirectional case fills an important gap, but it lacks enough motivational examples for why such settings deserve study.

The algorithm name 2Direction is confusing as we don't really know what it means. It is better to follow the tradition and use the abbreviation of the full algorithm name.

The authors show the motivation of the new strategy is the existing approaches failed in the bidirectional case but don't show the concrete reason why failed. Is the failure superficial or essential?

The experiments should be included in the main body.

**Questions:**

See Weaknesses

---

> ### Author Rebuttal · Authors · 2023-08-05
>
> Thank you for the review!
>
> > For this **new setting**, the authors propose new strategies to make the adaption of AGD possible, which is nontrivial. Meanwhile, the bounds seem OK to me and can be applicable in the smooth transition from uplink to downlink.
>
> > Such a generalization for the bidirectional case fills an important gap, but it lacks enough motivational examples for why such settings deserve study.
>
> The bidirectional setting is not a new setting. The first methods that consider this setting can be traced back to at least in 2019 \[3, 10, 11\]. This topic is increasingly popular in the community \[6\] (ICML 2023), \[7]\ (ICML 2023) , \[8\] (NeurIPS 2021). See also the recent paper \[9\] (ICML 2023) on fine-tuning large language models, where bidirectional compression was essential to obtain the strong empirical results (btw: their theoretical communication complexity is worse than that of vanilla gradient descent; our results point to *better* rates than vanilla accelerated gradient descent). In short, this is a very important setting.
>
> The bidirectional compression is often overlooked because the analysis of unidirectional methods is much easier and by the assumption that the broadcasting from a server to the workers **is free.** Clearly, the assumption that your phone/laptop can receive data from the server for free is an oversimplification. The communication speeds in both directions have some limited speed in reality. Please try to visit speedtest(dot)net and check download and upload speeds. These speeds will be equal in most locations, and it is quite unlikely that the download speed will equal $\infty.$
>
> > The algorithm name 2Direction is confusing as we don't really know what it means. It is better to follow the tradition and use the abbreviation of the full algorithm name.
>
> 2Direction = TwoDirection $\approx$ Bidirectional. It is just a game of words. One can think that 2Direction is an abbreviation of a "Bidirectional Method."
>
> As authors, we have the right to choose a name for our method. Some authors use abbreviations of full algorithm names (when such abbreviations lead to "good" names), some choose playful names (e.g., Hogwild! - a test of time award winner; CocktailSGD \[9\]; or DoubleSqueeze \[10\] - our name is in some sense similar to this one), some authors choose female/male names (e.g., consider the celebrated Katyusha method \[1\], celebrated Adam method, or Diana / Marina \[2\]), and so on.
>
> This is clearly not a weakness of our paper; we see it at best as a minor recommendation based on personal preference of the reviewer. We happen to have a different preference in this particular case. As such, we believe that our choice of the algorithm name should not affect accept / reject recommendations.
>
> > The authors show the motivation of the new strategy is the existing approaches failed in the bidirectional case but don't show the concrete reason why failed. Is the failure superficial or essential?
>
> The failure of preceding methods seems essential - neither the authors nor us know how to prove that these methods have better communication complexity than vanilla GD (with the exception of \[6\]), let alone accelerated GD. This is not a proof of impossibility, but we believe the preceding methods and simply not designed well, and a major redesign of the algorithms was needed.
>
> In other words, we believe that our new strategy is essential. In Lines 167-176, we explain why. It is clear from the proof that the designed modification is required to obtain the convergence of a Lyapunov function in our method. If you compare the Lyapunov functions of non-accelerated \[4\] and accelerated methods \[5\], you will see that the control variables "converge" to the non-fixed vectors (please see explanation in Lines 167-176).
>
> > The experiments should be included in the main body.
>
> Our work is theoretical, we believe that the theoretical insights are by an order of magnitude more important than the experimental section. Having said that, if the paper gets accepted, we expect to get an extra page and we plan to do as you advise.
>
> If you have any additional questions, then let us know please.
>
> \[1\] Z Allen-Zhu et al Katyusha: The first direct acceleration of stochastic gradient methods
>
> \[2\] Gorbunov et al. MARINA: Faster non-convex distributed learning with compression
>
> \[3\] Liu et al A double residual compression algorithm for efficient distributed learning, 2019
>
> \[4\] Gorbunov et al. A unified theory of SGD: Variance reduction, sampling, quantization and coordinate descent.
>
> \[5\] Z. Li et al Acceleration for compressed gradient descent in distributed and federated optimization
>
> \[6\] Gruntkowska K. EF21-P and Friends: Improved Theoretical Communication Complexity for Distributed Optimization with Bidirectional Compression
>
> \[7\] Dorfman R. DoCoFL: Downlink Compression for Cross-Device Federated Learning
>
> \[8\] Philippenko C. Preserved central model for faster bidirectional compres335 sion in distributed settings.
>
> \[9\] Jue Wang, Yucheng Lu, Binhang Yuan, Beidi Chen, Percy Liang, Christopher De Sa, Christopher Re, Ce Zhang. CocktailSGD: Fine-tuning Foundation Models over 500Mbps Networks, 2023
>
> \[10\] Hanlin Tang, Xiangru Lian, Chen Yu, Tong Zhang, Ji Liu. DoubleSqueeze: Parallel Stochastic Gradient Descent with Double-Pass Error-Compensated Compression, 2019
>
> \[11\] Horvath et al, Natural Compression for Distributed Deep Learning, 2019

---

> > ### Comment · Area_Chair_zZCP · 2023-08-14
> > **It is worth to discuss the relationship with [10, 9, 3]**
> >
> > How is the 2Dicrection compression approach different from the "error compensated" compression in double directions in [3, 9, 10]?

---

> > > ### Author Response · Authors · 2023-08-14
> > >
> > > From the theoretical point of view, we can get substantially better theoretical guarantees. The works [9,10] consider only the non-convex setting and obtain the convergence guarantees of the norm of gradients, so their result is much weaker in the convex case. Also, [9,10] require stronger assumptions (e.g., Assumption 3.3. in [9] and Assumption 1.3 in [10]). In the convex case, [3] is not ignored in Table 1 (Dore method in Table 1). This work requires $\frac{\omega}{\alpha n} \frac{L_{\max}}{\mu} \log \frac{1}{\varepsilon}$ rounds to converge. This complexity is strictly higher (= worse) than our guarantees (see 2Direction in Table 1).
> > >
> > > For example, the theory in [9] is strictly worse than the previously best-known theory for bi-directional compressed methods provided in [12]
> > >
> > > (also, there is a mistake in their proof - their final complexity is worse in its dependence on the contraction factor of the compressor). Their paper is in this sense not interesting from the theoretical point of view. However, it's an excellent empirical work. Our work massively improves on the previous theoretical SOTA from [6], which provides much better guarantees than [12]. So, in brief,
> > >
> > > **our theory >> theory from [6] > theory from [12] > theory from [9].**
> > >
> > > Such comparisons can be made with all other previous results., including [3], [8], [10], [11]...
> > >
> > > From the design's view, we discuss the difference between our approach and the previous approaches in detail in Section 4. Let us briefly repeat it here:
> > >
> > > *On the workers' side*, we do *not* use the "error compensated" compression. Our compression technique is based on works of \[5\], which is very different and only works with unbiased compressors.
> > >
> > > *On the server's side*, our compression technique is new. The difference of designs we discuss in Lines 167-176. Algorithmically, our approach and the "error compensated" compression are different. One can compare the formulas (8) and (9).
> > >
> > > [12] Fatkhullin et al, EF21 with Bells & Whistles: Practical Algorithmic Extensions of Modern Error Feedback, 2021

---

### Author Rebuttal · Authors · 2023-08-06

Dear AC and Reviewers,

Thank you for your work and effort. In this comment, we quickly clarify our main contribution to the community.

The field of optimization methods with compressed communication methods is very popular, rapidly growing, and will only increase in importance (e.g., see Jue Wang, Yucheng Lu, Binhang Yuan, Beidi Chen, Percy Liang, Christopher De Sa, Christopher Re, Ce Zhang. CocktailSGD: Fine-tuning Foundation Models over 500Mbps Networks, ICML 2023 where the authors fine-tune a large language model in  distributed manner, and bidirectional compression was essential).

There are hundreds, and maybe even thousands of papers that design optimization methods to reduce communication overhead.

**However, as far as we know, none of the previous papers provided theoretical communication guarantees that would be no worse than the communication guarantees of the vanilla accelerated Nesterov's method (AGD), let alone guarantees that can in some regimes be substantially better. In short, we believe that our work is an important theoretical breakthrough in an important subfield of ML.**

Before our paper, the best theoretical guarantees were obtained by AGD that **does not compress**. The communication complexity of AGD is $d \times \sqrt{\frac{L}{\mu}} \ln \frac{1}{\varepsilon}$ (= # of send bits/coordinates $\times$ # of iterations/rounds). **Our paper is the first one that broke this fundamental baseline in the bidirectional setting.**

**We believe that the low scores given to us by some reviewers do not reflect the quality and import of our work.**

Thank you again, we really appreciate your reviews and comments! Please ask us any questions!

authors

---

### Decision · Program_Chairs · 2023-09-21

**Decision:**

Accept (poster)

**Comment:**

This paper focuses on how to reduce the communication cost for accelerated gradient algorithms in the distributed setting. The proposed bi-directional compression accelerated gradient algorithm improves the communication cost of non-accelerated gradient algorithm. The theoretical result looks promising. The theoretical proof is nontrivial and uses many symbiotic computations.

The reviewers had a long discussion about the decision about this paper and did not achieve a consensus. Consider that this is the first work to explore how to optimize the communication cost for Nesterov's accelerated gradient algorithms in distributed setting (which is non trivial and challenging), we decided to accept this paper. In the meantime, we want to point out the readability and the clarity need to be improved. In particular,

It is a tough work to verify the correctness of the current proof. The theoretical proof needs to be simplified, although some portion of complexity is by nature of the (Nesterov's) accelerated gradient algorithm. For example, authors can compare the proof for AGD and use the key results in proving AGD as the reference. Symbolic computations are not encouraged to use in the proof, since it buries the proof insights and makes the proof look over of construction. Some symbolic computations can be replaced by choosing the learning rate appropriately to achieve the same purpose.

The algorithm can be more of structure. The current algorithm description is too fragmented and too long. The assumed background for readers can be with AGD. The comparison to other existing bi-directional compression algorithms such as [3, 9, 10] need to be discussed too. Provide the explanation why the biased compression operator can only be applied to one side, unlike [10].

Empirical study: although we treat this paper as a theoretical work, it does not mean the empirical study is not important. The most important result is still expected to be included in the main body of this paper.